# Metabolic modelling reveals the aging-associated decline of host–microbiome metabolic interactions in mice

Lena Best [1], Thomas Dost[1,15], Daniela Esser[1,2,15], Stefano Flor[1,15], Andy Mercado Gamarra [1,15], Madlen Haase[3,15], A. Samer Kadibalban[1,15], Georgios Marinos [1,4,15], Alesia Walker[5,15], Johannes Zimmermann [1,6,7,15], Rowena Simon[3], Silvio Schmidt [3], Jan Taubenheim [1], Sven Künzel[8], Robert Häsler[9,14], Sören Franzenburg [9], Marco Groth [10], Silvio Waschina[11], Philip Rosenstiel [9], Felix Sommer [9], Otto W. Witte[3], Philippe Schmitt-Kopplin [5,12], John F. Baines[8,13], Christiane Frahm[3,16] & Christoph Kaleta [1,16] ✉

Aging is accompanied by considerable changes in the gut microbiome, yet the molecular mechanisms driving aging and the role of the microbiome remain unclear. Here we combined metagenomics, transcriptomics and metabolomics from aging mice with metabolic modelling to characterize host–microbiome interactions during aging. Reconstructing integrated metabolic models of host and 181 mouse gut microorganisms, we show a complex dependency of host metabolism on known and previously undescribed microbial interactions. We observed a pronounced reduction in metabolic activity within the aging microbiome accompanied by reduced beneficial interactions between bacterial species. These changes coincided with increased systemic inflammation and the downregulation of essential host pathways, particularly in nucleotide metabolism, predicted to rely on the microbiota and critical for preserving intestinal barrier function, cellular replication and homeostasis. Our results elucidate microbiome–host interactions that potentially influence host aging processes. These pathways could serve as future targets for the development of microbiome-based anti-aging therapies.

Aging and aging-related diseases are central contributors to morbidity and mortality in Western societies[1]. Although research has identified specific hallmarks of aging[2] and revealed the conservation of aging-associated changes across species and tissues[3], the primary causative factors of aging remain elusive[2]. The microbiome, comprising a diverse bacterial community that resides within and on host organisms, is gaining recognition for its interplay with host aging processes. It is implicated in many aging-associated physiological processes[4], showing notable shifts in its composition as the host ages and strong correlations with aging-related phenotypes[5]. Microbiome transfer experiments revealed that introducing young microbiota to old hosts extends their lifespan[6,7] and reverses specific aspects of aging in animal models[8]. However, some studies have also shown beneficial effects of aged microbiota[9] or signatures specific to healthy aging in centenarians[6] that indicate that some aging-associated changes in the microbiota might also be compensatory by counteracting aging-associated

changes in the host[7]. Pathological changes in the host's gastroenteric system, such as obstipation, constipation and barrier dysfunction, are comorbidities of many aging-related diseases and often precede the manifestation of these diseases by many years[8]. Moreover, the aging-associated loss of intestinal barrier function, which facilitates the translocation of living bacteria and their products into the bloodstream, is implicated as a driver of systemic inflammaging, a hallmark of aging characterized by a constant low-grade inflammation even without the presence of a detectable pathogen[10,11].

However, it remains unclear which microbiome changes are causes of aging in the host and which are consequences[12]. The primary reasons for this uncertainty are the high plasticity and complexity of the microbiota, which comprise dozens to hundreds of species[13], the low species-level conservation of microorganisms across human cohorts[14] and the myriad of metabolites through which the microbiota and host can interact[15]. One approach to overcome this complexity is constraint-based metabolic modelling[16]. This method builds on in silico representations of the metabolic networks of individual species— so-called genome-scale metabolic networks—and allows the prediction of metabolic fluxes in individual species or entire communities[17]. This approach enables the integration of different types of omics datasets to derive context-specific metabolic networks (that is, networks representing the metabolic state of particular tissues or cells)[18]. Therefore, several studies have used constraint-based metabolic modelling to investigate changes in microbiome–host interactions in various diseases[19] and identify specific microbial processes linked to therapeutic response[17,20].

In this study, we used tissue transcriptomic, metagenomic and metabolomic data to elucidate the metabolic mechanisms through which the gut microbiota could contribute to host aging. We extensively characterized microbiome–host interactions at the level of global associations between host transcript levels and microbiome functions and then focused on metabolic interactions using an integrated metabolic model of the host and the microbiota. Our results revealed many known interactions between the host and the microbiota and postulated numerous hitherto unknown ones. Next we investigated how these interactions change in the context of aging. We observed a considerable reduction in microbiome metabolic activity with age, which seemed to be driven by substantial changes in within-microbiota ecological interactions. We subsequently connected aging-related changes in the host with alterations in the microbiota and discovered that aging-regulated gene networks were significantly enriched for both microbiome-dependent genes and microbiota-dependent host functions, as predicted by our models. These functions showed a marked decline with age. Our findings indicate that the microbiome is a major contributor to aging-associated metabolic decline, which we also observe at the metabolome level and thereby pinpoints metabolic pathways through which the microbiome may influence aging in the host.

## Results

### Taxonomic and functional description of the mouse microbiome

We studied the effects of aging in 52 male wild-type C57BL/6J/Ukj mice, separated into 5 age groups between 2 months and 30 months old, representing early adulthood until late age with ~5% survival[21]. We obtained transcriptome sequencing data for the colon, liver and brain, as well as shotgun (167 Gbp) and long-read sequencing data (13.7 Gbp) for faecal samples, which we used to reconstruct 181 metagenome-assembled genomes (MAGs; total 367 Mbp) of bacteria comprising their gut microbiome (Fig. 1a and Extended Data Fig. 1a–e). Taxonomic classification with the Genome Taxonomy Database Toolkit (GTDB-Tk)[22] assigned 175 MAGs to known taxa (with the prefix 'GCA_' or 'GCF_'), whereas 6 MAGs did not have a matching genome (prefixed 'UNK_'). Of those 181 MAGs, 25 were considered high-quality drafts according to established criteria[23], and the rest were considered medium-quality drafts.

Notably, we used more stringent cut-offs (≥80% completeness and ≤10% contamination) than those suggested in ref. 23 for medium-quality MAGs to require less gap filling and thus obtain more reliable metabolic models for downstream analysis (Fig. 1a). Most of the MAGs were attributed to the phyla Bacillota (previously Firmicutes; $n = 97$) and Bacteroidota ($n = 65$). The reconstructed genomes from rarer phyla included Pseudomonadota (previously Proteobacteria; $n = 9$), Cyanobacteriota (previously Cyanobacteria; $n = 4$), Campylobacterota ($n = 3$), Deferribacterota ($n = 1$), Desulfobacterota ($n = 1$) and Verrucomicrobiota ($n = 1$). Regarding overall abundance, the most abundant MAGs, with a coverage depth >1%, belonged to Bacteroidota in the families Bacteroidaceae ($n = 5$) and Muribaculaceae ($n = 12$). The genome sizes of the MAGs ranged from 0.9 Mbp to 6.7 Mbp (Fig. 1a).

To functionally annotate the assembled MAGs, we used gapseq[24] to reconstruct their corresponding genome-scale metabolic networks. In a principal component analysis of the networks (Fig. 1b), principal component (PC) 1 mainly separated models by the completeness score ($R^2 = 0.15$) of the underlying MAGs and the taxonomic rank 'order' ($R^2 = 0.87$). The completeness of the MAGs significantly impacted the prevalence of pathway gaps within the models. Consequently, the occurrence of such gaps ($R^2 = 0.55$) and the sizes of the models ($R^2 = 0.84$) or genomes ($R^2 = 0.58$) partially accounted for the observed differentiation along the first two PCs. PC2 separated the metabolic models by the phylum, GC content ($R^2 = 0.29$) and contamination score ($R^2 = 0.06$).

### Microbiome functions correlate with host transcripts

After reconstructing the metabolic models of the bacterial species of the mice, we first determined host functions associated with microbiome functions independent of age. Filtering by association strength and a false discovery rate (FDR)-adjusted $P \leq 0.1$, we identified 12,732 correlated microbiome reactions and host genes for the colon, 3,425 for the liver and 2,499 for the brain. Enriching these features with gene ontology (GO)[25] biological processes (host genes), and metabolic subsystems (microbiome reactions), we obtained 1,377 pairs of host–microbiome-associated processes for the colon, 283 for the liver and 167 for the brain; we further summarized these with level 2 GO biological processes and MetaCyc[26] superpathways (Fig. 2a–c, Extended Data Fig. 2a and Supplementary Tables 2.1–2.4).

The most strongly correlated host functions for the colon involved innate and adaptive immune processes and protein processing (Fig. 2a, Extended Data Fig. 2a and Supplementary Table 2.1). These included a negative correlation between host immune system processes and microbial galactose and arabinose degradation pathways. Moreover, we observed strong positive correlations between microbial purine metabolism and mitochondrial respiration in the host. Furthermore, we found that microbial pathways involved in lipid metabolism were correlated with host processes involved in tissue homeostasis, such as DNA damage responses and cell death. By directly inferring functions from quality-controlled metagenomic read data (HUMAnN3)[27], we found fewer, yet comparable, host–microbiome associations (Extended Data Fig. 2b and Supplementary Table 2.7). For the liver, we detected associations between central metabolic pathways of the microbiota and chromatin organization in the host as well as between T cell proliferation and microbial branched-chain amino acid metabolism (Fig. 2b, Extended Data Fig. 2a and Supplementary Table 2.2). For the brain, we found strong correlations between protein catabolic processes and microbial nucleotide metabolism (Fig. 2c, Extended Data Fig. 2a and Supplementary Table 2.3). To independently validate host genes associated with microbiome functions, we determined their regulation in response to microbial colonization. To this end, we generated transcriptomic data from five tissues (colon, liver, brain, gonadal white adipose tissue and quadriceps) of three groups of mice: conventionally raised wild-type mice (CONVR), germ-free (GF) mice and mice conventionalized with faecal material from WT mice (CONVD; $n = 8$ per group). Comparing

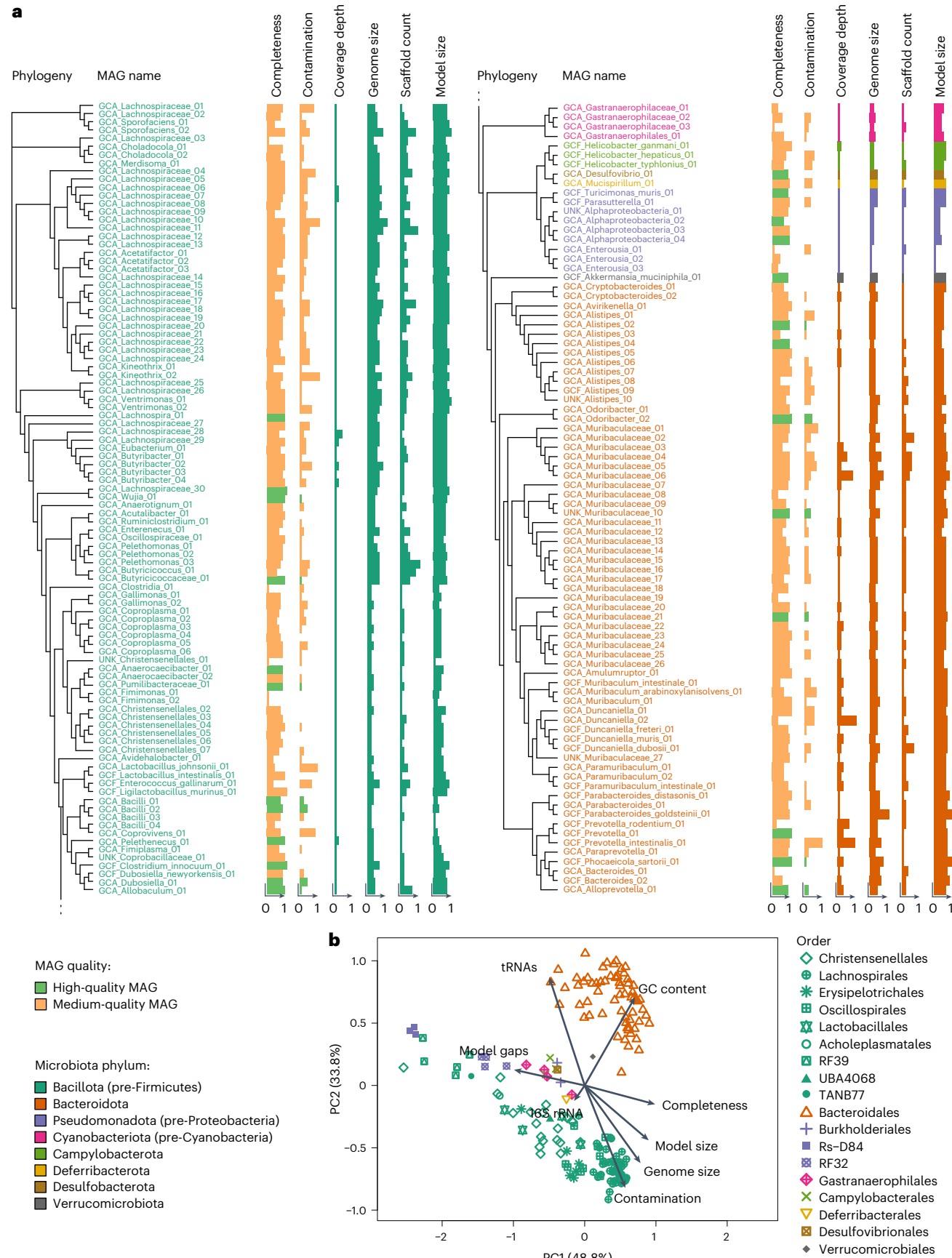

**Fig. 1 | Mouse microbiome and metabolic model characterization.**
**a**, Phylogenetic tree of the 181 MAGs. See Supplementary Table 1.2 for detailed metadata information. **b**, Principal component analysis of the metabolic models of the mouse microbiota. Metadata associations to PCs are overlaid as arrows; the shapes denote the taxonomic rank order; the colouring of the symbols are according to **a**.

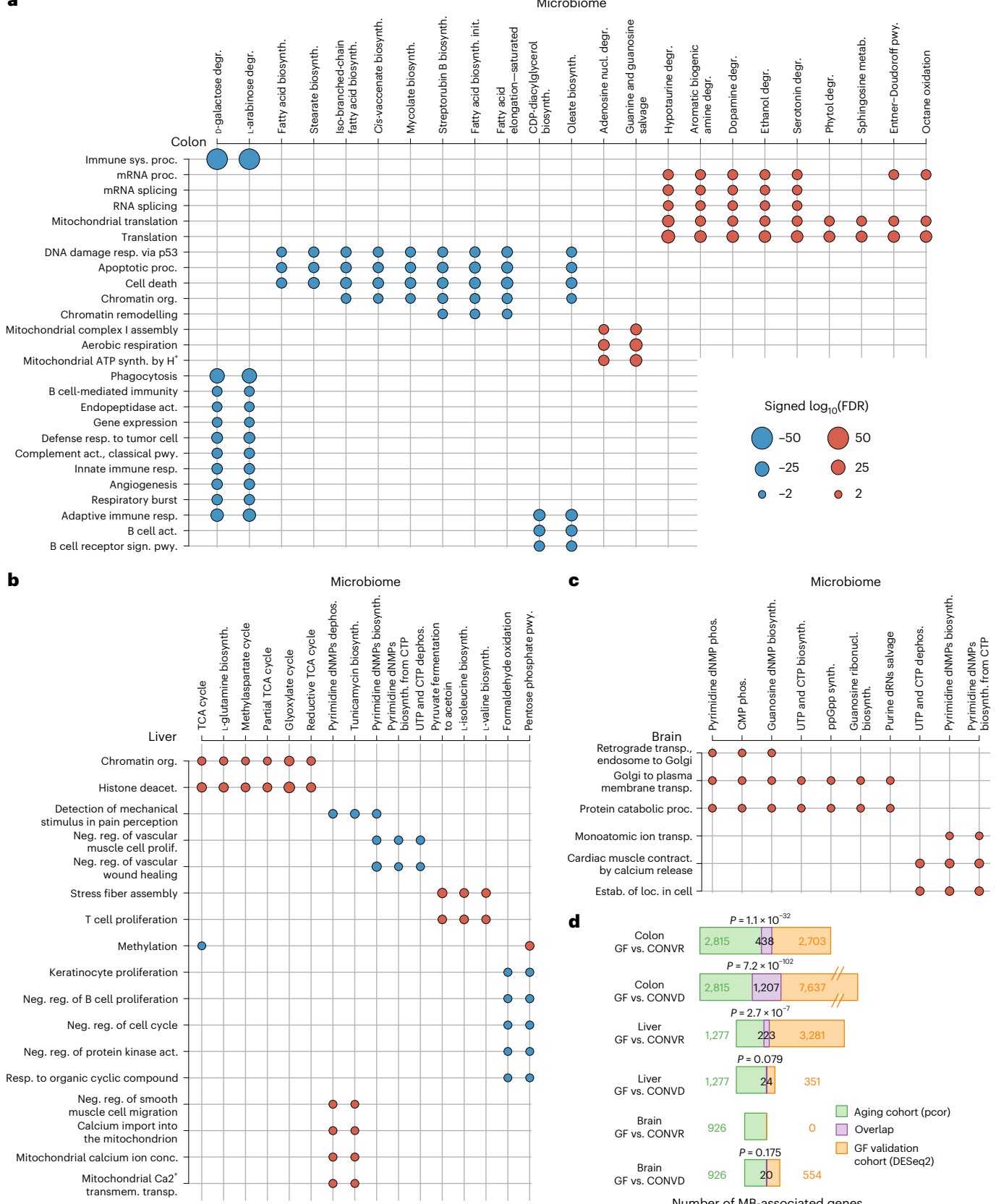

**Fig. 2 | Correlation-derived host–microbiome interactions. a–c**, Interactions between host biological processes and microbiome metabolic subsystems for the colon (**a**), liver (**b**) and brain (**c**). Only processes with the most significant associations (colon FDR ≤ 1 × 10⁻¹⁰, liver FDR ≤ 1 × 10⁻⁴ and brain FDR ≤ 1 × 10⁻³) with at least two interactions are shown. For complete data and full pathway names, see Supplementary Tables 2.1–2.4. **d**, Enrichment of microbiome-associated host genes among microbiome-colonization-responsive genes. FDR-corrected *P* values from upper-tailed (one-sided) hypergeometric tests of the overlap plotted above each bar.

genes responsive to microbial colonization (comparison CONVR versus GF and CONVD versus GF; Supplementary Tables 2.8–2.12) and those associated with microbiome function, we found a highly significant overlap in colon and liver, but not in brain (Fig. 2d). The lack of significant overlap with microbiome-responsive genes in the brain might be partially due to the relatively small number of microbiome-responsive genes in this tissue (Supplementary Table 2.12).

#### Host–microbiome interactions in the metaorganism model

We next aimed to gain a more mechanistic understanding of the underlying metabolic pathways mediating host–microbiome associations with an integrated metabolic metamodel of the host and the microbiome. In this metamodel, the host is represented by three different tissues (colon, liver and brain) connected through the bloodstream and interacting with the microbiome through the gut lumen (Fig. 3a). Each host tissue is represented by a unique instance of the human metabolic reconstruction Recon 2.2 (ref. 28), whereas the microbiome is represented by a combined model including all the metabolic reactions occurring in at least one bacterial metabolic model reconstructed from the MAGs (Fig. 3a). Subsequently, context-specific metabolic metamodels representing the metabolic state of each mouse were built based on tissue transcriptomic and metagenomic data using fastcore[29] (Supplementary Table 3.1).

To explore the extent to which the metamodel could reconstitute known host–microbiome interactions, we used it to predict metabolites exchanged between host and microbiota (Fig. 3b and Supplementary Table 3.2). In the colon, we observed many known interactions including a provision of the microbiome with bile acids as well as fucose, a part of mucins[30], by the host and a microbial production of short-chain fatty acids. Moreover, we observed that the microbiota produced many nucleotides, including nucleotide derivatives such as NAD and coenzyme A. For the liver, we observed a provision of primary bile acids to the microbiota and microbial production of nucleotides and short-chain fatty acids. For the brain, we observed that the host was provided with the microbial fermentation product ethanol and several pyrimidines. The brain and colon provided the pyrimidine precursor orotate and the nucleotide degradation product uracil to the microbiota, while the microbiota provided uridine and deoxycytidine in return. Overall, we found that among the predicted interactions shown in Fig. 3b, 42 (51%) were already supported by previous experimental evidence across all three organs (Supplementary Table 3.9), thereby strongly supporting the ability of the metamodel to capture metabolic microbiome–host interactions.

To elucidate the underlying metabolic pathways connecting the host and microbiota that might mediate the extensive associations we have observed, we sampled elementary flux modes (EFMs)[31] in the metamodel with the EFMSampler[32]. Each host reaction was defined as an indicator reaction through which EFMs were sampled. By recording the frequency at which microbial reactions occurred in the EFMs of a host indicator reaction, we obtained an interaction matrix of the frequency at which microbiome reactions occurred in the pathways sampled for individual host indicator reactions. Using these interaction matrices, we found that correlated host gene–microbiome reaction pairs (see Fig. 2) had a higher frequency of model-predicted interactions compared with randomly sampled pairs for liver and colon, but not for brain (Extended Data Fig. 3a). These findings suggest a coupling of host metabolic transcription and microbiome metabolic functionality, even though this might be biased by the utilization of the same data basis for determining host–microbiome correlations and reconstructing the metamodel.

To validate the metamodel's ability to identify microbiome–host interactions, we examined how microbial colonization influenced predicted microbiome-dependent host reactions using gene expression data from our GF mouse cohort. In addition to the colon, liver and brain, we analysed gonadal white adipose tissue and quadriceps

to assess whether microbiome dependency in one tissue could inform predictions for others. For each tissue, we identified upregulated, downregulated and unregulated genes, mapped them to reactions and evaluated their predicted microbiome dependency. Upregulated reactions showed significantly higher microbiome dependency than unregulated reactions in seven comparisons, while downregulated reactions showed higher dependency in three (Fig. 3c). Only two instances showed lower dependency in regulated reactions, strongly supporting the ability of the metamodel to capture functional host–microbiome interactions. To show that these results were not due to modelling-inherent biases, we repeated the analysis with gene labels randomized and did not find a single case with a higher number of significant associations across 1,000 randomized repetitions (Extended Data Fig. 3b). We further validated the metamodel by showing a strong correlation between model-predicted microbiome dependence of serum metabolites and microbiome-driven variance of those metabolites in an independent human metabolomics cohort[33] (Spearman's $\rho = 0.43$, $P = 1.5 \times 10^{-3}$; Extended Data Fig. 3c and Supplementary Table 3.10).

To further functionally characterize the host–microbiome-interaction matrix, we performed enrichments for host and microbial metabolic subsystems (Fig. 3d–f). In the colon, we found host pathways associated with energy metabolism, nucleotide metabolism, vitamin metabolism and amino acid metabolism (Fig. 3d) depending on fermentation products, nucleotide metabolism and vitamin biosynthesis pathways of the microbiota. In the liver, energy-producing pathways and bile acid synthesis were prominent on the host side and fermentation pathways on the microbiome side (Fig. 3e). In the brain, microbiome-dependent host reactions were enriched in nucleotide metabolism, folate metabolism and the metabolism of neurotransmitter precursors, such as tryptophan and tyrosine (Fig. 3f). Although most host–microorganism interactions were relatively generic, relying on basic microbial metabolic functions (such as glycolysis and fermentation), we also identified specific interactions, such as colonic nucleotide interconversion dependent on microbial ATP synthesis and colonic coenzyme A catabolism reliant on microbial production of phosphopantothenate, a coenzyme A precursor.

#### Aging is linked to reduced microbiome metabolic activity

We next explored functional and taxonomic changes in the aging microbiome. Consistent with previous reports in mice[34,35], we observed that age was associated with a decrease in the abundance of Bacillota and an increase in Bacteroidota (Fig. 4a), also when inferring taxonomic changes from metagenomic data directly (Extended Data Fig. 4a and Supplementary Table 4.15). To obtain a better functional understanding of these species-level changes, we used community flux balance analysis (FBA)[17] to predict microbial metabolic activities (Supplementary Methods). In contrast to the metaorganism modelling approach used in the previous section, which does not differentiate between individual microbial species owing to computational limitations, community FBA models each microbial species individually. Summarizing age-associated reactions on the pathway level, we mainly observed negative associations (Fig. 4b) involving many biosynthetic pathways essential for bacterial replication, such as synthesis of amino acids, nucleotides, vitamins and cell wall components. Similarly, for metabolic interactions between the microbiota and the host as well as within the microbiota, we mainly observed strong reductions in both the consumption and production of metabolites (Fig. 4c and Extended Data Fig. 4b), including the production of the short-chain fatty acid butyrate, and increased production of few metabolites, including pro-inflammatory succinate[36]. Consistent with a generally reduced microbial metabolism, we also found that model-predicted and metagenomics-derived microbial growth rates decreased considerably with age (Fig. 4d) and were strongly correlated (Extended Data Fig. 4c). Furthermore, we evaluated the change of FBA-predicted community

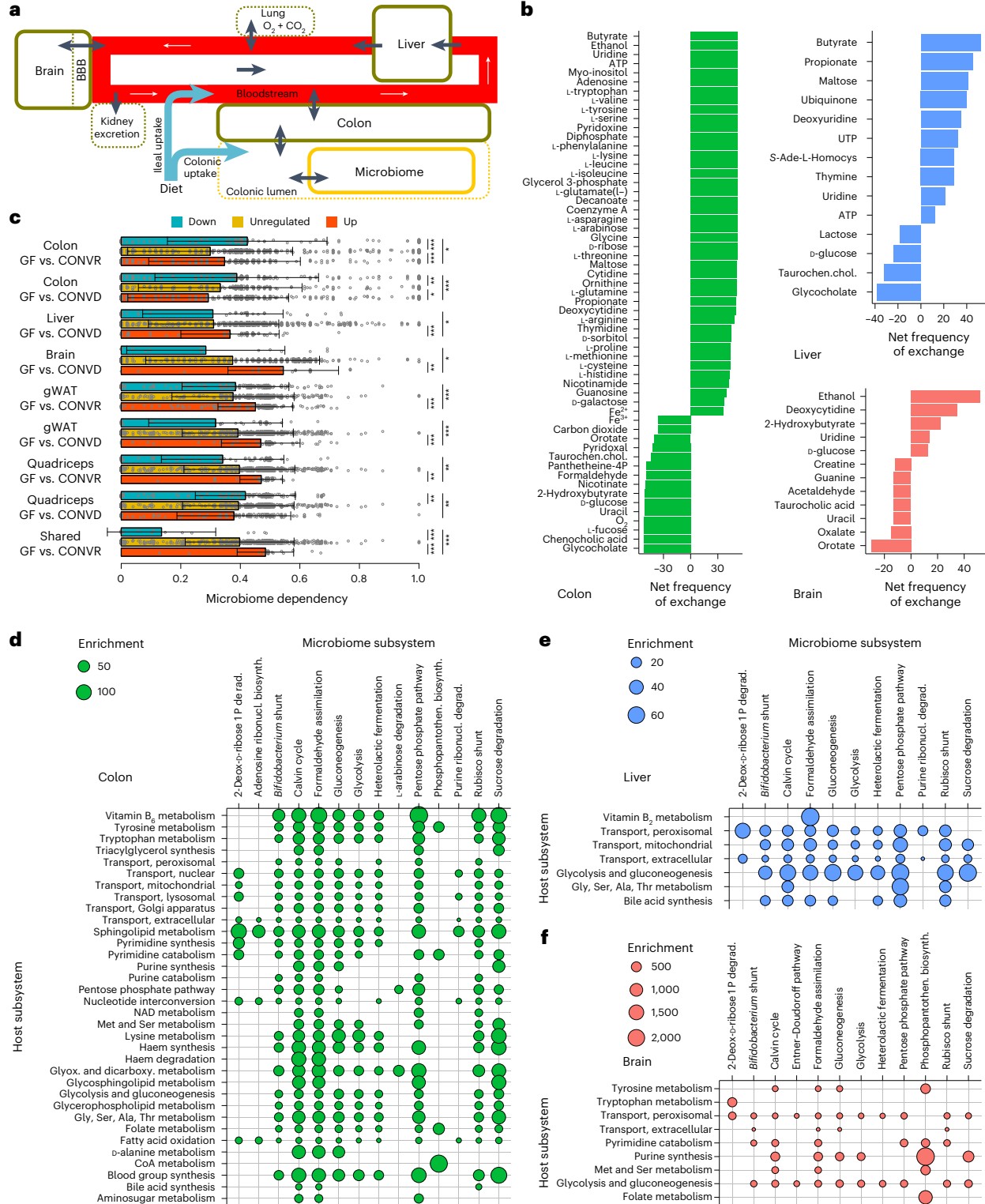

**Fig. 3 | Model-predicted host–microbiome interactions. a**, Structure of the metamodel. The solid borders indicate compartments of the metamodel. The black arrows indicate metabolite exchanges between compartments. The dashed borders indicate compartments represented only by exchange reactions. The white arrows indicate the direction of metabolic exchanges along the bloodstream. BBB, blood–brain barrier. **b**, Frequency of microbiome dependence of metabolite import (positive) and export (negative) across organs. Metabolites with the highest frequency of exchange across 52 models are shown (Supplementary Table 3.2). For metabolite abbreviations, see Supplementary Table 3.2. **c**, Microbiome dependency of microbiome-responsive host genes in a cohort of GF, conventionalized and conventionally raised mice (n = 8 each). The y axis indicates sets of genes

differentially regulated in tissues and contrasts; the x-axis shows the microbiome dependency of corresponding reactions. 'Shared' indicates genes regulated in at least three tissues. FDR-corrected P values of Dunn's tests following a group-level Kruskal–Wallis test are shown next to the bar plots of means with error bars representing the standard deviation. Only comparisons with a Kruskal–Wallis test $P < 0.05$ are shown. $*P < 0.05$; $**P < 0.01$; $***P < 0.001$. Exact P values are provided in Supplementary Table 3.11. **d**–**f** Subsystem enrichment of model-predicted interactions between host and microbiome reactions for subsystems connected with at least two host subsystems and an FDR-corrected enrichment $P < 10^{-4}$ (one-sided Fisher's exact test; Supplementary Tables 3.3–3.5). For pathway abbreviations, please see Supplementary Tables 3.3–3.5.

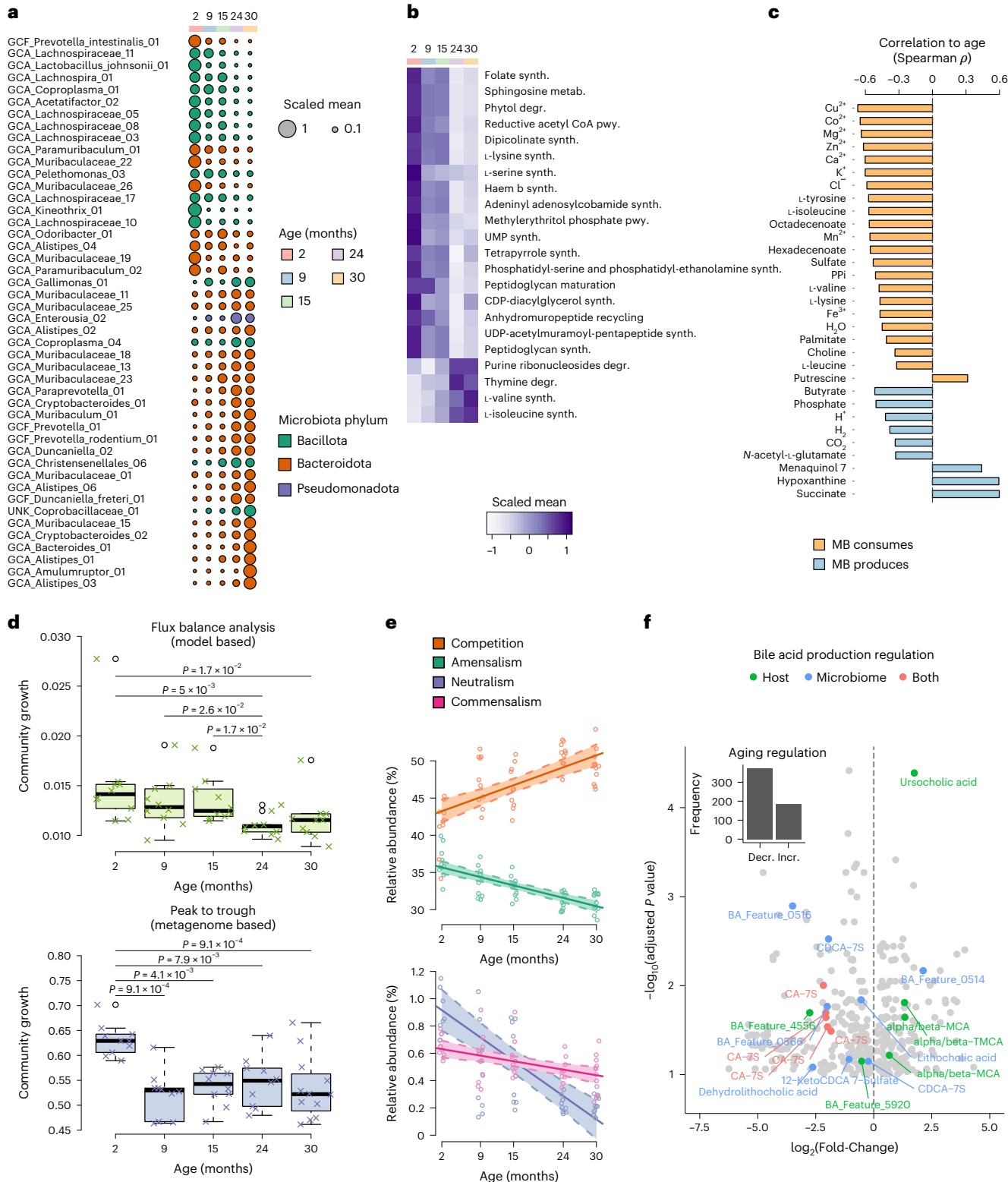

**Fig. 4 | Microbiome alterations associated with host age. a**, Aging-associated changes in MAG abundance. **b**, Subsystem-level aging-associated changes in microbiome internal reaction fluxes. **c**, Aging-associated changes of host–microbiota metabolic exchange. **d**, Comparison of microbiome community growth rates derived from FBA or the PTR (30 months: n = 12; all others, n = 10; FDR-corrected P values from Dunn's test following Kruskal–Wallis test). **e**, Aging-associated changes in model-predicted ecological interactions in the microbiota. Linear-model-derived regression with 95% confidence intervals (30 months: n = 12; all others, n = 10). **f**, Aging-associated changes in faecal metabolite concentrations in mice. All age-associated metabolites are shown (FDR-adjusted P ≤ 0.1 from Spearman correlations; $\log_2$(fold change (FC)) of 3 to 28 months; 3 months: n = 15; 9 months: n = 16; 15 months: n = 15; 24 months: n = 17; 28 months: n = 18; Supplementary Table 4.12). The origin of bile acids is indicated. 'Both' refers to bile acids produced by the host but regulated by the microbiota. Metabolites with the prefix 'BA_Feature' have not been fully resolved. incr., increase; decr., decrease; CA-7S, cholic acid-7-sulfate; CDCA-7S, chenodeoxycholic acid-7-sulfate; MCA, muricholic acid; TMCA, tauromuricholic acid. Box plot elements: centre line, median; box limits, 25–75% quantiles; whiskers, 1.5× interquartile range (IQR); points, outliers.

growth upon removal of single bacterial members, and observed that MAGs suppressed in aging had a beneficial effect on community productivity and growth, while MAGs which were enriched in old mice showed a negative impact (Extended Data Fig. 4d,e).

To gain insight into the potential microbiome-intrinsic causes of the observed aging-associated suppression of metabolism, we used community FBA[17] to predict the frequencies of ecological interactions (Methods). We observed a significant decrease in amensal, commensal and neutral interactions at the expense of increased competitive interactions (Fig. 4e). These shifts in community interactions were also observed at the level of individual microbial ecological strategies derived from the universal adaptive strategies theory framework[37,38], indicating a shift in the community towards the dominance of ruderals, which are first colonizers of niches and poor interaction partners owing to reduced catabolic diversity[37] (Extended Data Fig. 4f).

To further explore the predicted loss of microbiome metabolic activity and metabolic cooperativity with age, we performed an untargeted metabolomics analysis of faecal samples from an independent cohort of 82 mice across all age groups. We determined the correlation between the abundance of identified metabolomic features and age and found that 374 of 561 features (67%) showed significant downregulation (FDR-adjusted $P \leq 0.1$; Fig. 4f). Within this dataset, we specifically annotated bile acids using reference standards because of their previously documented role in host aging[39]. Consistent with the reduction in microbiome metabolic activity with age, we found that the concentrations of host-regulated bile acids were significantly increased (four out of six features). By contrast, the concentrations of microbiome-regulated bile acids were mostly reduced (seven out of eight features). Intriguingly, metabolomic features annotated as cholic acid-7-sulfate, which is produced by the host but regulated by the microbiota[40], were exclusively downregulated with age. Also, further microbiome-regulated metabolites, including valine, betaine, nicotinamide, enterolactone and 3-hydroxykynurenine, were downregulated with age (Supplementary Table 4.12). Moreover, we found an increase in the pro-inflammatory microbial metabolite D-galactose, for which we observed a strong association with host immune processes in the colon (Fig. 2b), although only significant before FDR correction (Extended Data Fig. 4g).

### Aging decline of microbial metabolism impacts host functions

Next, we investigated how the aging-associated loss of microbiome metabolic function potentially impacted host functions. To this end, we used differential gene expression analysis and GO term enrichment to identify aging-regulated genes and processes. Consistent with our previous work, we found a considerable conservation of aging-regulated genes across tissues[3] including 157 transcripts that were consistently downregulated and 526 genes that were consistently upregulated. Upregulated genes were mostly enriched for immune-associated processes and downregulated genes in cellular maintenance and tissue regeneration processes (Fig. 5, Extended Data Fig. 5a–d and Supplementary Tables 5.1–5.9). Exploring connections between aging and the microbiome, we found a highly significant enrichment of microbiome-correlated transcripts among aging-regulated genes across all tissues (Fig. 6a). Along with the loss of microbiome metabolic function with age (see Fig. 4), we also found a pronounced loss of host–microbiome associations with age (Extended Data Fig. 6a–e). We found a higher number of GO biological processes for both aging and microbiome-associated genes in the colon than in the liver and brain (Fig. 6b, Extended Data Fig. 6f,g and Supplementary Tables 6.1–6.4). Notably, tissue homeostasis and organ regeneration processes were downregulated in the colon with age but positively correlated with microbial metabolic pathways. Conversely, aging-induced processes, primarily defence, inflammatory and immune responses, were negatively associated with microbial metabolism (Fig. 6b). Brain development was negatively correlated with microbial metabolism and

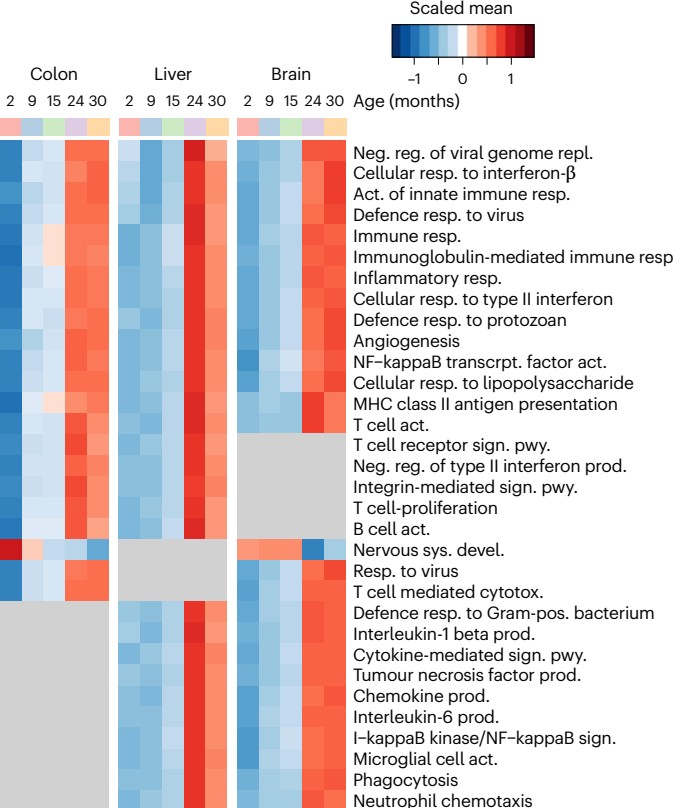

**Fig. 5 | Aging-associated transcriptomic changes across host tissues.** Enriched GO biological processes, shared by at least two organs, are shown as the average expression of all associated features, stratified by age group and organ (hypergeometric test FDR cutoff for displayed terms: colon, $10^{-4}$; liver, $10^{-6}$; brain $10^{-6}$; 30 months: $n = 12$; all others $n = 10$). For complete data and full pathway names, see Supplementary Tables 5.4–5.6.

downregulated with aging (Extended Data Fig. 6g). On the microbial side, glycolysis, nucleotide synthesis and D-galactose degradation were suppressed with age and mostly negatively correlated with host gene expression (Fig. 6c). In connection with the aging-associated increase in microbial production of the pro-inflammatory metabolite succinate, observed in community modelling (see Fig. 4c), we also identified many aging-associated changes in succinate-metabolizing microbial pathways correlated with host gene expression (for example, oxalate and itaconate degradation).

Given the observed reduction in microbiome–host associations, we next aimed to identify the underlying metabolic pathways potentially mediating those changes. To achieve this, we defined aging-regulated metabolic modules among the metamodel's metabolic reactions using the host–microbiome-interaction matrix obtained from EFM sampling. Metabolic modules were determined according to sampled EFMs (compare with Fig. 3d–f), selecting reactions present in at least 20% of the EFMs for each indicator reaction (Methods). Aging regulation of those modules was then inferred from the over-representation of aging-regulated reactions in each module. In the colon, liver and brain, we identified aging-induced (51, 88 and 99, respectively) and aging-repressed (2,509, 1,702 and 524, respectively) metabolic modules (Supplementary Tables 6.5–6.7), with aging-repressed modules being significantly more dependent on the microbiome across all tissues (Fig. 6d). These modules revealed downregulation of colon metabolic pathways linked to fatty acid oxidation, N-glycan synthesis and sphingolipid metabolism, which are central to cellular homeostasis. In the liver, downregulated modules were enriched in bile acid synthesis and triacylglycerol synthesis, aligning with age-related shifts in bile acid profiles (see Fig. 4f). In the

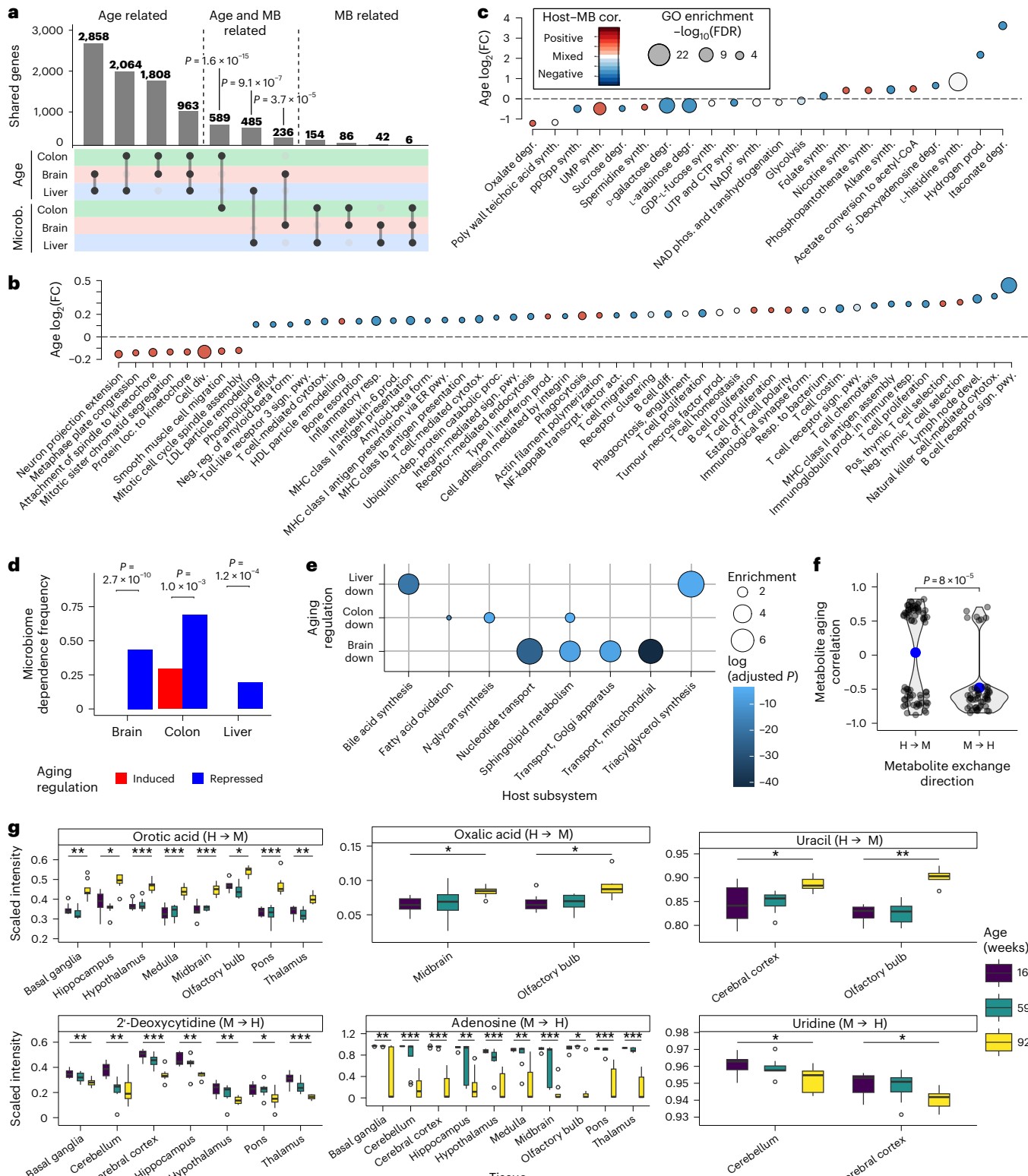

**Fig. 6 | Aging-associated changes in host–microbiome interactions.**
**a**, Overlap between aging-regulated and microbiome-regulated host genes
(*P* values via upper-tailed hypergeometric test). **b**, Colon-specific gene
expression changes with age in processes correlated with microbiome metabolic
functions (legend shared with **c**). **c**, Aging-dependent changes in microbiome
processes correlated with host gene expression. For complete data and
full pathway names, see Supplementary Tables 6.1 and 6.4. **d**, Frequency of
microbiome dependence of aging-regulated metabolic modules across host
tissues (*P* values via one-sided Fisher's exact test). **e**, Subsystem-level enrichment
of indicator reactions of aging-regulated metabolic modules (one-sided

Fisher's exact test). The *x* axis represents enriched host subsystems; the *y* axis
represents aging-regulated gene sets. **f**, Aging association of brain metabolites
predicted to be exchanged between the microbiome and host. Data from ref. 41
(Supplementary Table 6.9). *P* values obtained by two-sided Wilcoxon rank-sum
test. **g**, Aging-associated metabolome changes for selected model-predicted
microbiota-produced and microbiota-consumed metabolites (*P* values via
Kruskal–Wallis test). Data from ref. 41 (*n* = 64 mice). Box plot elements: centre
line, median; box limits, 25–75% quantiles; whiskers, 1.5× IQR; points, outliers.
Significance: *\*P* ≤ 0.05; *\*\*P* ≤ 0.01; *\*\*\*P* ≤ 0.001. Exact *P* values are provided in
Supplementary Table 6.9.

brain, aging-regulated modules were enriched in nucleotide as well as sphingolipid metabolism and transport pathways (Fig. 6e and Supplementary Table 6.8). Given the strong effect of aging-suppressed microbiome metabolism on the brain (Fig. 6d,e), we analysed its impact on the brain metabolome using a public mouse dataset[41]. Correlations between metabolite concentrations and mouse age revealed that metabolites provided from host to microbiota accumulated with age, while microbiota-derived metabolites were depleted (Fig. 6f). This included accumulation of nucleotide precursors such as orotate and uracil, and depletion of salvage pathway products such as adenosine, 2-deoxycytidine and uridine (Fig. 6g).

## Discussion

In this study, we performed a model-based analysis of aging-associated alterations in host–microbiota interactions in mice. We reconstructed 181 MAGs using shotgun and long-read sequencing, converting them into constraint-based metabolic networks. Our investigation revealed extensive associations between microbiome functions and the host's colon, liver and brain transcriptome. Many correlations involved host immune processes, mitochondrial function and chromatin modification, alongside microbiome-derived metabolites such as D-galactose, known to promote neurodegeneration and inflammation[42], and leucine, a regulator of T cell function[43]. Associations also highlighted microbial fermentation and nucleotide metabolism, consistent with the roles of microorganism-produced short-chain fatty acids and nucleotides in colonic energy balance[44] and intestinal barrier function[45] (Extended Data Fig. 7).

A key aspect of our analysis was the reconstruction of metabolic metaorganism models, adapted from whole-body metabolic models for humans[46]. These models successfully recovered well-documented microbiome–host interactions involving short-chain fatty acids, bile acids and other microbial metabolites. Notably, 51% of the predicted high-confidence interactions were corroborated by existing literature. Furthermore, genes found to be regulated by microbial colonization also showed a higher microbiome dependence in the metamodels, underscoring the metamodel's ability to infer host–microbiome interactions accurately.

Examining reaction-level dependencies via EFM analysis revealed that the host most often depended on central metabolic reactions of the microbiome. This strategy may reduce reliance on specific bacterial species, broadening the potential pool of interaction partners[47], consistent with conserved gut microbiome functions across human cohorts[48]. However, the focus on central pathways might also reflect biases in the model's representation of bacterial metabolism.

Another interesting aspect of the predicted microbiome–host exchanges are metabolites that the host can produce itself, such as nucleotides. The reasons for the existence of such exchanges could include advantages from a division of labour, as frequently observed within microbial communities[49], a reliance of the host on the microbiota as a metabolic backup system to increase phenotypic plasticity[50], or evolutionary addiction, whereby mutual dependencies develop owing to the constant exposure of the host to microbially produced metabolites[51].

Aging-associated changes of the microbiome communities revealed increases in Bacteroidota and decreases in Bacillota species, reflecting human studies linking Bacteroidota persistence to poorer health and Bacillota enrichment to healthier aging[5]. Furthermore, we observed reduced microbiome growth and metabolic activity, specifically for the production of butyrate as key changes in aging. This aligns with findings of decreased serum butyrate levels in aged mice and humans[35,52]. By contrast, metabolic modelling predicted increased production of the pro-inflammatory metabolite succinate[53], a known indicator of a dysbiotic gut environment[54], which we found to be associated with key processes deregulated in aging on the host side including, as previously reported, DNA damage response[55] and protein

homeostasis[56]. Reduced microbial growth could underlie increased constipation risk[57] and longer colonic transit times[58] observed during aging. Reduced microbial growth and a decreased capacity to turn nutrients into biomass have previously also been observed as a direct effect of increased transit times in a bioreactor setup mimicking the colon[59].

Our analysis suggested that age-related changes in gut ecology involved increased competition and decreased cross-feeding, reducing dietary resource utilization efficiency. These trends were reflected in faecal metabolomics, in which most metabolic features decreased with age. Host-regulated bile acids increased, while microbiome-regulated bile acids declined. Anti-inflammatory metabolites such as valine, betaine and 3-hydroxykynurenine[60–62] decreased, while pro-inflammatory metabolites such as D-galactose increased[63].

Aging-associated inflammation and suppressed cellular replication across host tissues were consistent with our previous findings on conserved aging signatures[3]. Colon-specific changes in the transcriptome, supported by previous studies, included altered gut motility[64], reduced colonic barrier function[65] and proliferation[66], while the liver showed a decrease in mitochondrial biogenesis, as previously observed[67].

Aging also considerably affected host–microbiome interactions. Aging-regulated host genes were enriched for those correlated with microbiome functions, particularly in downregulated metabolic modules central to cellular homeostasis, such as fatty acid oxidation and nucleotide uptake. These findings align with our correlation analyses showing positive associations between microbial metabolism and host homeostasis and negative associations with inflammation. We previously observed a similar loss of microbiome–host interactions as a key component of pathology in inflammatory bowel disease[20]. The observed loss of host–microbiome interactions across all organs indicates that the microbiome might contribute to crucial aspects of the systemic aging process, such as metabolic decline[68,69] and the loss of cellular proliferation, along with stem cell exhaustion[2].

Microbially produced nucleotides emerged as a key metabolic exchange, with model predictions indicating host provision of precursors (for example, orotate) and degradation products (for example, uracil) to the microbiota, which provided nucleotides in return. This exchange aligns with our observation of widespread correlations across all tissues between host gene expression and microbial nucleotide metabolism. Despite host capability for de novo nucleotide synthesis, recent studies emphasize the microbiota's contributing role, particularly in the colon[45,70]. Bacterial species such as Escherichia coli and Bacteroides spp. actively excrete ATP, non-lytically, during growth[71,72], and bacterial ATP contributes to intestinal barrier function[45] as well as immune modulation via purinergic receptors[70]. An aging-associated decline in nucleotide co-metabolism could underlie diminished intestinal barrier integrity[45], linked to age-related diseases[73,74], reduced systemic proliferative capacity[75] and impaired mitochondrial function[76]. Furthermore, microbiome involvement in brain nucleotide salvage, crucial for DNA repair and cellular homeostasis[77], might relate to neurodegeneration[78].

In summary, we identified pronounced aging-associated changes in microbiome–host interactions, largely driven by reduced microbial metabolic activity. Although limited by its reliance on modelling, our study validated many interactions through independent analysis and literature, offering insights into the systemic aging process. Notably, while metabolic metamodels identify specific metabolite exchanges, transporter promiscuity and modelling limitations may imply the exchange of structurally related compounds in vivo. A further limitation of our study was the exclusive use of male mice, as the logistical challenges of establishing a separate aging cohort for females precluded the inclusion of both sexes. Focusing initially on males ensured consistency by avoiding sex-specific variability. Consequently, sex-specific changes were not investigated in this study but will be considered in future research. Besides chronological age, future work should also incorporate epigenetic clocks and biological age markers

such as frailty, loss of motor function and cognitive decline[79]. Finally, our identification of a loss of microbiome metabolic activity indicates a potentially crucial aging-associated change that could contribute to many aging-associated pathologies in the host. Therefore, microbiome metabolic activity could be a target for future microbiome-based therapies. Our modelling approach could play a crucial role in designing targeted interventions aimed at mitigating microbiome-driven aspects of aging.

## Methods

### Mouse strains

**Main study and metabolomics cohorts.** The mice used for the aging study were an in-house strain derived from the C57BL/6J strain (The Jackson Laboratory). These C57BL/6J/Ukj mice lack two common mutations found in the C57BL/6J strain: the DIP686 mutation in the crumbs family member 1 (*Crb1*) gene, which is vital for eyesight in aging mice, and a mutation in the nicotinamide nucleotide transhydrogenase (*Nnt*) gene, which encodes mitochondrial NAD(P) transhydrogenase, protecting against oxidative stress. Preserving both these genes is advantageous for metabolic and aging studies in mice.

**GF mice cohort.** The GF mice used for the analysis of host responses to microbial colonization were rederived axenic conventional Jackson Laboratory C57BL/6J, strain 000664 mice (The Jackson Laboratory). The mice were housed in the Experimental Biomedicine facility at the University of Gothenburg, Sweden.

### Animal handling

**Main study cohort.** Male C57BL/6J/Ukj mice were bred in the Central Experimental Animal Facility at Jena University Hospital (Jena, Germany). The mice were housed at $22 \pm 2\ °C$ with a 14:10 h day–night cycle and a relative humidity of $55\% \pm 10\%$. They were co-housed according to their birth cohort (similar ages) in standard cages (GM500, Type III; Tecniplast Deutschland), and a maximum of two mice from the same cage were used for experiments. The mice had unlimited access to water and food (mouse V1534-300, ssniff Spezialdiäten). Next-generation RNA sequencing of host tissues and metagenomics of faecal samples were conducted in 52 mice of different ages spanning the mouse's adult lifespan (2–3 months (mean = 2.5 months), 9–10 months (mean = 9.8 months), 15–17 months (mean = 15.9 months), 24–25 months (mean = 24.8 months) and 28–31 months (mean = 29.1 months)). For simplicity, the five age groups are referred to as 2 months (n = 10), 9 months (n = 10), 15 months (n = 10), 24 months (n = 10) and 30 months (n = 12) throughout the paper (Supplementary Table 1.1). In our study, we focused exclusively on male mice for two primary reasons. First, we aimed to minimize potential confounding factors arising from fluctuations in sex hormones in female mice, which are known to influence metabolic processes across tissues during aging[80]. Second, addressing sex differences in aging would have required a fully stratified experimental design[3] and, consequently, a separate cohort of female mice. Given that only 10–15% of animals typically reach the age of 30 months, achieving comparable sample sizes and statistical power for the oldest age group alone would have necessitated approximately 100 female mice.

**Metabolomics cohort.** An independent mouse cohort was used for the metabolomics analysis of faecal samples. This cohort comprised 83 male mice in five age groups: 3 months (n = 16), 9 months (n = 16), 15 months (n = 16), 24 months (n = 17) and 28 months (n = 18) (Supplementary Table 4.11). These mice were bred and housed in the same mouse facility under the same conditions.

**GF mice cohort.** For the analysis of host responses to microbial colonization, tissues of female C57BL/6J mice (n = 24) were obtained from the Experimental Biomedicine facility at the University of Gothenburg,

Sweden. Throughout the experiment, the mice had ad libitum access to chow and water and were exposed to a 12:12 h light–dark cycle. The mice were divided into three treatment groups: GF (n = 8), conventionally raised (CONVR, n = 8) and conventionalized (CONVD, n = 8). The mice within each group were not all littermates. A CONVR mouse, which was not part of the sampled CONVR group and was ~10 weeks old, served as the donor for the conventionalization process. GF mice were orally gavaged with gut microbiome at 10 weeks of age on average (Supplementary Table 1.1). The gut microbiome used for conventionalization was extracted from the caecum and mixed with reduced phosphate-buffered saline to obtain a final volume of 200 µl.

### Sample collection

**Main study and metabolomics cohort.** The mice were sacrificed by cervical dislocation in three cohorts (randomized by age) on three consecutive mornings. The left hemisphere of the brain was prepped on ice, transferred to liquid nitrogen for storage and used later for RNA extraction. Faeces were collected from the colon by squeezing the colon contents towards the distal end and snap-freezing one pellet in liquid nitrogen; the pellets were used later for metagenomic sequencing (for the first cohort) or metabolite measurement by hydrophilic interaction liquid chromatography ultrahigh-performance liquid chromatography–tandem mass spectrometry (for the second cohort). The colons were rinsed with sterile phosphate-buffered saline and cut longitudinally; a piece measuring the length of one-eighth of the left half of the mid colon was frozen in liquid nitrogen for later use in RNA extraction. A piece with a length of approximately 1 cm was cut from the end of the left lateral lobe of the liver and snap-frozen in liquid nitrogen for later use in RNA extraction. RNA was extracted from tissue samples of the liver, colon and left brain hemisphere using the phenol–chloroform extraction method with 1 ml of Qiazol Lysis Reagent (Qiagen)[81].

All studies were performed in strict compliance with the recommendations of the European Commission for the protection of animals used for scientific purposes and with the approval of the local government (Thüringer Landesamt für Verbraucherschutz, Germany; license: 02-024/15; TWZ-000-2017). Experiments were performed according to the ARRIVE guidelines[82].

**GF mouse cohort.** At ~12 weeks of age, all mice were sacrificed for the extraction of brain, colon, liver, gonadal white adipose tissue (gWAT) and quadriceps tissues. RNA was isolated from the brain, colon, liver and quadriceps via the 'RNeasy mini kit' (Qiagen) according to the manufacturer's protocol, while RNA from gonadal white adipose tissue was isolated using the TRIZOL method[83]. Briefly, 1 ml TRIzol was added to 50–75 mg pestle-homogenized tissue followed by vortexing, a 5-min incubation at room temperature and addition of 200 µl chloroform. After mixing, further incubation at room temperature for 2–3 min and centrifugation (12,000 g) at 4 °C for 5 min, the clear supernatant was mixed with 500 µl isopropanol followed by incubation at room temperature for 10 min. After further centrifugation (12,000 g) at 4 °C for 10 min, the supernatant was discarded and the pellet washed with 1 ml cold 75% EtOH followed by vortexing and centrifugation (7,500 g, 4 °C, 5 min). The pellet was dried and dissolved in RNase-free water. All animal protocols were approved by the Gothenburg Animal Ethics Committee (vote #2652-19).

### Metagenomic sequencing

Microbial DNA was extracted from colon contents with the DNeasy PowerSoil Kit (Qiagen) following the manufacturer's protocol. Next, the DNA was prepared at the Max Planck Institute for Evolutionary Biology (Plön, Germany) with the Illumina NexteraXT Library Kit. All 52 samples were pooled and sequenced for 2 × 150 cycles in paired-end mode on all four lanes of an Illumina NextSeq 500 machine. Demultiplexing was performed with one mismatch allowed in barcodes. The raw read data were merged sample-wise and subjected to quality control for

adaptor contamination and base call qualities. Adaptor sequences with an overlap of ≥3 bp and base calls with a Phred+33 quality score of <30 were trimmed from the 3′ ends of reads using Cutadapt (version 1.12). Illumina's Nextera transposon sequence and the reverse complement of TruSeq primer sequences were used as adaptor sequences.

Subsequently, reads were subjected to quality control using Prinseq lite (version 0.20.4) with a sliding window approach that applied a step size of 5 bp, a window size of 10 bp, a mean base quality of <30 and a minimum-length filter that discarded any reads shorter than 50 bp after all other quality control steps. To filter out host sequences, the remaining sequences were mapped to the mouse reference genome (GRCm38.99) with Bowtie (version 2.2.5). The remaining unmapped reads were then used for MAG assembly.

No significant differences were detected in the total microbial read depth or host contamination between age groups (Kruskal–Wallis test with post hoc Dunn's test and Benjamini–Hochberg multiple-testing correction conducted with the DunnTest function in the DescTools R package (version 0.99.50); Extended Data Fig. 1b,c).

Long-read sequencing was performed at the next-generation sequencing (NGS) core facility of the FLI Leibniz Institute on Aging (Jena, Germany). The DNA quality was assessed with an Agilent Bioanalyzer 2100 with a DNA 12000 Kit (Agilent Technologies) and quantified with an Invitrogen Quant-iT PicoGreen dsDNA Assay (Thermo Fisher Scientific). The sequencing library was prepared according to the Pacific Biosystems' manual 'Procedure & Checklist - 20 kb Template Preparation Using Blue-Pippin Size-Selection System' (version 10, January 2018) with the SMRT-bell Template Prep Kit 1.0 (Pacific Biosciences). Specifically, DNA from age-matched samples was pooled, fragmented (75 kb) by a Megaruptor (Diagenode) and size selected for >6-kbp fragments with a BluePippin and 0.75% Gel Cassette (programme: 0.75% DF Marker S1 High-Pass 6–10 kb vs3; Sage Science). Each pool was loaded onto a SMRTcell and sequenced on a Pacific Biosystems RSII machine with DNA-Sequencing Kit 4.0 v2, MagBeadBuffer Kit v2, MagBead Binding Buffer Kit v2 and DNA Polymerase Kit P6v2. The sequence output of these eight runs had an average read length of 7.8–9.7 kb with a minimum yield of 750 kbp per SMRTcell. The raw read data were subjected to quality control, processed into circular consensus sequences and subreads, and exported as FASTQ files via the SMRTportal (provided by Pacific Biosciences).

## MAG assembly and annotation

MAGs were constructed as follows (outlined in Extended Data Fig. 1a). Pacific Biosystems circular consensus sequences and subreads were used as is, while Illumina shotgun reads were filtered for low read quality, adaptors and host contamination (Metagenomic Sequencing). A full cohort assembly was done in metaSPAdes (SPAdes version 3.13.1) in hybrid mode with $k$-mer sizes of 21, 33, 55 and 77. Concatenated, quality-controlled, forward and reverse Illumina short read files of all samples were used as input. In addition, the assembly software was informed with the eight Pacific Biosystems long read banks (hybrid mode) in the form of filtered subreads and circular consensus sequences.

The resulting scaffolds were filtered for a minimum length of 1,000 bp and coverage ≥7.7815. The cut-offs were determined by scatter plotting coverage versus length, as described in ref. 84. The quality-controlled metagenomic reads were mapped back to the filtered scaffolds with Bowtie (version 2.2.5); the insert size was 0–1,000 bp in the very sensitive, non-deterministic, 'fr' stranded mode with end-to-end alignment. Non-unique mappings and unaligned reads were discarded. The scaffold coverage depth was determined with the jgi_summarize_bam_contig_depths script from MetaBAT (version 2.12.1). This coverage depth information was then used to sort the remaining scaffolds into bins, each representing single bacterial genomes, with the binning tools MetaBAT (version 2.12.1), CONCOCT (version 1.1.0) and MaxBin (version 2.2.4). For CONCOCT, the scaffolds were broken up into 10-kbp chunks. Bin refinement was conducted with the combined results of all three binners (252 bins) with DASTool (version 1.1.2); subsequently, quality metrics were calculated by CheckM (version 1.1.2). Bins with a quality estimate of >80% and a contamination estimate of <10% were considered for further analysis and are henceforth referred to as MAGs. In our reporting of medium- and high-quality MAG drafts, we referred to the standards and metrics laid out by The Genome Standards Consortium[23]. Accordingly, a MAG will be considered high quality with a completion >90%, contamination <5% and whether genes for 23S, 16S and 5S rRNA and at least 18 tRNAs are recovered. While 133 of our 181 MAGs fulfil these very strict completion and contamination cut-offs, only 25 of those 133 could be considered true high-quality MAGs only due to some missing rRNA or tRNA genes. The lack of those genes, however, does not impact the quality of our MAG-derived metabolic models. Only 18 of our MAGs showed a contamination score greater than 5%, ranging from 5.1% to 9.8% with a mean of 6.7%, and only 2 of those 18 had a completeness score <90% (Supplementary Table 1.2). We used slightly less strict cut-offs for contamination and completeness to include a larger variety of MAGs in our study. While the metabolic model construction from MAGs can partially compensate for lack of completeness via gap filling and for contamination by pathway-completeness checks, our more loose contamination cut-offs might reduce the accuracy of taxonomic assignments.

The 181 final MAGs were taxonomically annotated with GTDB-Tk (version 2.1.1) and database version r214. The tRNA genes were characterized using tRNAscan-SE (version 2.0.9). The 16S rRNA genes were detected by barrnap (version 0.9) in the 'kingdom bacteria' mode. A phylogenetic tree of the 181 MAGs (Fig. 1a) was created from a multiple sequence alignment created by GTDB-Tk (align/gtdbtk.bac120.user_msa.fasta.gz) with the European Bioinformatics Institute's online Simple Phylogeny tool (ClustalW version 2.1) and visualized with R statistical software. The complete characterization of the MAGs is provided in Supplementary Table 1.2.

For association with age, MAG abundances were calculated via the mean of the scaffold coverage depths across all scaffolds belonging to a MAG, normalized by total sample abundance and then correlated with age in linear models for each MAG across all samples. The $P$ values were corrected for multiple testing with the Benjamini–Hochberg FDR method. Significant age-associated MAGs (FDR-adjusted $P$ value ≤ 0.05) were plotted (Fig. 4a).

## Microbiome metabolic model construction

Metabolic models were constructed for each of the 181 mouse gut bacteria inferred from our MAGs in samples from the 52 mice. The reconstruction was performed in gapseq (version 1.2) with default settings including gap filling of models (git commit: 159ad378; sequence DB md5sum: bf8ba98)[24]. Gap filled reactions for each model are indicated in the reconstructed models in the Zenodo archive. The nutritional input for the computational models was designed according to the fortified rat and mouse diet (V1534-300; ssniff Spezialdiäten). The diet was reconstructed according to the vendor's information on its molecular constituents translated into the corresponding metabolites in the models, following the protocol described in ref. 85. We assumed an average daily uptake of 3.5 g of food based on reference values[86]. This amount was used to transform the percentages into grams and then millimoles (millimoles per day). Limited information was reported on fibre in the mouse diet; therefore, their values were imputed from the consumed quantities of cereal and grain products of a German human cohort[17]. Because the simulations depicted the intestinal setting, the absorption in the small intestine was considered when calculating the dietary input (see Supplementary Tables 1.3–1.11 for the respective calculations and references).

## Growth rate prediction from metagenomic data

To further validate the model growth rates, CoPTR[87] was used to estimate growth rates from the MAGs in each sample. This method uses the

peak-to-trough ratio (PTR) (that is, the ratio of sequencing coverage near the replication origin and the replication terminus) to estimate the growth of a MAG in a sample[88]. We first indexed the MAGs with the command 'coptr index –bt2-threads'. Next, using this index, we mapped our quality-controlled metagenomic reads against our 181 MAGs with the command 'coptr map –threads 4 –paired'. Then, read positions were extracted with the command 'coptr extract', and the PTR was estimated with the command 'coptr estimate'. Default parameters were used for all commands except 'coptr index' and 'coptr map' for which the number of threads was specified. In addition, '–paired' was set for the 'coptr map' command to inform the software about the use of paired-end reads. Community growth was determined for each mouse's microbiome community by calculating the median growth rate across all MAGs in its sample. We did not weight growth rate predictions by individual species' abundances to obtain a community-level growth rate. Thereby, we avoided spurious correlations with community growth rates predicted using community FBA as community FBA explicitly incorporates abundance information. Please note that while a previous study found little correlation between PTR estimates and experimentally measured growth rates[89], this study did not include CoPTR in the benchmark and CoPTR itself was explicitly validated on MAGs.

### Modelling of ecological relationships within the microbiome

The ecological relationships for each pair of bacteria across all species were predicted. To this end, the growth achieved by a single bacterium was compared with that achieved when each bacterium was co-grown with other bacteria. The relationships were characterized using the ecological relationships described in a previous study (Fig. 1 in ref. 90) as a reference. Growth was estimated by FBA for single growth and community FBA for combined growth. To achieve this, we used the R packages sybil[91] and MicrobiomeGS2 (www.github.com/Waschina/MicrobiomeGS2) and the linear programming solver IBM ILOG CPLEX 22.10. The six types of ecological relationships and their frequencies among each microbial community were inferred with the R EcoGS package (https://github.com/maringos/EcoGS). To this end, we considered for each microbiome each potential pair of species. The type of ecological interaction between the pair was determined by comparing individual growth rates with growth rates when both species were combined. Summing pairwise frequencies for each type of inferred interaction, we then obtained the frequency of an interaction in each community. To obtain relative frequencies, the abundance of ecological relations was normalized sample-wise to a sum of 1. Next, a linear model analysis of each ecological interaction type with age was conducted and $P$ values were adjusted for multiple testing using the Benjamini–Hochberg FDR method.

### Host–microbiome partial correlations

The transcriptomic data were normalized separately for each organ (colon, liver and brain) using variance-stabilizing transformation informed with age and sequencing batch (blind = FALSE) implemented in the R package DESeq2 (version 1.40.2)[92]. A near-zero variance filter was also applied using the nearZeroVar function of the R package caret (version 6.0-94). The active reactions of each mouse's microbiome community were predicted as described in 'Estimation of Functional Capacity of Microbiomes'. The host transcript abundances were correlated pairwise with microbiome active reactions (each transcript with each reaction), correcting for age and sequencing batch (only for the liver and brain), with Spearman's partial correlations (implemented in the R package ppcor (version 1.0)[93]). To balance stringent false discovery cut-offs with reasonable result counts, strong correlations with a Benjamini–Hochberg FDR-corrected[94] $P \leq 0.1$ and Spearman's $\rho \geq 0.55$ were considered for downstream analysis. Correlated feature pairs were obtained for the colon ($n = 12,732$), liver ($n = 3,425$) and brain ($n = 2,499$). They consisted of n unique features for the colon (microbiome, $n = 1,606$; host, $n = 2,815$), liver (microbiome, $n = 1,359$; host,

$n = 1,277$) and brain (microbiome, $n = 1,236$; host, $n = 926$). The strong correlations were stratified into either positive or negative correlations according to their correlation values and then annotated with GO biological processes[25] (host transcripts) or MetaCyc Pathways[26] (microbiome reactions) using hypergeometric over-representation tests with the phyper function of the R stats package (version 4.3.2; $x$ = 'correlated features enriched for the term' − 1, $m$ = 'the total of all correlated features,' $n$ = 'all features' − 'correlated features' and $k$ = 'the total of the features in the term'). Enriched terms (pathways and processes) with at least three features and an FDR-corrected over-representation $P \leq 0.05$ were reported (Supplementary Tables 2.1–2.3 and Fig. 2). After enrichment, we obtained $n$ process pairs for the colon ($n = 1,377$), liver ($n = 283$) and brain ($n = 167$), as shown in Fig. 2 and Supplementary Tables 2.1–2.3. The negative decadic logarithm of the over-representation FDR $P$ values was calculated and reported as is for positive correlations and multiplied by −1 for negative correlations. Only process pairs that were associated with at least two other pathways were plotted (Fig. 2) and filtered to highlight the most significant enrichments with FDR $P$ value cut-offs of $\leq 1 \times 10^{-10}$ for the colon (Fig. 2a), $\leq 1 \times 10^{-4}$ for the liver (Fig. 2b) and $\leq 1 \times 10^{-3}$ for the brain (Fig. 2c).

For a broader overview of host–microbiome associations, the GO biological processes were grouped by their higher-ranking level 2 GO biological process, and the MetaCyc pathways were grouped by their respective highest-level superpathways (see Supplementary Tables 2.5 and 2.6 for the process and pathway groups). The level 2 GO biological process groups were cellular process, metabolic process, biological regulation, localization, developmental process, response to stimulus, immune system process, multicellular organismal process, viral process, reproduction, homeostatic process and growth. The MetaCyc microbial superpathways were lipids, carbohydrates, utilization, energy metabolism, nucleotides, secondary metabolites, amino acids, other, signalling, carboxylates, cofactors, carriers, metabolic regulators, c1 compounds, electron transfer, noncarbon nutrients, cell structure, biosynthesis, detoxification, interconversion, glycans, tRNA and bioluminescence. The $-\log_{10}$(FDR-corrected $P$ values) were summed for each level 2 GO and MetaCyc superpathway pair. The values are plotted in Extended Data Fig. 2a and listed in Supplementary Table 2.4.

The pairwise correlations between all host features and all microbiome features were repeated, stratified by organ and age group. Thus, the ratio of significant host–microbiome correlations to all tested pairs was obtained for each organ and age group. These ratios were compared using Pearson's chi-squared test with Yates' continuity correction and Bonferroni's multiple testing correction to identify significant differences between consecutive age groups (Extended Data Fig. 6a–e).

The overlaps of aging-regulated and microbiome-associated transcripts between the three studied organs were determined to identify shared aging-regulated and microbiome-regulated host transcripts. The overlap between microbiome-associated and aging-associated transcripts was statistically evaluated using hypergeometric over-representation tests separately for each organ. The numbers of shared transcripts were plotted (Fig. 6a) for each possible combination for the colon (age associated, $n = 4,715$; microbiome associated, $n = 2,815$; shared, $n = 589$; hypergeometric $P = 4.9 \times 10^{-15}$), liver (age associated, $n = 8,285$; microbiome associated, $n = 1,277$; shared, $n = 485$; hypergeometric $P = 1.4 \times 10^{-6}$) and brain (age associated, $n = 6,505$; microbiome associated, $n = 926$; shared, $n = 236$; hypergeometric $P = 3.7 \times 10^{-5}$). Reported $P$ values from hypergeometric tests were corrected for multiple testing via Benjamini and Hochberg's method.

### Validation of microbiome-associated host genes

Differentially expressed genes and transcripts were derived from the GF validation cohort as described in the Supplementary Methods section 'Differential Gene Expression Analysis'. We identified the overlaps

of microbiome-associated transcripts, identified from our main study cohort via partial correlations (see Methods section 'Host–Microbiome Partial Correlations'), with the differentially expressed transcripts from our GF validation cohort in both treatment contrasts (GF versus CONVR and GF versus CONVD). The number of shared microbiome-associated transcripts between both cohorts was statistically evaluated using hypergeometric over-representation tests stratified by organ and treatment contrast. The numbers of shared transcripts were plotted (Fig. 2d) for each possible combination, for the colon (main cohort, $n = 2{,}815$; GF cohort versus CONVR, $n = 2{,}703$; shared, $n = 438$; hypergeometric $P = 1.1 \times 10^{-32}$ and GF cohort versus CONVD, $n = 7{,}637$; shared, $n = 1{,}207$; hypergeometric $P = 7.2 \times 10^{-102}$), for the liver (main cohort, $n = 1{,}277$; GF cohort versus CONVR, $n = 3{,}281$; shared, $n = 223$; hypergeometric $P = 2.7 \times 10^{-7}$ and GF cohort versus CONVD, $n = 351$; shared, $n = 24$; hypergeometric $P = 7.9 \times 10^{-2}$) and for the brain (main cohort, $n = 926$; GF cohort versus CONVR, $n = 0$; and GF cohort versus CONVD, $n = 554$; shared, $n = 20$; hypergeometric $P = 1.7 \times 10^{-1}$). Reported $P$ values from hypergeometric tests were corrected for multiple testing via Benjamini and Hochberg's method.

## Reconstruction of the generic metamodel

A two-step procedure was followed to obtain a metamodel for each mouse. In the first step, a generic metamodel representing the individual organs and the microbiome was assembled. In the second step, a specific metamodel of each mouse was derived by integrating expression and metagenomic data.

In the first step, we joined three times the human metabolic reconstruction Recon 2.2 (ref. 28) representing the individual organs with a microbiome metabolic model according to their physiological interactions (Fig. 3a). A mouse-specific metabolic reconstruction was not used, as the human reconstructions are by far the best curated and there is a high overlap in metabolic content between mice and humans[95]. All compartments interfaced with each other via common exchange environments, such as the gut lumen (microbiome and colon) and the bloodstream (colon, brain and liver). Some exchanges along the bloodstream were defined as directional, following the physiological interactions of the organs (see Fig. 3a). Metabolite uptake into the brain was restricted to metabolites known to cross the blood–brain barrier (see Supplementary Table 3.7 for a list). To compile this list of compounds, literature resources[46,96,97] were used; in addition, we selected the compounds in Recon 2.2 (ref. 28), along with those identified on the Virtual Metabolic Human website (www.vmh.life) whose physicochemical properties would allow them to cross the blood–brain barrier[98]. For the microbiome metabolic model, all individually reconstructed MAGs were merged into a single model by combining all microbial reactions of the individual bacterial cellular compartments into a single reaction space. This merged microbiome model could then interact with the human metabolic models via the lumen exchange environment. We decided to use a merged microbiome model instead of species-level metabolic reconstructions to maintain computational tractability of the metamodel for comprehensive downstream analysis (for example, flux variability analysis and EFM sampling).

To better account for organ- and microbiome-specific uptake and secretion of metabolites, exchange reactions of the individual compartments were split into irreversible forwards and backwards directions. To model the dietary uptake of the mice, the molar concentrations of all metabolites in their diet were derived and represented in the model following an established protocol[85]. In addition, information on the absorption of dietary metabolites before entry into the colon was obtained to differentiate between ileal and colonic uptake. The diet was integrated into the model by a direct inflow of absorbed compounds to the bloodstream and unabsorbed compounds to the colonic lumen for microbiome and colonic use.

Following the merging of the human and bacterial metabolic models, several energy-generating cycles (that is, sets of metabolic reactions that can form ATP from ADP without the consumption of other metabolites) were identified and resolved by correcting potential problems in the reversibility of participating reactions (Supplementary Table 3.6). The metamodel can be found under accession MODEL2310020001 in the EBI BioModels database (https://www.ebi.ac.uk/biomodels/)[99].

## Reconstruction of mouse context-specific metamodels

In the second step, we built a context-specific metamodel for each mouse. To achieve this, StanDep[100] was applied to the transcriptomic and metagenomic data to derive the core reactions for each tissue required to reconstruct context-specific models using fastcore[29]. Transcriptomic data were preprocessed by transforming counts into fragments per kilobase of transcript per million mapped reads (FPKMs). After removing genes with at least one sample with zero detected expression or a mean FPKM < 0.1 and $\log_2$ transformation, FPKM values were normalized using Combat[101] and then transformed back to their original scale. To identify core reactions, mouse genes were mapped to their corresponding human orthologues using Ensembl Biomart[102]. The gene expression data of all tissues were combined into a single matrix, and tissue and age groups were used as separating categories for StanDep. StanDep was applied with 'chi2dist' as the distance method and 'complete' as the linkage method. After screening optimal cluster numbers, predicted core reactions remained stable when using 39 clusters (that is, the Jaccard distance of derived core reactions for StanDep runs with increasing cluster numbers was below 0.05 using at least 39 clusters).

For metagenomic data, we obtained a reaction abundance matrix for each microbiome sample. To this end, reads were mapped to MAGs to derive species-level counts. These were then multiplied with a reaction contribution matrix indicating for each reaction in which species they are present (normalized to a sum of one for each species) and normalizing to a sum of one across all reactions in a sample. Reaction abundances were used as input to StanDep with the age group of the sample as the separating factor, 'chi2dist' as the distance method and 'complete' as the linkage method. Following the same procedure as for the gene activity data, 15 clusters were identified as optimal for reaction abundance data.

In addition, metabolic exchanges between individual organs and the bloodstream previously measured in pigs were included[103] by mapping IDs of exchanged metabolites to the corresponding metabolite identifiers in Recon 2.2. If an organ took up or secreted a metabolite, the corresponding uptake or secretion reactions were added to the core reactions. If the kidney took up a metabolite, the corresponding outflow reaction from the blood was added to the core reactions because the kidney was not modelled explicitly. Subsequently, the core reactions for each sample and the generic metamodel were used as input for fastcore to derive a context-specific metamodel for each mouse. To run fastcore, CORPSE (https://github.com/Porthmeus/CORPSE) was used as an interface to the corresponding functions of the TROPPO toolbox[104].

## Host and microbiome dependence of reactions

To determine microbiome- or host-dependent functions, flux variability analysis with and without the microbiota were conducted. FVA was performed[105] by maximizing and minimizing flux through each reaction using the 'flux_variability_analysis()' function of CobraPy[106] without optimization of growth (fraction_of_optimum = 0) as for most tissues in mammals there is only negligible cellular replication[107]. Because internal exchange reactions were split into irreversible forwards and backwards steps, they were treated separately by always blocking the corresponding opposing direction. FVA results were summarized by determining admissible flux ranges, by subtracting minimal from maximal flux. Microbiome-dependent exchange reactions in the host were identified by repeating the FVA but blocking each microbiome

reaction. An exchange reaction was deemed microbiome dependent if its flux range was reduced to less than 10% when blocking microbiome reactions. To elucidate the metabolites exchanged between the host and the microbiota, the microbiota-dependent uptake and secretion reactions of metabolites for a given organ were counted. If the number of cases of microbiome-dependent secretion subtracted from the frequency of microbiome-dependent uptake was larger than 10, a metabolite was classified as being provided by microbiota to the host or vice versa (20% of samples). For plotting (Fig. 3b), only metabolites with a difference of at least 35 for colon and a difference of at least 12 for brain and liver were shown.

### Identification of reaction-level host–microbiome dependencies

To determine the dependencies of individual host reactions on individual microbial reactions, EFMSampler[32] was used to sample EFMs with each host and microbial reaction as indicator reaction for sampling. The indicator reaction is used to define the specific reaction in a model through which EFMs should be determined. For each target reaction, EFMSampler was run in eight parallel threads using flux minimization as objective until either 10,000 EFMs were sampled or >200 s had elapsed and the average frequency of occurrence of reactions in EFMs was recorded. Subsequently, occurrence frequencies were averaged across all 52 mice. This yielded a matrix in which each column corresponded to a target reaction and each row indicated the frequency at which all other reactions occurred in EFMs containing that reaction. Thus, a value of '1' indicates microbiome reactions that always co-occur in EFMs of the target reaction and a value of '0' no co-occurrence

To compare EFM-predicted interactions to host–microbiome correlations, scores in the interaction matrix were compared between genes and microbiome reactions with significant associations. To this end, for each significant host gene–microbiome reaction association (FDR-adjusted $P \le 0.1$), the maximum interaction score in the sub-matrix containing reactions associated with the host gene and the microbiome reaction were determined and collected across all significant host gene–microbiome reaction associations in a tissue to derive a set of 'true' maximal interaction scores. The same analysis was performed 100 times for randomly drawn genes associated with reactions present in the tissue and randomly selected microbiome reactions to obtain 'random' maximal interaction scores. Then, true and randomly generated maximal interaction scores were compared using the Wilcoxon rank-sum test.

To analyse the most strongly interacting metabolic processes between the host and microbiome, an interaction was assumed if the microbiome reaction occurred in at least 50% of the EFMs sampled from that host reaction across all metamodels. Then, for each host–microbiome reaction pair, we determined which metabolic subsystems they were associated with and counted each corresponding host–microbiome subsystem pair across all such pairs in the interaction matrix. The enrichment of pairs was then tested using Fisher's exact test comparing for each pair the number of mutual interactions of reactions belonging to the host and microbiome subsystems to the frequency of interactions across the entire interaction matrix. An enrichment was assumed with an FDR-corrected $P \le 0.05$, calculated using the p.adjust function in R.

### Identification of aging-regulated metabolic modules

To identify aging-regulated metabolic modules, we defined sets of reactions associated with each indicator reaction used for EFM sampling. A reaction was assumed to belong to the metabolic modules of an indicator reaction if it occurred in at least 20% of the EFMs sampled for that indicator reaction. Unlike in the analysis of reaction-level dependencies between host and microbiota, we considered both the host and microbiome components of the EFMs; thus, metabolic modules contained both host and microbiome reactions. A metabolic

module was considered dependent on the microbiome if it contained at least 20 microbial reactions. To identify aging-regulated metabolic modules, aging-induced and aging-repressed genes (Supplementary Tables 5.1–5.3) were translated into the reactions with which they were associated in the metabolic model. Then, for each metabolic module, we tested whether the corresponding set of reactions was enriched for aging-induced or aging-repressed metabolic reactions using Fisher's exact test, assuming the entire set of reactions occurring in a tissue as background. Subsequently, we determined host pathways in which indicator reactions of aging-regulated modules dependent on the microbiota were enriched. To this end, we filtered microbiome-dependent aging-regulated modules with an enrichment of aging-regulated reactions with $P \le 0.01$. For each module, we determined the corresponding indicator reactions and performed a pathway enrichment based on the subsystem annotation of the Recon 2.2 model using Fisher's exact test. For transport reactions, we also added a subsystem annotation for the transport of nucleotides (encompassing deoxyribonucleic and ribonucleic acids) and amino acids. All reactions of a tissue occurring in at least one metamodel were used as background or universe in the Fisher's exact test of that tissue.

## Materials availability

This study did not generate new unique reagents.

### Reporting summary

Further information on research design is available in the Nature Portfolio Reporting Summary linked to this article.

## Data availability

Metagenomic raw read and MAG assembly data were deposited in the European Nucleotide Archive (ENA) under BioProject PRJEB73981 (ebi.ac.uk/ena/browser/view/PRJEB73981). Individual accession numbers for each MAG are listed in Supplementary Table 1.2. Gene expression data were published in the GEO database under record GSE262290 (ncbi.nlm.nih.gov/geo/query/acc.cgi?acc=GSE262290) and record GSE278548 (ncbi.nlm.nih.gov/geo/query/acc.cgi?acc=GSE278548). Metabolomics data have been made available at the MassIVE database (massive.ucsd.edu) with identifiers MSV000094409, MSV000094410 and MSV000094436. The metamodel can be found under accession MODEL2310020001 in the BioModels database (ebi.ac.uk/biomodels/MODEL2310020001)[99]. Detailed sample metadata, the microbial metabolic models and supplementary resources are available via Zenodo at https://doi.org/10.5281/zenodo.10844502 (ref. 108).

## Code availability

The source code used for data analysis is available via GitHub at github.com/sciwitch/MouseMicrobiomeAging.

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

## Acknowledgements

We acknowledge funding by the German Research Foundation to C.K. within the scope of CRC1182 (project A1.5), the Research Group miTarget (FOR5042) and the Cluster of Excellence 'Precision Medicine in Chronic Inflammation' (EXC2167). We also acknowledge funding from the German Research Foundation (DFG, project number: 416 418087534) to C.K. and C.F., project SO1141/10-1 to F.S., intramural grants of the medical faculty of Kiel University (grants nos K126408 and K126493) to F.S., the Carl Zeiss Foundation IMPULS programme (project number: P2019-01-006) to C.F. and the European Union's Horizon 2020 Research and Innovation Programme (under the Marie Sklodowska-Curie grant: agreement number 859890 (SmartAge)) to O.W.W., C.K. and C.F. P.R. was supported by BMBF iTREAT, DFG SFB1182 (C2), RU miTARGET (P3) and EXC 2167 RTFVI and TI1. This work was supported by the DFG Research Infrastructure NGS_CC (project number: 407495230) as part of the Next Generation Sequencing Competence Network (project number: 423957469). Short-read sequencing was conducted at the Competence Centre for Genomic Analysis (Kiel, Germany). We thank K. Cloppenborg-Schmidt for the excellent technical assistance in preparing the samples for metagenomic sequencing. We thank L. Mannerås Holm and F. Bäckhed (University of Gothenburg) for providing tissue samples from wild-type, GF and conventionalized C57BL/6J mice.

## Author contributions

Conceptualization: C.F., C.K., D.E., L.B., M.H., O.W.W. and S.S. Methodology: A.S.K., A.W., C.F., C.K., D.E., G.M., J.F.B., J.Z., L.B., M.G., M.H., F.S., P.S.-K., R.H., R.S., S. Flor, A.M.G., S.K., S.W. and T.D. Software: A.S.K., C.K., G.M., J.T., J.Z., L.B., S. Flor, S.W. and T.D. Investigation: A.W., C.F., C.K., D.E., L.B., M.G., M.H., P.S.-K., R.H., R.S., S. Franzenburg, F.S. and S.K. Formal analysis: A.S.K., A.W., C.K., G.M., J.Z., L.B., S. Flor, A.M.G., S.W. and T.D. Writing, original draft: A.S.K., A.W., C.K., G.M., J.Z., L.B., S. Flor and T.D. Writing, review and editing: C.F., J.F.B., J.T and F.S. Visualization: C.K. and L.B. Supervision: C.F., C.K., O.W.W., P.S.-K. and S.W. Funding acquisition: C.F., C.K., J.F.B., P.R., F.S. and O.W.W. Project administration: C.F., C.K. and L.B. The order of those authors who contributed equally was determined alphabetically. To accurately represent their equal contributions in any type of professional or academic documentation, they are permitted to rearrange the order of their names among the shared equally contributed positions at their discretion.

## Funding

## Competing interests

The authors declare no competing interests.

## Additional information

**Extended data** is available for this paper at https://doi.org/10.1038/s41564-025-01959-z.

**Correspondence and requests for materials** should be addressed to Christoph Kaleta.

[1]Research Group Medical Systems Biology, Institute of Experimental Medicine, Kiel University and University Hospital Schleswig-Holstein, Kiel, Germany. [2]Institute of Clinical Chemistry, University Hospital Schleswig-Holstein, Kiel/Lübeck, Germany. [3]Department of Neurology, Jena University Hospital, Jena, Germany. [4]CAU Innovation GmbH, Kiel University, Kiel, Germany. [5]Research Unit Analytical BioGeoChemistry, Helmholtz Munich, Neuherberg, Germany. [6]Evolutionary Ecology and Genetics, Zoological Institute, Kiel University, Kiel, Germany. [7]Antibiotic resistance group, Max Planck Institute for Evolutionary Biology, Plön, Germany. [8]Max Planck Institute for Evolutionary Biology, Plön, Germany. [9]Institute of Clinical Molecular Biology, Kiel University and University Hospital Schleswig-Holstein, Kiel, Germany. [10]Core Facility Next-Generation Sequencing, Leibniz Institute on Aging—Fritz Lipmann Institute, Jena, Germany. [11]Nutriinformatics, Institute of Human Nutrition and Food Science, Kiel University, Kiel, Germany. [12]Institute of Analytical Food Chemistry, Technical University München, Freising, Germany. [13]Section of Evolutionary Medicine, Institute of Experimental Medicine, Kiel University, Kiel, Germany. [14]Present address: Department of Dermatology and Allergy, University Hospital Schleswig-Holstein, Kiel, Germany. [15]These authors contributed equally: Thomas Dost, Daniela Esser, Stefano Flor, Andy Mercado Gamarra, Madlen Haase, A. Samer Kadibalban, Georgios Marinos, Alesia Walker, Johannes Zimmermann. [16]These authors jointly supervised this work: Christiane Frahm, Christoph Kaleta. ✉e-mail: c.kaleta@iem.uni-kiel.de

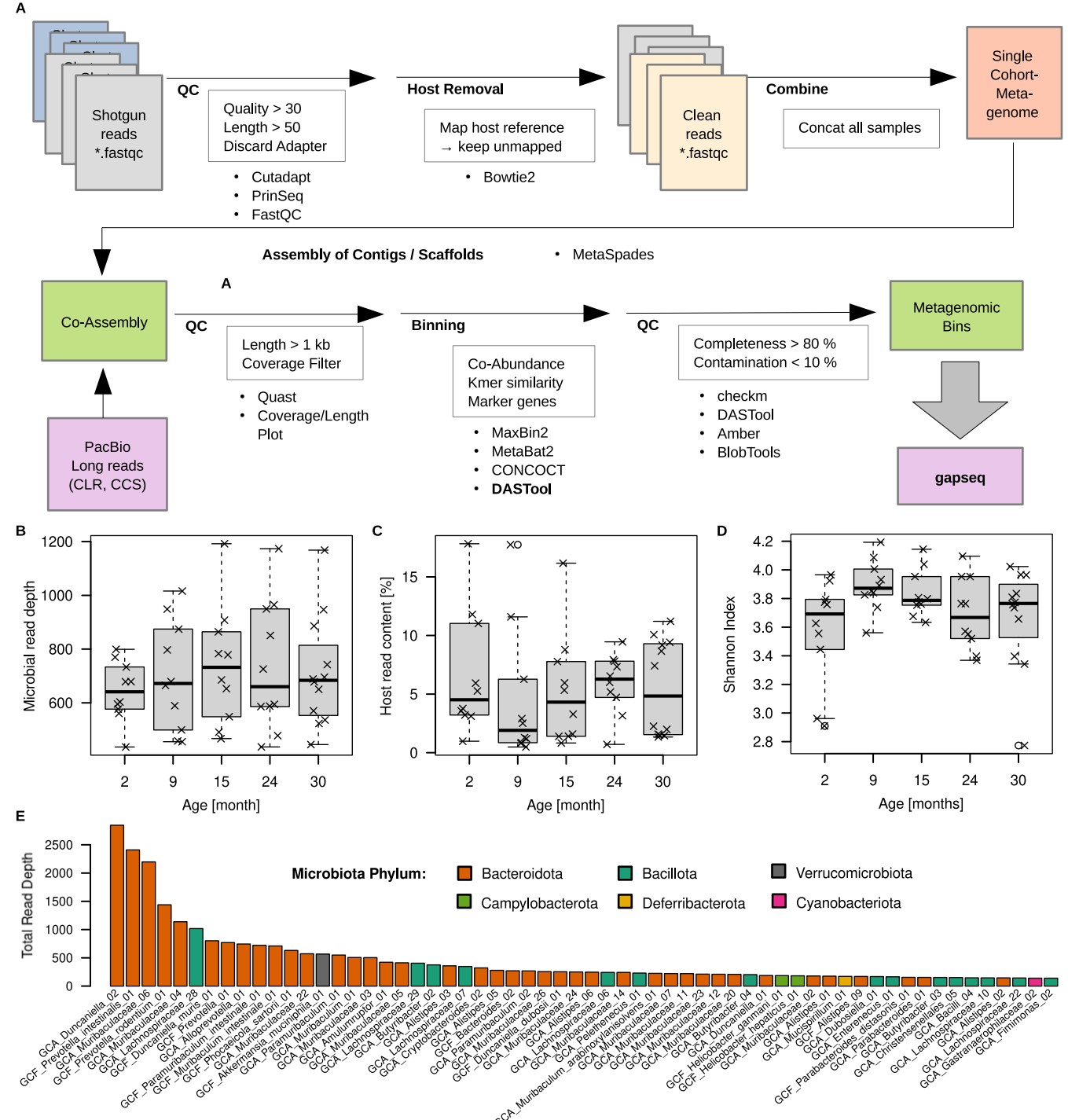

**Extended Data Fig. 1 | Metagenomic processing steps and results.**
**a** Metagenomics workflow. **b**, **c** Abundance of metagenomics reads derived
from **b** microbes or **c** mouse DNA. **d** Alpha diversity derived from metagenomic
reads mapped against MAGs. **b-d** Dunn's test FDR, all not significant; sample size:
30mo.: n = 12, all others n = 10. **e** Abundance profile of the 60 most abundant
MAGs in the full cohort. Bacteroidota predominated in the cohort. Boxplot
elements: center line, median; box limits, 25%–75% quantiles; whiskers, 1.5x IQR;
points, outliers.

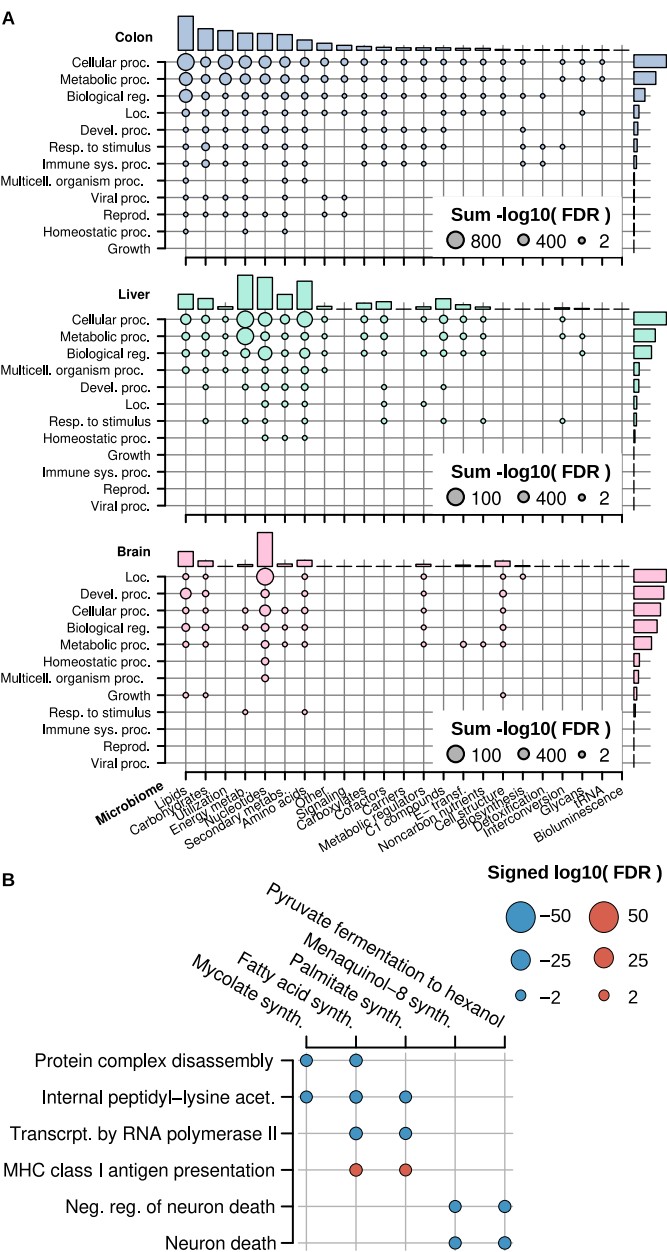

**Extended Data Fig. 2 | Correlation-derived host–microbiome interactions. a** High-level pathway overview of host–microbiome associations. For complete data and full pathway names, see Supplementary Table 2.4. **b** HUMAnN3 predicted microbial functions (MetaCyc) correlated to host colon transcription (GO biological processes) using Biobakery 3. Bacterial pyruvate fermentation and co-factor biosynthesis were negatively correlated with host side neuron survival, while microbial fatty acids were found positively associated with colon immune processes (MHC class I) and negatively with transcription regulation. These HUMAnN3-based results were generally in concordance, although much less detailed, to our MAG and metabolic model derived host-microbiome correlations (cf. Fig. 2a).

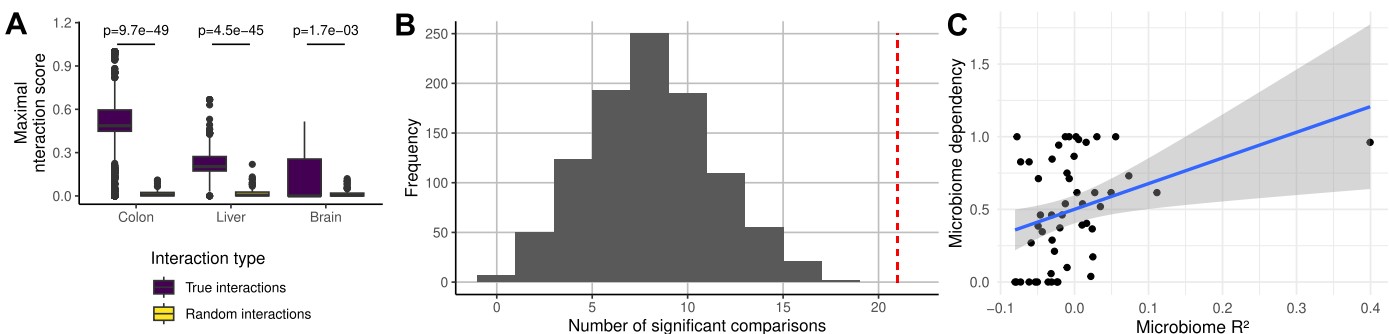

**Extended Data Fig. 3 | Metamodel validation. a** Maximal interaction scores for pairs of host–microbiome-associated processes versus 100 randomly selected pairs of host genes and microbiome reactions for each tissue (see Methods). P-values indicate the significance of the differences according to a two-sided Wilcoxon rank-sum test. **b** Randomization of germfree mouse analysis (cf. Fig. 3c). The identification of significant differences in microbiome dependence between significantly up-regulated, down-regulated as well as unregulated genes was repeated 1000 times after randomization of gene assignments and the number of significant associations was counted. The dashed red line indicates the number of significant associations in the original analysis. **c** Association between model-predicted microbiome dependency of metabolites in blood and explained variation of serum concentrations of the corresponding metabolite by microbiome composition in a human cohort (each dot is one metabolite). The association between microbiome dependency and explained variance is significant using Spearman's correlation (rho=0.43, p-value = 1.5×10-3). Boxplot elements: center line, median; box limits, 25%–75% quantiles; whiskers, 1.5x IQR; points, outliers.

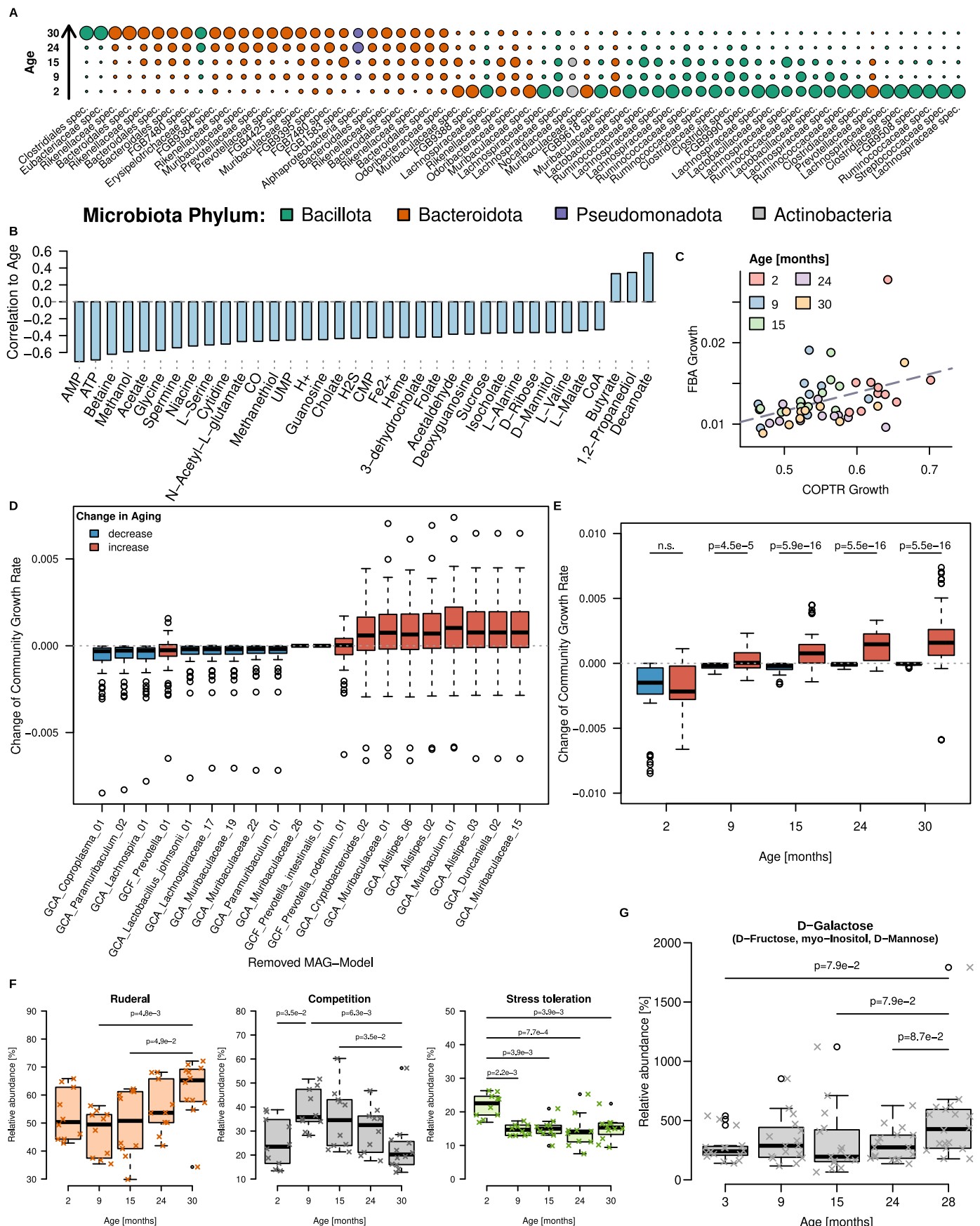

**Extended Data Fig. 4 | See next page for caption.**

**Extended Data Fig. 4 | Age-dependent microbiome changes. a** MetaPhlAn4 derived species with significant changes in abundance with host age. The MetaPhlAn-based approach identified 335 bacteria at the species level of which 69 were found differentially abundant with age. The species Lactobacillus johnsonii matched the MAG based analysis (Fig. 4a). However, due to different taxonomic databases and methods of annotation the resulting species names might vary. On phylum level, the overall aging pattern of both MetaPhlAn and MAG-based taxonomic analysis yielded comparable results, which was a general decrease of Bacillota spec. and an increase of Bacteroidota spec. with host age. **b** Age-associated changes in metabolic fluxes within the community. **c** Comparison of community growth rates derived from FBA or PTR. **d**, **e** Change of microbiome community growth rate upon removal of a single MAG from the community. The y-axis shows the difference of community growth rate compared to the full community while the x-axis names the MAG that was removed in **d** or the age group of the host in **e** (p-values via FDR corrected Wilcoxon's rank sum test). **f** Relative abundance of universal adaptive strategies in microbiome communities by age (p-values via FDR corrected Dunn's test). Sample size for **a-f**: 30mo.: n = 12, all others n = 10. **g** Metabolomics derived D-galactose concentration in mouse feces increases with age (Spearman's ρ = 0.22, unadjusted p = 0.04; Pairwise p-values via Dunn's test with FDR correction; 3mo.: n = 15, 9mo.: n = 16, 15mo.: n = 15, 24mo.: n = 17, 28mo.: n = 18). Boxplot elements: center line, median; box limits, 25%–75% quantiles; whiskers, 1.5x IQR; points, outliers.

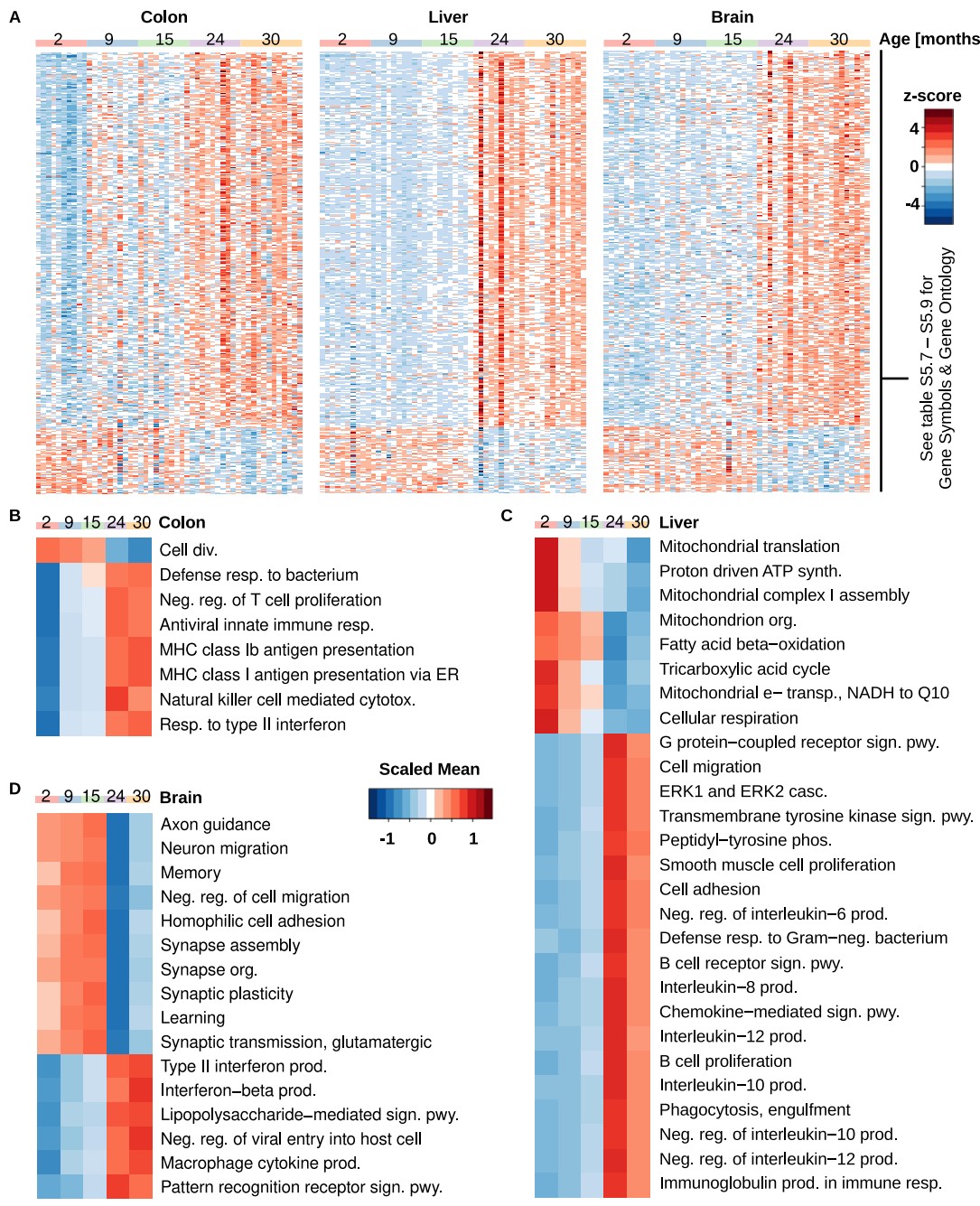

**Extended Data Fig. 5 | Aging-associated transcriptomic changes across host tissues. a** Each row represents a gene that shows common expression changes across all tissues studied (see Supplementary Table 5.7–5.9). **b–d** Enriched GO biological processes are shown as the average expression of all associated features, stratified by age group and organ. (Hypergeometric test FDR cutoff for displayed terms: colon, 10−4; liver, 10−6, brain 10−6; 30 months: n = 12; all others n = 10). For complete data and full pathway names, see Supplementary Tables 5.4–5.6.

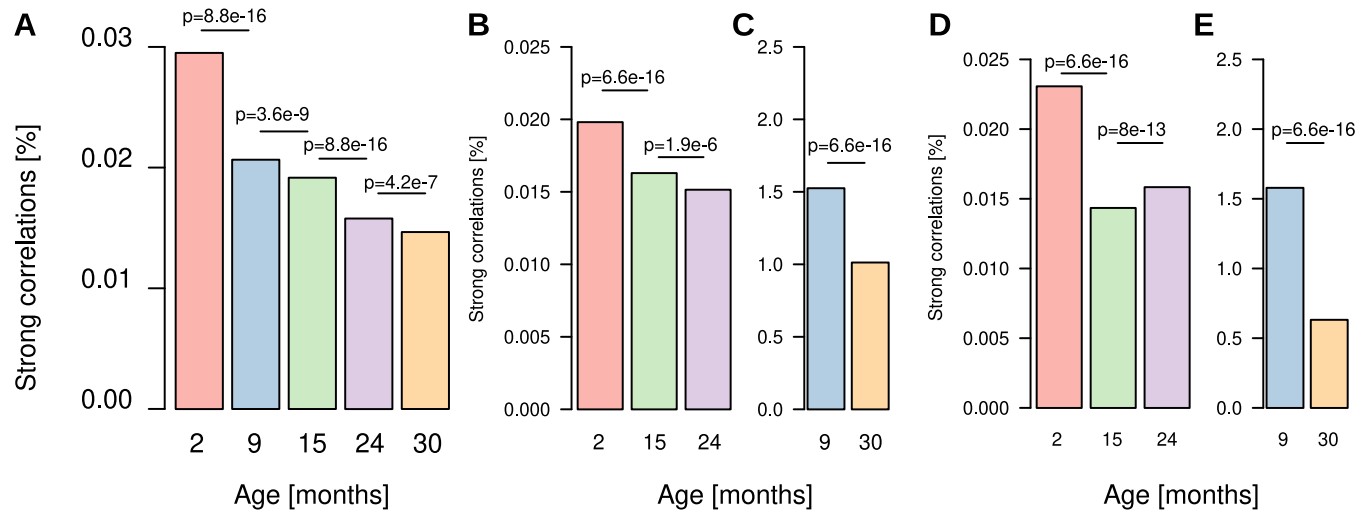

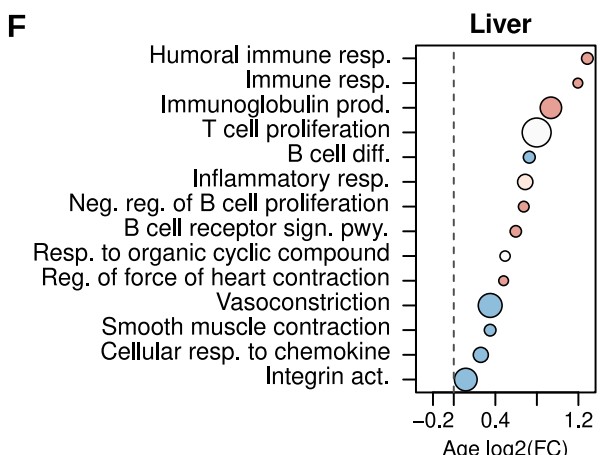

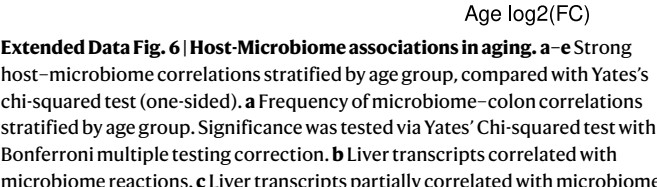

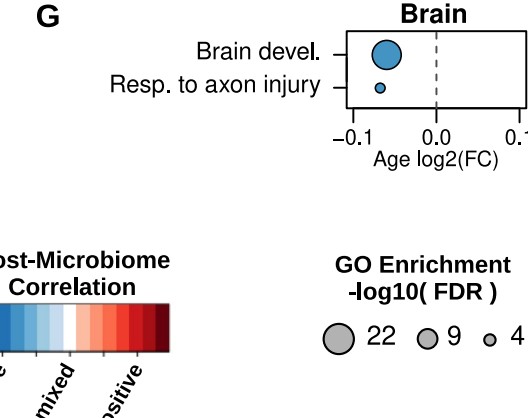

**Extended Data Fig. 6 | Host-Microbiome associations in aging. a–e** Strong host–microbiome correlations stratified by age group, compared with Yates's chi-squared test (one-sided). **a** Frequency of microbiome–colon correlations stratified by age group. Significance was tested via Yates' Chi-squared test with Bonferroni multiple testing correction. **b** Liver transcripts correlated with microbiome reactions. **c** Liver transcripts partially correlated with microbiome reactions, corrected for sequencing batch. **d** Brain transcripts correlated with microbiome reactions. **e** Brain transcripts partially correlated with microbiome reactions, corrected for sequencing batch. **f–g** Organ-specific gene expression changes with age in GO biological processes that were also correlated with microbiome metabolic functions (F = liver, G = brain). For complete data and full pathway names, see Supplementary Tables 6.2, 6.3.

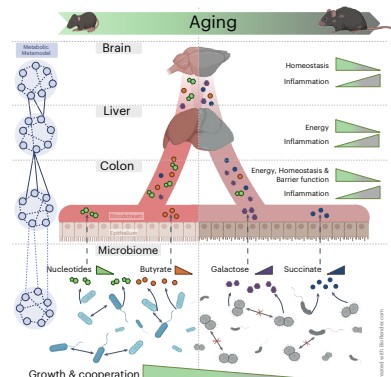

**Extended Data Fig. 7 | Summary of study highlights.** Multi-omics analysis and metabolic modeling of aging mice revealed a decline in microbiome metabolic activity, reducing beneficial host-microbe and microbe-microbe interactions. Aging-related microbiome changes led to the accumulation of pro-aging metabolites (D-galactose, succinate) and a decline in beneficial metabolites (nucleotides, butyrate). These shifts correlated with increased systemic inflammation and downregulation of essential host pathways, like energy and nucleotide metabolism, which are involved in intestinal barrier function and cellular homeostasis.

# Reporting Summary

## Statistics

For all statistical analyses, confirm that the following items are present in the figure legend, table legend, main text, or Methods section.

| n/a | Confirmed | |
|---|---|---|
| ☐ | ☒ | The exact sample size (*n*) for each experimental group/condition, given as a discrete number and unit of measurement |
| ☐ | ☒ | A statement on whether measurements were taken from distinct samples or whether the same sample was measured repeatedly |
| ☐ | ☒ | The statistical test(s) used AND whether they are one- or two-sided *Only common tests should be described solely by name; describe more complex techniques in the Methods section.* |
| ☐ | ☒ | A description of all covariates tested |
| ☐ | ☒ | A description of any assumptions or corrections, such as tests of normality and adjustment for multiple comparisons |
| ☐ | ☒ | A full description of the statistical parameters including central tendency (e.g. means) or other basic estimates (e.g. regression coefficient) AND variation (e.g. standard deviation) or associated estimates of uncertainty (e.g. confidence intervals) |
| ☐ | ☒ | For null hypothesis testing, the test statistic (e.g. *F*, *t*, *r*) with confidence intervals, effect sizes, degrees of freedom and *P* value noted *Give P values as exact values whenever suitable.* |
| ☒ | ☐ | For Bayesian analysis, information on the choice of priors and Markov chain Monte Carlo settings |
| ☒ | ☐ | For hierarchical and complex designs, identification of the appropriate level for tests and full reporting of outcomes |
| ☐ | ☒ | Estimates of effect sizes (e.g. Cohen's *d*, Pearson's *r*), indicating how they were calculated |

*Our web collection on statistics for biologists contains articles on many of the points above.*

## Software and code

Policy information about availability of computer code

| | |
|---|---|
| Data collection | No software was used for data collection. |
| Data analysis | Code: https://github.com/sciwitch/MouseMicrobiomeAging, Data: https://zenodo.org/records/13899808 |

For manuscripts utilizing custom algorithms or software that are central to the research but not yet described in published literature, software must be made available to editors and reviewers. We strongly encourage code deposition in a community repository (e.g. GitHub). See the Nature Portfolio guidelines for submitting code & software for further information.

## Data

Policy information about availability of data

All manuscripts must include a data availability statement. This statement should provide the following information, where applicable:
- Accession codes, unique identifiers, or web links for publicly available datasets
- A description of any restrictions on data availability
- For clinical datasets or third party data, please ensure that the statement adheres to our policy

Metagenomic raw read and MAG assembly data was deposited in the European Nucleotide Archive (ENA) under BioProject PRJEB73981 (ebi.ac.uk/ena/browser/view/PRJEB73981). Individual accession numbers for each MAG were listed in Supplementary Table S1.2. Gene expression data was published in the GEO database under record GSE262290 (ncbi.nlm.nih.gov/geo/query/acc.cgi?acc=GSE262290) and record GSE278548 (ncbi.nlm.nih.gov/geo/query/acc.cgi?acc=GSE278548). Metabolomics data has been made available at the MassIVE database (massive.ucsd.edu) with identifiers MSV000094409, MSV000094410 and MSV000094436. The

metamodel can be found under accession MODEL2310020001 in the BioModels database (ebi.ac.uk/biomodels/MODEL2310020001) 97. Detailed sample metadata, the microbial metabolic models and supplementary resources as well as source code used for data analysis (github.com/sciwitch/MouseMicrobiomeAging) were deposited in a Zenodo record (doi.org/10.5281/zenodo.10844503).

# Research involving human participants, their data, or biological material

Policy information about studies with <u>human participants or human data</u>. See also policy information about <u>sex, gender (identity/presentation), and sexual orientation</u> and <u>race, ethnicity and racism</u>.

| | |
|---|---|
| Reporting on sex and gender | Not applicable |
| Reporting on race, ethnicity, or other socially relevant groupings | Not applicable |
| Population characteristics | Not applicable |
| Recruitment | Not applicable |
| Ethics oversight | Not applicable |

Note that full information on the approval of the study protocol must also be provided in the manuscript.

# Field-specific reporting

Please select the one below that is the best fit for your research. If you are not sure, read the appropriate sections before making your selection.

☒ Life sciences ☐ Behavioural & social sciences ☐ Ecological, evolutionary & environmental sciences

For a reference copy of the document with all sections, see nature.com/documents/nr-reporting-summary-flat.pdf

# Life sciences study design

All studies must disclose on these points even when the disclosure is negative.

| | |
|---|---|
| Sample size | 52 main mouse cohort, 83 metabolomics mouse cohort, 24 germ-free mouse cohort. The animal numbers per age group are based on empirical data and previously published group sizes. |
| Data exclusions | No data were excluded from the analysis. For metabolomics, one sample in positive and negative mode each failed and therefore were discarded. |
| Replication | A replication – as in animal testing – was not necessary in this case, as we only took and analyzed the organs per age group. |
| Randomization | Animals for the age groups were randomly selected from a large breeding population. However, these animals are phenotypically recognizable due to their age, so blinding was not meaningful in this case. |
| Blinding | See randomization |

# Behavioural & social sciences study design

All studies must disclose on these points even when the disclosure is negative.

| | |
|---|---|
| Study description | *Briefly describe the study type including whether data are quantitative, qualitative, or mixed-methods (e.g. qualitative cross-sectional, quantitative experimental, mixed-methods case study).* |
| Research sample | *State the research sample (e.g. Harvard university undergraduates, villagers in rural India) and provide relevant demographic information (e.g. age, sex) and indicate whether the sample is representative. Provide a rationale for the study sample chosen. For studies involving existing datasets, please describe the dataset and source.* |
| Sampling strategy | *Describe the sampling procedure (e.g. random, snowball, stratified, convenience). Describe the statistical methods that were used to predetermine sample size OR if no sample-size calculation was performed, describe how sample sizes were chosen and provide a rationale for why these sample sizes are sufficient. For qualitative data, please indicate whether data saturation was considered, and what criteria were used to decide that no further sampling was needed.* |
| Data collection | *Provide details about the data collection procedure, including the instruments or devices used to record the data (e.g. pen and paper, computer, eye tracker, video or audio equipment) whether anyone was present besides the participant(s) and the researcher, and whether the researcher was blind to experimental condition and/or the study hypothesis during data collection.* |

| Timing | *Indicate the start and stop dates of data collection. If there is a gap between collection periods, state the dates for each sample cohort.* |
| Data exclusions | *If no data were excluded from the analyses, state so OR if data were excluded, provide the exact number of exclusions and the rationale behind them, indicating whether exclusion criteria were pre-established.* |
| Non-participation | *State how many participants dropped out/declined participation and the reason(s) given OR provide response rate OR state that no participants dropped out/declined participation.* |
| Randomization | *If participants were not allocated into experimental groups, state so OR describe how participants were allocated to groups, and if allocation was not random, describe how covariates were controlled.* |

# Ecological, evolutionary & environmental sciences study design

All studies must disclose on these points even when the disclosure is negative.

| Study description | *Briefly describe the study. For quantitative data include treatment factors and interactions, design structure (e.g. factorial, nested, hierarchical), nature and number of experimental units and replicates.* |
| Research sample | *Describe the research sample (e.g. a group of tagged Passer domesticus, all Stenocereus thurberi within Organ Pipe Cactus National Monument), and provide a rationale for the sample choice. When relevant, describe the organism taxa, source, sex, age range and any manipulations. State what population the sample is meant to represent when applicable. For studies involving existing datasets, describe the data and its source.* |
| Sampling strategy | *Note the sampling procedure. Describe the statistical methods that were used to predetermine sample size OR if no sample-size calculation was performed, describe how sample sizes were chosen and provide a rationale for why these sample sizes are sufficient.* |
| Data collection | *Describe the data collection procedure, including who recorded the data and how.* |
| Timing and spatial scale | *Indicate the start and stop dates of data collection, noting the frequency and periodicity of sampling and providing a rationale for these choices. If there is a gap between collection periods, state the dates for each sample cohort. Specify the spatial scale from which the data are taken* |
| Data exclusions | *If no data were excluded from the analyses, state so OR if data were excluded, describe the exclusions and the rationale behind them, indicating whether exclusion criteria were pre-established.* |
| Reproducibility | *Describe the measures taken to verify the reproducibility of experimental findings. For each experiment, note whether any attempts to repeat the experiment failed OR state that all attempts to repeat the experiment were successful.* |
| Randomization | *Describe how samples/organisms/participants were allocated into groups. If allocation was not random, describe how covariates were controlled. If this is not relevant to your study, explain why.* |
| Blinding | *Describe the extent of blinding used during data acquisition and analysis. If blinding was not possible, describe why OR explain why blinding was not relevant to your study.* |

Did the study involve field work? ☐ Yes ☐ No

## Field work, collection and transport

| Field conditions | *Describe the study conditions for field work, providing relevant parameters (e.g. temperature, rainfall).* |
| Location | *State the location of the sampling or experiment, providing relevant parameters (e.g. latitude and longitude, elevation, water depth).* |
| Access & import/export | *Describe the efforts you have made to access habitats and to collect and import/export your samples in a responsible manner and in compliance with local, national and international laws, noting any permits that were obtained (give the name of the issuing authority, the date of issue, and any identifying information).* |
| Disturbance | *Describe any disturbance caused by the study and how it was minimized.* |

# Reporting for specific materials, systems and methods

We require information from authors about some types of materials, experimental systems and methods used in many studies. Here, indicate whether each material, system or method listed is relevant to your study. If you are not sure if a list item applies to your research, read the appropriate section before selecting a response.

## Materials & experimental systems

| n/a | Involved in the study |
|---|---|
| ☒ ☐ | Antibodies |
| ☒ ☐ | Eukaryotic cell lines |
| ☒ ☐ | Palaeontology and archaeology |
| ☐ ☒ | Animals and other organisms |
| ☒ ☐ | Clinical data |
| ☒ ☐ | Dual use research of concern |
| ☒ ☐ | Plants |

## Methods

| n/a | Involved in the study |
|---|---|
| ☒ ☐ | ChIP-seq |
| ☒ ☐ | Flow cytometry |
| ☒ ☐ | MRI-based neuroimaging |

# Antibodies

| | |
|---|---|
| Antibodies used | *Describe all antibodies used in the study; as applicable, provide supplier name, catalog number, clone name, and lot number.* |
| Validation | *Describe the validation of each primary antibody for the species and application, noting any validation statements on the manufacturer's website, relevant citations, antibody profiles in online databases, or data provided in the manuscript.* |

# Eukaryotic cell lines

Policy information about cell lines and Sex and Gender in Research

| | |
|---|---|
| Cell line source(s) | *State the source of each cell line used and the sex of all primary cell lines and cells derived from human participants or vertebrate models.* |
| Authentication | *Describe the authentication procedures for each cell line used OR declare that none of the cell lines used were authenticated.* |
| Mycoplasma contamination | *Confirm that all cell lines tested negative for mycoplasma contamination OR describe the results of the testing for mycoplasma contamination OR declare that the cell lines were not tested for mycoplasma contamination.* |
| Commonly misidentified lines (See ICLAC register) | *Name any commonly misidentified cell lines used in the study and provide a rationale for their use.* |

# Palaeontology and Archaeology

| | |
|---|---|
| Specimen provenance | *Provide provenance information for specimens and describe permits that were obtained for the work (including the name of the issuing authority, the date of issue, and any identifying information). Permits should encompass collection and, where applicable, export.* |
| Specimen deposition | *Indicate where the specimens have been deposited to permit free access by other researchers.* |
| Dating methods | *If new dates are provided, describe how they were obtained (e.g. collection, storage, sample pretreatment and measurement), where they were obtained (i.e. lab name), the calibration program and the protocol for quality assurance OR state that no new dates are provided.* |

☐ Tick this box to confirm that the raw and calibrated dates are available in the paper or in Supplementary Information.

| | |
|---|---|
| Ethics oversight | *Identify the organization(s) that approved or provided guidance on the study protocol, OR state that no ethical approval or guidance was required and explain why not.* |

Note that full information on the approval of the study protocol must also be provided in the manuscript.

# Animals and other research organisms

Policy information about studies involving animals; ARRIVE guidelines recommended for reporting animal research, and Sex and Gender in Research

| | |
|---|---|
| Laboratory animals | Mus Musculus C57BL/6J/Ukj (3 months, 9 months, 15 months, 24 months and 30 months of age) and Mus Musculus C57BL/6J (10 weeks of age) |
| Wild animals | No wild animals were used in this study. |
| Reporting on sex | all mice of main and metabolomics cohort were male, all mice of germ-free cohort were female, sex is reported in Methods section, sex was determined be examining externally visible gonads |
| Field-collected samples | No field-collected samples were used in this study. |

| Ethics oversight | Aging/metabolomics mouse cohort: All studies were performed in strict compliance with the recommendations of the European Commission for the protection of animals used for scientific purposes and with the approval of the local government (Thüringer Landesamt für Verbraucherschutz, Germany; license: 02-024/15; TWZ-000-2017. Germfree mouse cohort: All animal protocols were approved by the Gothenburg Animal Ethics Committee (vote #2652-19). |
|---|---|

Note that full information on the approval of the study protocol must also be provided in the manuscript.

# Clinical data

Policy information about clinical studies

All manuscripts should comply with the ICMJE guidelines for publication of clinical research and a completed CONSORT checklist must be included with all submissions.

| Clinical trial registration | *Provide the trial registration number from ClinicalTrials.gov or an equivalent agency.* |
|---|---|
| Study protocol | *Note where the full trial protocol can be accessed OR if not available, explain why.* |
| Data collection | *Describe the settings and locales of data collection, noting the time periods of recruitment and data collection.* |
| Outcomes | *Describe how you pre-defined primary and secondary outcome measures and how you assessed these measures.* |

# Dual use research of concern

Policy information about dual use research of concern

## Hazards

Could the accidental, deliberate or reckless misuse of agents or technologies generated in the work, or the application of information presented in the manuscript, pose a threat to:

No | Yes

☐ ☐ Public health

☐ ☐ National security

☐ ☐ Crops and/or livestock

☐ ☐ Ecosystems

☐ ☐ Any other significant area

## Experiments of concern

Does the work involve any of these experiments of concern:

No | Yes

☐ ☐ Demonstrate how to render a vaccine ineffective

☐ ☐ Confer resistance to therapeutically useful antibiotics or antiviral agents

☐ ☐ Enhance the virulence of a pathogen or render a nonpathogen virulent

☐ ☐ Increase transmissibility of a pathogen

☐ ☐ Alter the host range of a pathogen

☐ ☐ Enable evasion of diagnostic/detection modalities

☐ ☐ Enable the weaponization of a biological agent or toxin

☐ ☐ Any other potentially harmful combination of experiments and agents

# Plants

| | |
|---|---|
| Seed stocks | *Report on the source of all seed stocks or other plant material used. If applicable, state the seed stock centre and catalogue number. If plant specimens were collected from the field, describe the collection location, date and sampling procedures.* |
| Novel plant genotypes | *Describe the methods by which all novel plant genotypes were produced. This includes those generated by transgenic approaches, gene editing, chemical/radiation-based mutagenesis and hybridization. For transgenic lines, describe the transformation method, the number of independent lines analyzed and the generation upon which experiments were performed. For gene-edited lines, describe the editor used, the endogenous sequence targeted for editing, the targeting guide RNA sequence (if applicable) and how the editor was applied.* |
| Authentication | *Describe any authentication procedures for each seed stock used or novel genotype generated. Describe any experiments used to assess the effect of a mutation and, where applicable, how potential secondary effects (e.g. second site T-DNA insertions, mosiacism, off-target gene editing) were examined.* |

# ChIP-seq

## Data deposition

☐ Confirm that both raw and final processed data have been deposited in a public database such as GEO.

☐ Confirm that you have deposited or provided access to graph files (e.g. BED files) for the called peaks.

| | |
|---|---|
| Data access links<br>*May remain private before publication.* | *For "Initial submission" or "Revised version" documents, provide reviewer access links.  For your "Final submission" document, provide a link to the deposited data.* |
| Files in database submission | *Provide a list of all files available in the database submission.* |
| Genome browser session<br>(e.g. UCSC) | *Provide a link to an anonymized genome browser session for "Initial submission" and "Revised version" documents only, to enable peer review.  Write "no longer applicable" for "Final submission" documents.* |

## Methodology

| | |
|---|---|
| Replicates | *Describe the experimental replicates, specifying number, type and replicate agreement.* |
| Sequencing depth | *Describe the sequencing depth for each experiment, providing the total number of reads, uniquely mapped reads, length of reads and whether they were paired- or single-end.* |
| Antibodies | *Describe the antibodies used for the ChIP-seq experiments; as applicable, provide supplier name, catalog number, clone name, and lot number.* |
| Peak calling parameters | *Specify the command line program and parameters used for read mapping and peak calling, including the ChIP, control and index files used.* |
| Data quality | *Describe the methods used to ensure data quality in full detail, including how many peaks are at FDR 5% and above 5-fold enrichment.* |
| Software | *Describe the software used to collect and analyze the ChIP-seq data. For custom code that has been deposited into a community repository, provide accession details.* |

# Flow Cytometry

## Plots

Confirm that:

☐ The axis labels state the marker and fluorochrome used (e.g. CD4-FITC).

☐ The axis scales are clearly visible. Include numbers along axes only for bottom left plot of group (a 'group' is an analysis of identical markers).

☐ All plots are contour plots with outliers or pseudocolor plots.

☐ A numerical value for number of cells or percentage (with statistics) is provided.

## Methodology

| | |
|---|---|
| Sample preparation | *Describe the sample preparation, detailing the biological source of the cells and any tissue processing steps used.* |
| Instrument | *Identify the instrument used for data collection, specifying make and model number.* |
| Software | *Describe the software used to collect and analyze the flow cytometry data. For custom code that has been deposited into a community repository, provide accession details.* |

| Cell population abundance | *Describe the abundance of the relevant cell populations within post-sort fractions, providing details on the purity of the samples and how it was determined.* |
|---|---|
| Gating strategy | *Describe the gating strategy used for all relevant experiments, specifying the preliminary FSC/SSC gates of the starting cell population, indicating where boundaries between "positive" and "negative" staining cell populations are defined.* |

☐ Tick this box to confirm that a figure exemplifying the gating strategy is provided in the Supplementary Information.

# Magnetic resonance imaging

## Experimental design

| Design type | *Indicate task or resting state; event-related or block design.* |
|---|---|
| Design specifications | *Specify the number of blocks, trials or experimental units per session and/or subject, and specify the length of each trial or block (if trials are blocked) and interval between trials.* |
| Behavioral performance measures | *State number and/or type of variables recorded (e.g. correct button press, response time) and what statistics were used to establish that the subjects were performing the task as expected (e.g. mean, range, and/or standard deviation across subjects).* |

## Acquisition

| Imaging type(s) | *Specify: functional, structural, diffusion, perfusion.* |
|---|---|
| Field strength | *Specify in Tesla* |
| Sequence & imaging parameters | *Specify the pulse sequence type (gradient echo, spin echo, etc.), imaging type (EPI, spiral, etc.), field of view, matrix size, slice thickness, orientation and TE/TR/flip angle.* |
| Area of acquisition | *State whether a whole brain scan was used OR define the area of acquisition, describing how the region was determined.* |

Diffusion MRI ☐ Used ☐ Not used

## Preprocessing

| Preprocessing software | *Provide detail on software version and revision number and on specific parameters (model/functions, brain extraction, segmentation, smoothing kernel size, etc.).* |
|---|---|
| Normalization | *If data were normalized/standardized, describe the approach(es): specify linear or non-linear and define image types used for transformation OR indicate that data were not normalized and explain rationale for lack of normalization.* |
| Normalization template | *Describe the template used for normalization/transformation, specifying subject space or group standardized space (e.g. original Talairach, MNI305, ICBM152) OR indicate that the data were not normalized.* |
| Noise and artifact removal | *Describe your procedure(s) for artifact and structured noise removal, specifying motion parameters, tissue signals and physiological signals (heart rate, respiration).* |
| Volume censoring | *Define your software and/or method and criteria for volume censoring, and state the extent of such censoring.* |

## Statistical modeling & inference

| Model type and settings | *Specify type (mass univariate, multivariate, RSA, predictive, etc.) and describe essential details of the model at the first and second levels (e.g. fixed, random or mixed effects; drift or auto-correlation).* |
|---|---|
| Effect(s) tested | *Define precise effect in terms of the task or stimulus conditions instead of psychological concepts and indicate whether ANOVA or factorial designs were used.* |

Specify type of analysis: ☐ Whole brain ☐ ROI-based ☐ Both

| Statistic type for inference<br>(See Eklund et al. 2016) | *Specify voxel-wise or cluster-wise and report all relevant parameters for cluster-wise methods.* |
|---|---|
| Correction | *Describe the type of correction and how it is obtained for multiple comparisons (e.g. FWE, FDR, permutation or Monte Carlo).* |

## Models & analysis

| n/a | Involved in the study |
|-----|----------------------|
| ☐ | ☐ Functional and/or effective connectivity |
| ☐ | ☐ Graph analysis |
| ☐ | ☐ Multivariate modeling or predictive analysis |

**Functional and/or effective connectivity**
*Report the measures of dependence used and the model details (e.g. Pearson correlation, partial correlation, mutual information).*

**Graph analysis**
*Report the dependent variable and connectivity measure, specifying weighted graph or binarized graph, subject- or group-level, and the global and/or node summaries used (e.g. clustering coefficient, efficiency, etc.).*

**Multivariate modeling and predictive analysis**
*Specify independent variables, features extraction and dimension reduction, model, training and evaluation metrics.*

