## [Peer Review File · Nature Microbiology]

Metabolic modeling reveals the aging-associated decline of host-microbiome metabolic interactions in mice

Corresponding Author: Professor Christoph Kaleta

Version 0:

Reviewer comments:

Reviewer #1

(Remarks to the Author)

In this study, Best et al. investigated metabolic microbiota-host interactions in aging employing multi-omics measurements (i.e., metagenomics, transcriptomics and metabolomics) of male mice at different age. Through various types of association and enrichment analysis, as well as different flavors of genome-scale metabolic modeling the authors correlate microbiome composition (and putative functions) to differences in gene expression in host tissues, namely liver and brain. The presented findings remain rather generic and descriptive based on the performed associations.

The main conclusions of this work are poorly supported by the shown data. As experimental validations of the main claims are lacking, the study remains descriptive relying on associations. While human-centered studies are often based on association due to limitations of sample access and experimental restrictions, studies focusing on mouse models should have the ambition to experimentally validate key claims made.

The paper is not written in a very clear language that would help the reader to easily follow the logic of the study and the performed analysis. In several instances, the applied experiments and analytical procedures lack sufficient details to evaluate their suitability to derive the drawn conclusions. Also, certain assumption made by the authors seem poorly justified. It seems that the authors re-mine the same data multiple times with different, but similar approaches, which are not clearly motivated and explained. For example, it remains unclear what the differences between the meta-organism flux balance analysis and community flux balance analysis are.

Although the presented work is of interest to the field of microbiome research it seems preliminary in its current presentation. Rather than providing novel biological insights into metabolic microbiome-host interactions, the works main strength seems to consists of the data provided and the application and combination of computational methods to mine the data.

Specific comments:

1. It remains unclear how the abundance of a metabolic pathway (based on compositional metagenomics data) provides any information about the metabolic activity of a given microbial community. Several of the performed analyses are based on this assumption, leading the authors to claim that the metabolic activity of the aging microbiome is reduced.
2. The authors employ flux-balance analysis for several of their analysis using bacterial growth as objective function. Generally, in the gut microbiome and particularly in the large intestine (from which metagenomic samples were collected) it remains unclear why this assumption is justified. The applied method to estimate bacterial growth rates from metagenomic data is controversially discussed in the field, as the model is based on *E. coli* and little experimental evidence exists that the same principle applies to gut bacterial taxa.
3. The authors suggest that the microbiome provides pyrimidines and ethanol to the brain of the host. This is an interesting finding. In particular ethanol production by gut microbes and its delivery to the brain, circumventing first-pass metabolism by the intestinal epithelium and the liver, warrants experimental validation.
4. Line 220. The statement that metamodel results are consistent with correlation analyses might be a circular argument. As same data was used for both types of analysis, it is not surprising that the results are consistent.
5. It is not clear whether and how the performed flux analyses contribute to the findings or whether generic correlations of metabolic pathways/GO terms would result in the same associations.
6. It remains unclear whether and how the genome-scale metabolic models were gap-filled.

7. The terminology of 'metabolites identified' (e.g., line 193) used throughout the manuscript is not precise. The authors seem to refer to exchange reactions of the model rather than actual metabolite identification (e.g., metabolomics).
8. Text in Fig. 2 is not readable.
9. Fig. 3. Subsystem enrichment. Was FDR correction performed?
10. Line 278. It is unclear what is meant by 'metabolic clusters'
11. Line 337. The authors focus their analysis on succinate (based on predictions, not measurements), but it remains unclear why succinate was selected among all the metabolites shown in fig. 4C.
12. Line 451-455. The authors state that the host most often depend on central metabolic microbiome reactions. Would this observation not be biased by the coverage of reactions and/or the fact that microbiome gene annotation and genome-scale metabolic model construction are best for these reactions?

Reviewer #2

(Remarks to the Author)

Summary -

The authors present an interesting exploration of host-microbial metabolic interplay in a dataset of C57BL/6J mice at different ages (2, 8, 15, 24, and 30 months). The data set includes transcriptomic data from brain, liver, and colon tissue, along with fecal metagenomic data, for each mouse. Much of the work presented in the manuscript is centered around genome-scale metabolic modeling and flux balance analysis of both the microbiota and the host, to explore how the microbiota may contribute to metabolic changes that occur with aging. This is an ambitious and complex analysis, and the authors have done an impressive job navigating many of the potential pitfalls. The paper is well written. Many of the results are intuitive and match with the literature, and there are several newer insights as well. Overall, I commend the authors on their work. However, I have a few concerns about methods and framing and a few suggestions, which I outline below.

Specific Comments -

It's always difficult to validate these types of models, but some additional follow-up validation beyond comparison to the literature would up the impact of the work.

lines 51-54: The effects of FMT from young mice into older mice appear to be more complicated than what's presented here. The literature is a bit split on the topic. For example, in the Boehme paper cited by the authors, they state that it shows evidence that FMTs from young mice reduce inflammation and improve neurological health in older mice. However, one of the more striking results in that paper is that IL-10 (and anti-inflammatory cytokine) is significantly increased by FMT from an age-matched (older) mouse. In another paper, germ free mice receiving an FMT from older animals showed greater hippocampal neurogenesis and intestinal growth than germ free mice that received FMTs from younger animals (<https://pubmed.ncbi.nlm.nih.gov/31723038/>). These ambiguities are discussed in the following commentary (<https://www.nature.com/articles/s43587-022-00294-w>).

lines 86-87: Related to the point above, it may be important to separate the aging process from chronic disease and frailty. These things are all correlated, to be sure, but it's possible to find chronologically older individuals that are much healthier than their peers. These older individuals may show signs of aging, but some of these signatures may be compensatory and promote resilience, while other signatures may promote frailty. Some discussion of how there may be signatures of healthy and unhealthy aging would be useful here. Similar ideas are presented in terms of 'biological age' (e.g., <https://pubmed.ncbi.nlm.nih.gov/31724055/>).

lines 185-186: It looks like you're using a 'bag of genes' approach to modeling the microbiota and the host. This is probably prudent when using de novo models that haven't been curated (i.e., any inaccuracies within a given model are masked by averaging across everything). However, there are emerging methods for efficiently building multi-compartmental community-scale models (<https://journals.asm.org/doi/10.1128/msystems.01270-22>). Do you think you would gain additional insights if each taxon were compartmentalized?

Figure 3: I was surprised to see that ATP is one of the metabolites produced by the microbiota that is taken up by the host. Why would microbes export so much ATP? I see there are some experimental studies of how microbially-derived ATP can impact immune function, but I wonder if this is largely due to cell lysis or due to active transport of ATP out of the cell? Do you think that this ATP transport may be over-estimated due to the 'bag of genes' approach used here? Should this export be confined to specific taxa? Would that impact the global result?

Also in Figure 3, you show that the brain interaction scores are no different from random, but then you go into analyzing these results anyway. Why?

Figure 4 and S4: Not sure how model growth rates were obtained based on the methods. Were these just fixed based on the

relative abundances? I'm not totally sure how to interpret these trends if it's the median relative abundance that's decreasing with age. And there appears to be a contradiction between growth rates and the fraction of 'ruderal' species. Ruderals (fast growers) are increasing as a proportion of the community with age, but the median community growth rate is declining?

lines 252-255: It's interesting to note that in humans, the maintenance of Bacteroidota in old age is associated with poorer health, while healthier older individuals are relatively enriched in Bacillota (see your Wilmanski ref).

line 269: In your community FBA model, it was unclear how you dealt with individual growth rates. Reference 43 appears to be the wrong citation for this method (<https://pubmed.ncbi.nlm.nih.gov/30273559/>). From the methods section, it looks you constrain the individual growth rates to be equivalent to the relative abundances, and then optimize for community growth? Or do you allow for growth rates to emerge from the objective function somehow? If you're constraining the rates using abundances, what does it mean to compare these 'rates' to the PTRs? I suppose the assumption is that abundance is equivalent to rate in an exponentially-growing system? Maybe add more discussion about these points in the text.

lines 447-450: It looks like you didn't implement any objective functions for the microbiota and the host in these meta-models, correct? These analyses are all based on flux variability analysis?

454-457: Do you think it's circular to say that the host tended to rely upon central metabolic reactions? Aren't these metabolisms, by definition, best-captured by these models? Or do you think this result is surprising and not something you'd expect from randomly sampling the models?

lines 729-732: Were bile acids and mucins added to the diet before it was used to constrain the microbiota models?

line 809: Do you mean 'constraining' instead of 'draining'? Were the taxon biomasses fixed based on data?

lines 834-843: Why model pair-wise? What if pairwise interactions are influenced by the community context? It should be possible to knock individual taxa out from a model, one at a time, and see how this impacts the growth of all the other taxa. This would provide community-specific interaction effects for each pair of taxa. A bit more complex, because the same pair may vary in their interaction across communities. But it may also be more realistic.

line 881: Why have different FDR cutoffs? It makes it a bit harder to compare the number of hits across analyses, to see how strong the global signals are.

line 1159: So was FVA the only analysis run on the meta-models? Did you not try to implement objective functions?

Reviewer #3

(Remarks to the Author)

The manuscript by Best L, et al., presents an extensive analysis of mice microbiomes in relation to ageing, combining multi-omics data. The manuscript is well-organized and written, and the flow is easy to follow. The main concern is about the use of MAGs (although derived from hybrid assembly) as a proxy for both taxonomic and functional profiling of the microbiome in mice, instead of directly using (and comparing with the profiles from MAGs) tools to directly taxonomic and functional profile metagenomes as whole. Indeed, MAGs are usually reconstructed from species for which enough sequenced data is available, so might miss low-abundant species that can still play key roles.

* "reduction in metabolic activity within the aging microbiome [...] attributed to reduced beneficial interactions in the microbiome"  Could these be affected by the compositional change rather than the reduced interactions? Or are the reduced interactions a consequence of a different composition?

* "These pathways could serve as future targets for the development of microbiome-based therapies against aging"  "against" might be a strong concluding word, considering rephrasing.

* Why not use other tools, either k-mer or marker-based, for the taxonomic profiling? These could give a broader view of the microbiome than using only MAGs.

* Lines 104-105: "Notably, we used more stringent cutoffs ($\geq 80\%$ completeness and $\leq 10\%$ contamination)"  While this might be true for completeness, the usual threshold is at 50%, contamination is usually capped at 5%, so 10% might allow for more external DNA sequences that can hinder taxonomic assignment and phylogenetic reconstruction.

* As per my comment above on the taxonomic assignment, functional profiling could have been done directly from the metagenomes and not using MAGs as a proxy. Why this was not done? The authors could also compare functional profiles using both methods to improve the reliability of the results.

* The whole "Metabolic metaorganism models reveal widespread metabolic interactions between host and microbiota" paragraph is well described, although only based on modelling and enrichment analysis. It would be very nice to have experimental validation to corroborate some results.

Decision Letter:

1st July 2024

Dear Professor Kaleta,

Thank you for your patience while your manuscript "Metabolic modeling reveals the aging-associated decline of host-microbiome metabolic interactions in mice" was under peer-review at Nature Microbiology. It has now been seen by 3 referees, whose expertise and comments you will find at the end of this email. While they find your work of interest, they also raise some concerns regarding certain issues. We are very interested in the possibility of publishing your study in Nature Microbiology, but would like to consider your response to these concerns in the form of a revised manuscript before we make a final decision on publication.

In particular, you will see that all referees have asked that you experimentally validate the main findings and provide more methods details as well as justification for your approaches. We feel that these are critical points which would need to be addressed for us to further consider a revised manuscript, alongside the remaining issues outlined in the referees' reports.

Please include a data availability statement as a separate section after Methods but before references, under the heading "Data Availability". This section should inform readers about the availability of the data used to support the conclusions of your study. This information includes accession codes to public repositories (data banks for protein, DNA or RNA sequences, microarray, proteomics data etc...), references to source data published alongside the paper, unique identifiers such as URLs to data repository entries, or data set DOIs, and any other statement about data availability. At a minimum, you should include the following statement: "The data that support the findings of this study are available from the corresponding author upon request", mentioning any restrictions on availability. If DOIs are provided, we also strongly encourage including these in the Reference list (authors, title, publisher (repository name), identifier, year). For more guidance on how to write this section please see: <http://www.nature.com/authors/policies/data/data-availability-statements-data-citations.pdf>

* If you have not done so already we suggest that you begin to revise your manuscript so that it conforms to our Article format instructions at <http://www.nature.com/nmicrobiol/info/final-submission>. Refer also to any guidelines provided in this letter.

When submitting the revised version of your manuscript, please pay close attention to our [href="https://www.nature.com/nature-portfolio/editorial-policies/image-integrity">Digital Image Integrity Guidelines](https://www.nature.com/nature-portfolio/editorial-policies/image-integrity) and to the following points below:

Link Redacted

Note: This url links to your confidential homepage and associated information about manuscripts you may have submitted or be reviewing for us. If you wish to forward this e-mail to co-authors, please delete this link to your homepage first.

Nature Microbiology is committed to improving transparency in authorship. As part of our efforts in this direction, we are now requesting that all authors identified as 'corresponding author' on published papers create and link their Open Researcher and Contributor Identifier (ORCID) with their account on the Manuscript Tracking System (MTS), prior to acceptance. This applies to primary research papers only. ORCID helps the scientific community achieve unambiguous attribution of all scholarly contributions. You can create and link your ORCID from the home page of the MTS by clicking on 'Modify my Springer Nature

account'. For more information please visit www.springernature.com/orcid.

If you wish to submit a suitably revised manuscript we would hope to receive it within 6 months. If you cannot send it within this time, please let us know. We will be happy to consider your revision, even if a similar study has been accepted for publication at Nature Microbiology or published elsewhere (up to a maximum of 6 months).

Yours sincerely,

Reviewer Expertise:

Referee #1: metabolic modeling, gut microbiome
Referee #2: metabolic modelling, gut microbiome, ageing
Referee #3: gut microbiome, omics, ageing

Reviewer Comments:

Reviewer #1 (Remarks to the Author):

In this study, Best et al. investigated metabolic microbiota-host interactions in aging employing multi-omics measurements (i.e., metagenomics, transcriptomics and metabolomics) of male mice at different age. Through various types of association and enrichment analysis, as well as different flavors of genome-scale metabolic modeling the authors correlate microbiome composition (and putative functions) to differences in gene expression in host tissues, namely liver and brain. The presented findings remain rather generic and descriptive based on the performed associations.

The main conclusions of this work are poorly supported by the shown data. As experimental validations of the main claims are lacking, the study remains descriptive relying on associations. While human-centered studies are often based on association due to limitations of sample access and experimental restrictions, studies focusing on mouse models should have the ambition to experimentally validate key claims made.

The paper is not written in a very clear language that would help the reader to easily follow the logic of the study and the performed analysis. In several instances, the applied experiments and analytical procedures lack sufficient details to evaluate their suitability to derive the drawn conclusions. Also, certain assumption made by the authors seem poorly justified. It seems that the authors re-mine the same data multiple times with different, but similar approaches, which are not clearly motivated and explained. For example, it remains unclear what the differences between the meta-organism flux balance analysis and community flux balance analysis are.

Although the presented work is of interest to the field of microbiome research it seems preliminary in its current presentation. Rather than providing novel biological insights into metabolic microbiome-host interactions, the work's main strength seems to consist of the data provided and the application and combination of computational methods to mine the data.

Specific comments:

1. It remains unclear how the abundance of a metabolic pathway (based on compositional metagenomics data) provides any information about the metabolic activity of a given microbial community. Several of the performed analyses are based on this assumption, leading the authors to claim that the metabolic activity of the aging microbiome is reduced.
2. The authors employ flux-balance analysis for several of their analyses using bacterial growth as objective function. Generally, in the gut microbiome and particularly in the large intestine (from which metagenomic samples were collected) it remains unclear why this assumption is justified. The applied method to estimate bacterial growth rates from metagenomic data is controversially discussed in the field, as the model is based on *E. coli* and little experimental evidence exists that the same principle applies to gut bacterial taxa.
3. The authors suggest that the microbiome provides pyrimidines and ethanol to the brain of the host. This is an interesting finding. In particular ethanol production by gut microbes and its delivery to the brain, circumventing first-pass metabolism by the intestinal epithelium and the liver, warrants experimental validation.
4. Line 220. The statement that metamodel results are consistent with correlation analyses might be a circular argument. As same data was used for both types of analysis, it is not surprising that the results are consistent.
5. It is not clear whether and how the performed flux analyses contribute to the findings or whether generic correlations of metabolic pathways/GO terms would result in the same associations.
6. It remains unclear whether and how the genome-scale metabolic models were gap-filled.
7. The terminology of 'metabolites identified' (e.g., line 193) used throughout the manuscript is not precise. The authors seem to

refer to exchange reactions of the model rather than actual metabolite identification (e.g., metabolomics).

8. Text in Fig. 2 is not readable.

9. Fig. 3. Subsystem enrichment. Was FDR correction performed?

10. Line 278. It is unclear what is meant by 'metabolic clusters'

11. Line 337. The authors focus their analysis on succinate (based on predictions, not measurements), but it remains unclear why succinate was selected among all the metabolites shown in fig. 4C.

12. Line 451-455. The authors state that the host most often depend on central metabolic microbiome reactions. Would this observation not be biased by the coverage of reactions and/or the fact that microbiome gene annotation and genome-scale metabolic model construction are best for these reactions?

Reviewer #2 (Remarks to the Author):

Summary -

The authors present an interesting exploration of host-microbial metabolic interplay in a dataset of C57BL/6J mice at different ages (2, 8, 15, 24, and 30 months). The data set includes transcriptomic data from brain, liver, and colon tissue, along with fecal metagenomic data, for each mouse. Much of the work presented in the manuscript is centered around genome-scale metabolic modeling and flux balance analysis of both the microbiota and the host, to explore how the microbiota may contribute to metabolic changes that occur with aging. This is an ambitious and complex analysis, and the authors have done an impressive job navigating many of the potential pitfalls. The paper is well written. Many of the results are intuitive and match with the literature, and there are several newer insights as well. Overall, I commend the authors on their work. However, I have a few concerns about methods and framing and a few suggestions, which I outline below.

Specific Comments -

It's always difficult to validate these types of models, but some additional follow-up validation beyond comparison to the literature would up the impact of the work.

lines 51-54: The effects of FMT from young mice into older mice appear to be more complicated than what's presented here. The literature is a bit split on the topic. For example, in the Boehme paper cited by the authors, they state that it shows evidence that FMTs from young mice reduce inflammation and improve neurological health in older mice. However, one of the more striking results in that paper is that IL-10 (and anti-inflammatory cytokine) is significantly increased by FMT from an age-matched (older) mouse. In another paper, germ free mice receiving an FMT from older animals showed greater hippocampal neurogenesis and intestinal growth than germ free mice that received FMTs from younger animals (<https://pubmed.ncbi.nlm.nih.gov/31723038/>). These ambiguities are discussed in the following commentary (<https://www.nature.com/articles/s43587-022-00294-w>).

lines 86-87: Related to the point above, it may be important to separate the aging process from chronic disease and frailty. These things are all correlated, to be sure, but it's possible to find chronologically older individuals that are much healthier than their peers. These older individuals may show signs of aging, but some of these signatures may be compensatory and promote resilience, while other signatures may promote frailty. Some discussion of how there may be signatures of healthy and unhealthy aging would be useful here. Similar ideas are presented in terms of 'biological age' (e.g., <https://pubmed.ncbi.nlm.nih.gov/31724055/>).

lines 185-186: It looks like you're using a 'bag of genes' approach to modeling the microbiota and the host. This is probably prudent when using de novo models that haven't been curated (i.e., any inaccuracies within a given model are masked by averaging across everything). However, there are emerging methods for efficiently building multi-compartmental community-scale models (<https://journals.asm.org/doi/10.1128/msystems.01270-22>). Do you think you would gain additional insights if each taxon were compartmentalized?

Figure 3: I was surprised to see that ATP is one of the metabolites produced by the microbiota that is taken up by the host. Why would microbes export so much ATP? I see there are some experimental studies of how microbially-derived ATP can impact immune function, but I wonder if this is largely due to cell lysis or due to active transport of ATP out of the cell? Do you think that this ATP transport may be over-estimated due to the 'bag of genes' approach used here? Should this export be confined to specific taxa? Would that impact the global result?

Also in Figure 3, you show that the brain interaction scores are no different from random, but then you go into analyzing these results anyway. Why?

Figure 4 and S4: Not sure how model growth rates were obtained based on the methods. Were these just fixed based on the relative abundances? I'm not totally sure how to interpret these trends if it's the median relative abundance that's decreasing with

age. And there appears to be a contradiction between growth rates and the fraction of 'ruderal' species. Ruderals (fast growers) are increasing as a proportion of the community with age, but the median community growth rate is declining?

lines 252-255: It's interesting to note that in humans, the maintenance of Bacteroidota in old age is associated with poorer health, while healthier older individuals are relatively enriched in Bacillota (see your Wilmanski ref).

line 269: In your community FBA model, it was unclear how you dealt with individual growth rates. Reference 43 appears to be the wrong citation for this method (<https://pubmed.ncbi.nlm.nih.gov/30273559/>). From the methods section, it looks you constrain the individual growth rates to be equivalent to the relative abundances, and then optimize for community growth? Or do you allow for growth rates to emerge from the objective function somehow? If you're constraining the rates using abundances, what does it mean to compare these 'rates' to the PTRs? I suppose the assumption is that abundance is equivalent to rate in an exponentially-growing system? Maybe add more discussion about these points in the text.

lines 447-450: It looks like you didn't implement any objective functions for the microbiota and the host in these meta-models, correct? These analyses are all based on flux variability analysis?

454-457: Do you think it's circular to say that the host tended to rely upon central metabolic reactions? Aren't these metabolisms, by definition, best-captured by these models? Or do you think this result is surprising and not something you'd expect from randomly sampling the models?

lines 729-732: Were bile acids and mucins added to the diet before it was used to constrain the microbiota models?

line 809: Do you mean 'constraining' instead of 'draining'? Were the taxon biomasses fixed based on data?

lines 834-843: Why model pair-wise? What if pairwise interactions are influenced by the community context? It should be possible to knock individual taxa out from a model, one at a time, and see how this impacts the growth of all the other taxa. This would provide community-specific interaction effects for each pair of taxa. A bit more complex, because the same pair may vary in their interaction across communities. But it may also be more realistic.

line 881: Why have different FDR cutoffs? It makes it a bit harder to compare the number of hits across analyses, to see how strong the global signals are.

line 1159: So was FVA the only analysis run on the meta-models? Did you not try to implement objective functions?

Reviewer #3 (Remarks to the Author):

The manuscript by Best L, et al., presents an extensive analysis of mice microbiomes in relation to ageing, combining multi'omics data. The manuscript is well-organized and written, and the flow is easy to follow. The main concern is about the use of MAGs (although derived from hybrid assembly) as a proxy for both taxonomic and functional profiling of the microbiome in mice, instead of directly using (and comparing with the profiles from MAGs) tools to directly taxonomic and functional profile metagenomes as whole. Indeed, MAGs are usually reconstructed from species for which enough sequenced data is available, so might miss low-abundant species that can still play key roles.

* "reduction in metabolic activity within the aging microbiome [...] attributed to reduced beneficial interactions in the microbiome"  Could these be affected by the compositional change rather than the reduced interactions? Or are the reduced interactions a consequence of a different composition?

* "These pathways could serve as future targets for the development of microbiome-based therapies against aging"  "against" might be a strong concluding word, considering rephrasing.

* Why not use other tools, either k-mer or marker-based, for the taxonomic profiling? These could give a broader view of the microbiome than using only MAGs.

* Lines 104-105: "Notably, we used more stringent cutoffs ($\geq 80\%$ completeness and $\leq 10\%$ contamination)"  While this might be true for completeness, the usual threshold is at 50%, contamination is usually capped at 5%, so 10% might allow for more external DNA sequences that can hinder taxonomic assignment and phylogenetic reconstruction.

* As per my comment above on the taxonomic assignment, functional profiling could have been done directly from the metagenomes and not using MAGs as a proxy. Why this was not done? The authors could also compare functional profiles using both methods to improve the reliability of the results.

* The whole "Metabolic metaorganism models reveal widespread metabolic interactions between host and microbiota" paragraph is well described, although only based on modelling and enrichment analysis. It would be very nice to have experimental validation to corroborate some results.

Version 1:

Reviewer comments:

Reviewer #1

(Remarks to the Author)

I do appreciate the author's effort to thoroughly reply to my previously raised concerns through clarifications in the text, additional analysis, and the inclusion of new data.

Although, an experimental validation of a specific example of the model predictions of metabolic interactions (e.g. measurement of metabolite levels) is still not provided, the authors provide additional analyses to consolidate the validity of their computational models. In order to help the reader to better put these additional analyses into their biological context, I suggest the following additions to the manuscript:

1) The authors mention that they found literature evidence for 42 of the 83 predicted interactions. To better evaluate experimental nature (and quality) of these interactions, I propose that the authors add a supplementary table, in which each of the 42 previously reported interactions is linked to the primary literature together with a short description of its underlying experimental evidence.

2) For the prediction of the microbiome dependence of specific serum metabolites, I suggest to list the metabolites together with their microbiome dependency and their measured serum variance in a supplementary table.

3) I suggest using 'metabolic features' instead of 'metabolic clusters' also in Fig. 4F (and not only in the text).

Reviewer #2

(Remarks to the Author)

The authors have done a great job addressing my comments. I have no further concerns. Congrats on the nice paper.

Reviewer #3

(Remarks to the Author)

The authors addressed the raised concerns and there are no further requests from this reviewer.

Decision Letter:

Our ref: NMICROBIOL-24041239A

12th December 2024

Dear Dr. Kaleta,

Thank you for submitting your revised manuscript "Metabolic modeling reveals the aging-associated decline of host-microbiome metabolic interactions in mice" (NMICROBIOL-24041239A). It has now been seen by the original referees and their comments are below. The reviewers find that the paper has improved in revision, and therefore we'll be happy to publish it, in principle, in Nature Microbiology, pending minor revisions to satisfy the referees' final requests and to comply with our editorial and formatting guidelines.

In addition to the adding the supplementary information, as asked by R1, we'd also like you to add to the discussion why only male mice were used in the study - something we found peculiar during our editorial assessment.

Thank you again for your interest in Nature Microbiology Please do not hesitate to contact me if you have any questions.

Sincerely,

Reviewer #1 (Remarks to the Author):

I do appreciate the author's effort to thoroughly reply to my previously raised concerns through clarifications in the text, additional analysis, and the inclusion of new data.

Although, an experimental validation of a specific example of the model predictions of metabolic interactions (e.g. measurement of metabolite levels) is still not provided, the authors provide additional analyses to consolidate the validity of their computational models. In order to help the reader to better put these additional analyses into their biological context, I suggest the following additions to the manuscript:

- 1) The authors mention that they found literature evidence for 42 of the 83 predicted interactions. To better evaluate experimental nature (and quality) of these interactions, I propose that the authors add a supplementary table, in which each of the 42 previously reported interactions is linked to the primary literature together with a short description of its underlying experimental evidence.
- 2) For the prediction of the microbiome dependence of specific serum metabolites, I suggest to list the metabolites together with their microbiome dependency and their measured serum variance in a supplementary table.
- 3) I suggest using 'metabolic features' instead of 'metabolic clusters' also in Fig. 4F (and not only in the text).

Reviewer #2 (Remarks to the Author):

The authors have done a great job addressing my comments. I have no further concerns. Congrats on the nice paper.

Reviewer #3 (Remarks to the Author):

The authors addressed the raised concerns and there are no further requests from this reviewer.

Version 2:

Decision Letter:

14th February 2025

Dear Professor Kaleta,

I am pleased to accept your Article "Metabolic modeling reveals the aging-associated decline of host-microbiome metabolic interactions in mice" for publication in Nature Microbiology. Thank you for having chosen to submit your work to us and many congratulations.

You may wish to make your media relations office aware of your accepted publication, in case they consider it appropriate to organize some internal or external publicity. Once your paper has been scheduled you will receive an email confirming the publication details. This is normally 3-4 working days in advance of publication. If you need additional notice of the date and time of publication, please let the production team know when you receive the proof of your article to ensure there is sufficient time to coordinate. Further information on our embargo policies can be found here:

<https://www.nature.com/authors/policies/embargo.html>

After the grant of rights is completed, you will receive a link to your electronic proof via email with a request to make any corrections within 48 hours. If, when you receive your proof, you cannot meet this deadline, please inform us at rjsproduction@springernature.com immediately. You will not receive your proofs until the publishing agreement has been received through our system.

Authors may need to take specific actions to achieve [compliance](https://www.springernature.com/gp/open-research/funding/policy-compliance-faqs) with funder and institutional open access mandates. If your research is supported by a funder that requires immediate open access (e.g. according to [Plan S principles](https://www.springernature.com/gp/open-research/plan-s-compliance)) then you should select the

gold OA route, and we will direct you to the compliant route where possible. For authors selecting the subscription publication route, the journal's standard licensing terms will need to be accepted, including [self-archiving policies](https://www.nature.com/nature-portfolio/editorial-policies/self-archiving-and-license-to-publish). Those licensing terms will supersede any other terms that the author or any third party may assert apply to any version of the manuscript.

Have a great weekend! :)

P.S. Click on the following link if you would like to recommend Nature Microbiology to your librarian <http://www.nature.com/subscriptions/recommend.html#forms>

** Visit the Springer Nature Editorial and Publishing website at [www.springernature.com/editorial-and-publishing-jobs](http://editorial-jobs.springernature.com?utm_source=ejP_NMicro_email&utm_medium=ejP_NMicro_email&utm_campaign=ejP_NMicro) for more information about our career opportunities. If you have any questions please click [here](mailto:editorial.publishing.jobs@springernature.com).

Responses to Reviewer Comments (our own response in *blue in italics*):

Reviewer #1 (Remarks to the Author):

In this study, Best et al. investigated metabolic microbiota-host interactions in aging employing multi-omics measurements (i.e., metagenomics, transcriptomics and metabolomics) of male mice at different age. Through various types of association and enrichment analysis, as well as different flavors of genome-scale metabolic modeling the authors correlate microbiome composition (and putative functions) to differences in gene expression in host tissues, namely liver and brain. The presented findings remain rather generic and descriptive based on the performed associations.

1) The main conclusions of this work are poorly supported by the shown data. As experimental validations of the main claims are lacking, the study remains descriptive relying on associations. While human-centered studies are often based on association due to limitations of sample access and experimental restrictions, studies focusing on mouse models should have the ambition to experimentally validate key claims made. The paper is not written in a very clear language that would help the reader to easily follow the logic of the study and the performed analysis. In several instances, the applied experiments and analytical procedures lack sufficient details to evaluate their suitability to derive the drawn conclusions. Also, certain assumptions made by the authors seem poorly justified. It seems that the authors re-mine the same data multiple times with different, but similar approaches, which are not clearly motivated and explained. For example, it remains unclear what the differences between the meta-organism flux balance analysis and community flux balance analysis are.

We thank the reviewer for their constructive feedback. We acknowledge that although the two other reviewers found the manuscript clear and easy to read (e.g. comment #1 by reviewer 2 and comment #1 by reviewer 3), there is still room for improvement. We accordingly improved wording in the manuscript and clarified our assumptions. The other points above are responded to in the more specific comments below, including additional description of our underlying assumptions and validity of results. Concerning model validation, please see our detailed response to comment #3 of reviewer 2. Concerning the difference between meta-model-based analysis and community flux balance analysis, we employ both approaches since they allow us to obtain complementary insights into the function of the microbiome. While the meta-organism model allows to predict interactions between host and microbiota, it contains only a simplified model of the microbiome basically comprising all of the reactions detected in at least one microbial species from the reconstructed metagenome-assembled genomes (also referred to as bag-of-genes approach). As we discuss in response #5 to reviewer 2, this is necessary for computational reasons. However, this approach comes with the down-side that it does not allow for an inference of metabolic activities in individual species and interactions between microbial species. Thus, we also employ community flux balance analysis which explicitly models the individual species of the microbiota. We have added the following statement to the manuscript (l. 330 - 334):

“In contrast to the metaorganism modeling approach used in the previous section, which does not differentiate between individual microbial species due to computational limitations,

community FBA models each microbial species individually. This approach does not explicitly account for the host but allows for the inference of interactions between species within the microbial community.”

2) Although the presented work is of interest to the field of microbiome research it seems preliminary in its current presentation. Rather than providing novel biological insights into metabolic microbiome-host interactions, the work's main strength seems to consist of the data provided and the application and combination of computational methods to mine the data.

We thank the reviewer for this positive assessment of our work.

Specific comments:

3) It remains unclear how the abundance of a metabolic pathway (based on compositional metagenomics data) provides any information about the metabolic activity of a given microbial community. Several of the performed analyses are based on this assumption, leading the authors to claim that the metabolic activity of the aging microbiome is reduced.

Especially in work using metagenomic data, the abundance of pathways obtained from mapping of metagenomic reads against genes contained within pathways is used as a proxy for metabolic activity as we do in the section “Microbiome functions correlate widely with host transcripts across tissues”. Contrary to the reviewer’s statement, we did not use such compositional information to conclude about reduced metabolic activity of the microbiome with aging since that observation was obtained from microbial community modeling and CoPTR estimates as we already describe in the main manuscript (l. 328 - 344).

4) The authors employ flux-balance analysis for several of their analyses using bacterial growth as objective function. Generally, in the gut microbiome and particularly in the large intestine (from which metagenomic samples were collected) it remains unclear why this assumption is justified. The applied method to estimate bacterial growth rates from metagenomic data is controversially discussed in the field, as the model is based on *E. coli* and little experimental evidence exists that the same principle applies to gut bacterial taxa.

We thank the reviewer for this comment. Due to peristalsis and digestion there is a constant outflow of bacterial biomass. Thus, in order to maintain population density, bacteria constantly need to reproduce and hence the optimization of growth is a reasonable objective function for the gut microbiome. To address this point, we have added a corresponding explanation in the methods section (l. 990-993):

“The constant outflow of bacterial biomass via feces requires a constant growth of resident bacterial species to maintain population density and therefore supports optimization of growth as a reasonable assumption for modeling gut microbial communities.”

Concerning the reliability of CoPTR estimates, the reviewer likely refers to a prior work ¹ in which little correlation between PTR-predicted growth rates of bacteria and actually measured growth rates was found. However, CoPTR was not included in that analysis since it was published two years after this benchmark. CoPTR was explicitly tested and validated

using metagenomic assemblies as is the case for our data. We have added a corresponding discussion in the methods section (l. 941-944):

“Please note that while a prior work found little correlation between PTR estimates and experimentally measured growth rates¹⁰⁶, this work did not include CoPTR in the benchmark and CoPTR itself was explicitly validated on metagenome-assembled genomes.”

Also, while there might be limitations with respect to the accuracy of CoPTR estimates, our observation of reduced microbial metabolic activity is actually supported by several additional independent lines of evidence including community-modeling predictions and fecal metabolomic data. Experimental work has shown that increasing transit times in a colon-mimicking in vitro system resulted in reduced microbial growth rates and strongly reduced microbial efficiency to produce biomass² which are exactly our observations along with the prior observation of increased colonic transit times as part of the aging process. We now also discuss this work in the context of our observations of reduced microbial metabolic activity with aging (l. 583-585):

“Interestingly, reduced microbial growth and a decreased capacity to turn nutrients into biomass have previously also been observed as a direct effect of increased transit times in a bioreactor setup mimicking the colon⁷².”

5) The authors suggest that the microbiome provides pyrimidines and ethanol to the brain of the host. This is an interesting finding. In particular ethanol production by gut microbes and its delivery to the brain, circumventing first-pass metabolism by the intestinal epithelium and the liver, warrants experimental validation.

Around 90%-98% of microbially-produced ethanol is metabolized in the liver in humans, thus part of microbially-produced ethanol can still reach the systemic circulation and therefore is also available to the brain³. Indeed, endogenously produced ethanol can typically also be detected in blood samples of healthy volunteers⁴. While the actual amounts might be small, our model predicts whether the uptake of a metabolite is dependent on the microbiome, not the absolute amounts. Thus, since the microbiome is the only source of ethanol in mice, some uptake into the brain is highly likely though the amounts might be relatively small.

6) Line 220. The statement that metamodel results are consistent with correlation analyses might be a circular argument. As the same data was used for both types of analysis, it is not surprising that the results are consistent.

We thank the reviewer for this comment. Indeed we used the same data basis for determining correlations and reconstructing the metamodel. We have repeated the analysis with an unpublished data set from a second independent cohort of mice for which we have tissue transcriptomic and metagenomic data available (n=83 mice). We used data from this cohort to determine host-microbiome correlations for colon and performed the same comparison based on the predicted host-microbiome-interactions from the metamodels reconstructed in the present paper. Also for this independent data set, we observed a strong enrichment of predicted host-microbiome-interactions among host-microbiome correlations:

However, as this independent data set is associated with another manuscript we are currently preparing, and we believe its inclusion would only marginally contribute to the present manuscript, we have decided not to include it. However, instead we now acknowledge that the enrichment of predicted host-microbiome-interactions among host-microbiome correlations could be biased due to a data overlap. We have reworded and amended the corresponding paragraph accordingly in the results and discussion section (l. 244 - 252 and l. 546 - 549):

“We used these interaction matrices to investigate to which extent correlated host gene-microbiome reaction pairs (cf. Fig. 2) also showed a higher frequency of model-predicted interactions compared to randomly sampled pairs. While correlated host gene-microbiome reaction pairs had significantly higher model-predicted interaction frequencies than randomly sampled pairs for colon and liver, the interaction frequencies of randomly sampled pairs were slightly higher for the brain (Fig. 3C; see Methods). These findings suggest a coupling of host metabolic transcription and microbiome metabolic functionality, even though this might be biased by the utilization of the same data basis for determining host-microbiome correlations and reconstructing the metamodel.”

“Moreover, we found that host genes that were correlated with microbiome functions were highly enriched for model-predicted interactions between the host and the microbiota, although this could be biased by the use of the same dataset for both determining host-microbiome correlations and reconstructing the metamodel.”

7) It is not clear whether and how the performed flux analyses contribute to the findings or whether generic correlations of metabolic pathways/GO terms would result in the same associations.

The constraint-based modeling analyses are of central importance for our work since they allow us to infer metabolic interactions within the microbiota and between host as well as microbiome, which is not possible by merely looking at the functional content of the metagenomic data that we have obtained (cf. response #2 to reviewer 3). Moreover, as we also discuss in response #2 to reviewer 3, the inclusion of modeling-based information on metabolic network structure considerably increases the number of associations between host transcripts and microbiome metabolic functions.

8) It remains unclear whether and how the genome-scale metabolic models were gap-filled.

As we describe in the methods section, we used standard settings for gapseq which includes a gap-filling step. For clarification we now explicitly mention that models were gap filled in the methods (l. 888). Additionally, Supplementary table S1.2 holds the information of how many reactions were filled in during the gap-filling step for each model (column “# Gaps”). For further details on which reactions were gap-filled, we recommend to evaluate the individual models that can be obtained from the supplementary models file “obj_metamouse-2023-05-10.rds” available from Zenodo (<http://doi.org/10.5281/zenodo.10844503>). Within each model the individual reactions can be checked for their origin within the “react_attr” table. The column “gs.origin” holds an integer number between 0 and 10, where the codes 1-4 correspond to the four gap-filling steps described in the original gapseq publication and 9 refers to reactions that were added for pathway completions ⁵.

9) The terminology of ‘metabolites identified’ (e.g., line 193) used throughout the manuscript is not precise. The authors seem to refer to exchange reactions of the model rather than actual metabolite identification (e.g., metabolomics).

We corrected this statement to (additions in red, l. 219-220):

“Among the metabolites predicted ~~identified~~ **to be exchanged between the gut microbiome and in the colon**”

10) Text in Fig. 2 is not readable.

We apologize for the poor readability of the text included in Figure 2. We have now increased the font size of that figure and additionally uploaded all our main figures in high resolution or as scalable vector graphics to our Zenodo record (doi.org/10.5281/zenodo.10844502).

11) Fig. 3. Subsystem enrichment. Was FDR correction performed?

P-values were corrected for multiple testing using FDR control. We added a corresponding statement to the figure legend.

12) Line 278. It is unclear what is meant by ‘metabolic clusters’

We have replaced this phrase with “metabolomic features” throughout the manuscript.

13) Line 337. The authors focus their analysis on succinate (based on predictions, not measurements), but it remains unclear why succinate was selected among all the metabolites shown in fig. 4C.

Succinate is one of three metabolites predicted to be increasingly produced by microbiota of old mice (Fig. 4C). Furthermore, succinate metabolizing pathways pop up throughout all three analyzed organs in our host microbiome correlation analysis (see suppl. Tables S2.1 - S2.3) and with bacterial oxalate and itaconate degradation we found two very strong age and host associated pathways that catalyze the succinate to succinyl-CoA interconversion (Fig. 6C). Additionally, succinate has a documented pro-aging and pro-inflammatory role ^{6,7}. Accordingly, we added the following statement in the results section (l. 420 - 432):

“Because we found an aging-associated increase in microbial production of the pro-inflammatory microbial metabolite succinate^{43,49} (Fig. 4C), we further explored how host genes associated with microbial succinate metabolism were regulated during aging. In the colon, we found bacterial interconversion of oxalate to succinate in the oxalate degradation pathway negatively associated with age and positively with host DNA repair, cell division and cell adhesion (Fig. 6C, Supplementary Table S2.1). Moreover, a bacterial succinate to succinyl-CoA interconversion reaction of the itaconate degradation pathway was positively associated with host age and negatively associated with host side protein processing (catabolism, phosphorylation, ubiquitination) as well as telomere maintenance and DNA damage response (Fig. 6C, Supplemental Table S2.1). This indicates that besides a pro-inflammatory role, aging-associated changes in microbial succinate production could potentially contribute to a reduced DNA damage response as one of the hallmarks of aging^{2,50}.”

And the following text to the discussion section (l. 577 - 580):

“In contrast, metabolic modeling predicted increased production of the pro-inflammatory metabolite succinate^{66,67}, a known indicator for a dysbiotic gut environment⁴⁹, which we found to be associated with key processes deregulated in aging on the host side including DNA damage responses⁵⁰ and protein homeostasis⁶⁸.”

14) Line 451-455. The authors state that the host most often depended on central metabolic microbiome reactions. Would this observation not be biased by the coverage of reactions and/or the fact that microbiome gene annotation and genome-scale metabolic model construction are best for these reactions?

There might indeed be a bias due to the better coverage of central metabolic reactions by the reconstructed models. However, there are many other microbial pathways that in principle should have a similar high coverage such as amino acid or nucleotide metabolic pathways but we do not see a similar high frequency of host-microbiome-interactions involving these pathways. To address this point, we have added a corresponding statement to our discussion (l. 558-561):

“However, this emphasis on interactions within central metabolic pathways may also be biased by the more extensive understanding of central metabolic reactions compared to other, less well-characterized areas of bacterial metabolism represented in the reconstructed models.”

Reviewer #2 (Remarks to the Author):

Summary -

1) The authors present an interesting exploration of host-microbial metabolic interplay in a dataset of C57BL/6JUKj male mice at different ages (2, 8, 15, 24, and 30 months). The data set includes transcriptomic data from brain, liver, and colon tissue, along with fecal metagenomic data, for each mouse. Much of the work presented in the manuscript is centered around genome-scale metabolic modeling and flux balance analysis of both the microbiota and the host, to explore how the microbiota may contribute to metabolic changes that occur with aging. This is an ambitious and complex analysis, and the authors have done an impressive job navigating many of the potential pitfalls. The paper is well written. Many of the results are intuitive and match with the literature, and there are several newer insights as well. Overall, I commend the authors on their work. However, I have a few concerns about methods and framing and a few suggestions, which I outline below.

We thank the reviewer for this positive assessment of our work.

Specific Comments -

2) It's always difficult to validate these types of models, but some additional follow-up validation beyond comparison to the literature would up the impact of the work.

We acknowledge that ideally predictions from our model should be experimentally validated. To address this point which was also raised by the other reviewers, we have conducted two additional analyses. First, we have scrutinized the literature in detail for support of predicted metabolic interactions between the different organs and the microbiome. For 42 of the 83 predicted interactions (51%) displayed in Fig. 3B, we could find literature evidence supporting them. We added a corresponding statement to the main manuscript (l . 230 - 235) and added Supplementary Table S3.9 documenting the literature support:

“Overall, we found that among the predicted interactions displayed in Fig. 3B, 42 (51%) were already supported by prior experimental evidence across all three organs (Supplementary Table S3.9), thereby strongly supporting the ability of the metamodel to capture metabolic microbiome-host-interactions. This evidence encompassed both a direct assessment of metabolites exchanged between host and microbiota as well as systematic analyses using labeled compounds exclusively metabolized by the microbiome.”

We think that while it would in principle be possible to validate further interactions predicted by the metamodel, the corresponding experiments would be very laborious and would only add to a limited extent to the manuscript since a considerable portion of the predicted interactions already have experimental support. Moreover, it would require us to focus on one particular interaction and therefore a major reorganization of the content of our manuscript away from the more global perspective we are taking right now.

Second, we investigated to which extent metamodel-predicted microbiome-dependent reactions responded to microbial colonization. To this end, we generated transcriptomic data from five tissues of germfree mice, germfree mice conventionalized with fecal material from

wild-type mice and conventionally raised mice. We used this data to determine genes that responded to microbial colonization in two comparisons: germfree vs. conventionally raised mice and germfree vs. conventionalized mice. Thus, we obtained a list of genes commonly regulated in response to microbial colonization in at least three tissues. This yielded a total of twelve comparisons: five organs plus shared differentially regulated genes in the comparison of germfree vs. conventionally raised and germfree vs. conventionalized mice. Across these twelve comparisons, we found that mostly up-regulated ($n=7$ comparisons) but to some extent also down-regulated genes ($n=3$ comparisons) were associated with reactions with significantly higher microbiome dependency than unregulated genes. Reassuringly, we found only two cases in which unregulated genes showed a significantly higher microbiome dependency than regulated genes in their associated reactions. Thus, reactions associated to genes regulated in response to microbial colonization show a higher model-predicted microbiome dependency than reactions associated with unregulated genes, thereby strongly supporting the ability of the metamodel to predict host reactions dependent on the microbiome. Also we found that up-regulated genes in response to microbial colonization were more frequently associated with reactions with higher microbiome dependency than down-regulated genes ($n=6$ vs. $n=3$ comparisons). In order to exclude that these significant differences were caused by the mapping of genes to the reactions with which they are associated, we repeated the analysis 1000 times after randomizing gene labels. Compared to the 21 significant differences we found in the original analysis, In the randomization runs, we identified a maximum of 18 significant differences between up-regulated genes, down-regulated genes and un-regulated genes (on average 8.6 significant differences). We present these additional analyses in the results (l. 253 - 278, including Fig. 3D):

“To further validate the metamodel's ability to identify microbiome-host interactions, we examined how microbial colonization affected predicted microbiome-dependent host reactions. To this end, we used information on genes responsive to microbial colonization from our germ-free mice cohort. In addition to colon, liver and brain we also assessed gonadal white adipose tissue and quadriceps to determine whether predicted microbiome dependency in modeled tissues could inform about dependency in other tissues. For each tissue and comparison, we identified up-regulated, down-regulated, and unregulated genes, translating them into corresponding reactions, and then assessed the model-predicted microbiome dependency of those reactions. Across all comparisons, we found that up-regulated reactions showed significantly higher microbiome dependency than unregulated reactions in seven instances, while down-regulated reactions showed higher dependency in three instances (Fig. 3D). Reassuringly, only two cases showed lower microbiome dependency in up- or down-regulated reactions compared to unregulated ones, thereby strongly supporting the ability of the metamodel to predict microbiome-dependent host reactions. Overall, microbial colonization was more often associated with higher microbiome dependency in up-regulated than in down-regulated reactions (six vs. three cases). These findings demonstrate a strong association between metamodel-predicted microbiome dependency and host responses to microbial colonization. To exclude that these results were due to the mapping of differentially expressed genes to reactions, we repeated the analysis with gene labels randomized and did not find a single case with a higher number of significant associations across 1000 randomized repetitions (Supplementary Fig. 3A). As additional validation, we used the metamodel to predict microbiome dependence of specific serum metabolites and compared these predictions with previously published data on the

variance in serum metabolite concentrations explained by microbiome composition in humans⁴⁰. We observed a strong positive correlation between predicted microbiome dependence and the explained variance (Spearman's $\rho=0.43$, $p=1.5\times 10^{-3}$, Supplementary Fig. 3B).”

and the discussion section of our manuscript (l. 540 - 546):

“Overall, 51% of the predicted interactions with highest confidence were already reported previously in the literature. As an additional validation, we observed that reactions linked to genes regulated in response to microbial colonization across various tissues exhibited a significantly higher average microbiome dependence compared to reactions linked to unregulated genes. In general, microbial colonization was more frequently associated with an up-regulation of microbiome-dependent reactions.”

Fig. 3D: Microbiome dependency of reactions whose associated genes are regulated in response to microbial colonization. The y-axis indicates sets of genes that are up- or down-regulated in the corresponding tissues and contrasts, while the x-axis indicates the microbiome dependency of the corresponding reactions. The “Shared” label indicates genes that are regulated in at least three tissues. FDR-corrected p-values of Dunn’s tests between

groups following a group-level Kruskal-Wallis-test are displayed next to the bar plots of means with error bars representing the standard deviation. Only comparisons with a Kruskal-Wallis-test p -value <0.05 for each group are shown. *, $p<0.05$, **, $p<0.01$; ***, $p<0.001$.

As a second validation approach, we used the metamodel to predict microbiome dependence of serum metabolites. To this end, we tested for each metabolite exchanged via the bloodstream to which extent its production depended on the microbiome. For each metamodel of each mouse, we defined a serum metabolite as microbiome dependent if its production capacity was reduced by at least 50% if the microbiome was removed. Across all 52 mice, we thereby determined the microbiome dependency of each serum metabolite. Subsequently, we matched those metabolites to data from a human cohort⁸ in which it was tested how much variation in serum metabolite concentration could be explained by microbiome composition in a human cohort. We found a strong positive correlation between predicted microbiome dependency and explained variance. We thus have added the following text to the main manuscript (l. 273 - 278):

“As additional validation, we used the metamodel to predict microbiome dependence of specific serum metabolites and compared these predictions with previously published data on the variance in serum metabolite concentrations explained by microbiome composition in humans⁴⁰. We observed a strong positive correlation between predicted microbiome dependence and the explained variance (Spearman’s $\rho=0.43$, $p=1.5\times 10^{-3}$, Supplementary Fig. 3B).”

and the following figure as Supplementary Figure 3B:

Suppl. Fig. 3B: Association between model-predicted microbiome dependency of metabolites in blood and explained variation of serum concentrations of the corresponding metabolite by microbiome composition in a human cohort. Each dot corresponds to a metabolite. The association between microbiome dependency and explained variance is significant using Spearman’s correlation ($\rho=0.43$, p -value= 1.5×10^{-3}).

Furthermore, we used the germ-free mouse cohort to validate the microbiome association of host genes. We determined the overlap of microbiome-associated transcripts (Fig. 2A-F) with the differentially expressed transcripts from our germ-free validation cohort in both

treatment contrasts (GF. vs. CONVR and GF vs. CONVD). The number of shared microbiome-associated transcripts between both cohorts was statistically evaluated using hypergeometric overrepresentation tests stratified by organ and treatment contrast, correcting for multiple testing via Benjamini and Hochberg's method. The numbers of shared transcripts are plotted for each possible combination in Fig. 2G.

We found a highly significant overlap between microbiome-responsive genes in colon and liver, but not in brain. However, the lack of significant overlap with microbiome-responsive genes in the brain might be partially due to the relatively small number of microbiome-responsive genes that we identified for this tissue from the germ-free cohort (cf. Supplementary Table S2.12).

We added the following paragraph and figure panel to the respective results section (l. 173 - 187, and Fig. 2G):

“In order to validate host genes associated with microbiome functions that we have identified, we determined the extent to which those genes overlapped with genes responsive to microbial colonization. To this end, we generated transcriptomic data from five tissues (colon, liver, brain, gonadal white adipose tissue and quadriceps) of three groups of mice: conventionally raised wild-type mice (CONVR), germ-free mice (GF), and mice conventionalized with fecal material from WT mice (CONVD; n=8 per group). For each tissue and comparison, we identified genes responsive to microbial colonization (comparison CONVR vs. GF and CONVD vs. GF; Supplementary Tables S2.8 - S2.12) and compared those to genes for which we have identified an association with microbiome functions. We found a highly significant overlap between microbiome-responsive genes in colon and liver, but not in brain (Fig. 2G). The lack of significant overlap with microbiome-responsive genes in the brain might be partially due to the relatively small number of microbiome-responsive genes that we identified for this tissue (Supplementary Table S2.12). This analysis therefore independently supports the association of the identified genes with microbiome function and precludes that those associations might be driven by potential indirect associations.”

Figure 2G: Validation of microbiome association of host genes. Genes that were found microbiome associated via partial correlations (green) (c.f. A-F) compared to sets of genes

responsive to microbial colonization across tissues via DESeq2 (orange). Multiple testing corrected p-values from hypergeometric tests for observing a bigger or equally large overlap (purple) are plotted above each bar respectively.

3) lines 51-54: The effects of FMT from young mice into older mice appear to be more complicated than what's presented here. The literature is a bit split on the topic. For example, in the Boehme paper cited by the authors, they state that it shows evidence that FMTs from young mice reduce inflammation and improve neurological health in older mice. However, one of the more striking results in that paper is that IL-10 (an anti-inflammatory cytokine) is significantly increased by FMT from an age-matched (older) mouse. In another paper, germ free mice receiving an FMT from older animals showed greater hippocampal neurogenesis and intestinal growth than germ free mice that received FMTs from younger animals (<https://pubmed.ncbi.nlm.nih.gov/31723038/>). These ambiguities are discussed in the following commentary (<https://www.nature.com/articles/s43587-022-00294-w>).

We thank the reviewer for pointing this out. Indeed, there is somewhat of a mixture of effects of fecal microbiota transplants on specific aspects of aging and the role of particular metabolites in the context of aging. On a side note, in Kundu et al., donor mice for the young microbiota transplants were 5-6 weeks of age and thus not yet in adulthood which might be an additional confounder since the microbiome is still somewhat dynamic in this period of life. We now phrase our statement concerning the role of the microbiome in aging more cautiously (l. 54 - 59):

“Microbiome transfer experiments revealed that introducing young microbiota to old hosts extends lifespan ^{10,11} and reverses specific aspects of aging in animal models ¹². However, some works have also shown beneficial effects of aged microbiota ⁹ or signatures specific to healthy aging in centenarians ¹⁰ that indicate that some aging-associated changes in the microbiota might also be compensatory by counteracting aging-associated changes in the host ¹¹.”

4) lines 86-87: Related to the point above, it may be important to separate the aging process from chronic disease and frailty. These things are all correlated, to be sure, but it's possible to find chronologically older individuals that are much healthier than their peers. These older individuals may show signs of aging, but some of these signatures may be compensatory and promote resilience, while other signatures may promote frailty. Some discussion of how there may be signatures of healthy and unhealthy aging would be useful here. Similar ideas are presented in terms of 'biological age' (e.g., <https://pubmed.ncbi.nlm.nih.gov/31724055/>).

We thank the reviewer for raising this point. As we already mention in the discussion, an assessment of more fine grained markers of aging (e.g. frailty, cognitive function) probably would even provide more details about aging-related pathways. Indeed, we are currently working on a follow up manuscript in which we investigate associations between microbiome function and cognitive function in another mouse cohort. On the other hand, as suggested, using biological clocks e.g. epigenetic clocks such as the one that we build previously for a mouse cohort (cf. ⁹) could yield insights into which molecular processes make an individual younger or older than their peers in the same age group. We have added a corresponding statement in the discussion of our work (l. 653 - 655):

“Also previous work has made the distinction between biological and chronological age to indicate individuals that age faster or slower than their peers from the same age group ⁹⁵ which could be assessed using epigenetic clocks ⁹⁶.”

5) lines 185-186: It looks like you're using a 'bag of genes' approach to modeling the microbiota and the host. This is probably prudent when using de novo models that haven't been curated (i.e., any inaccuracies within a given model are masked by averaging across everything). However, there are emerging methods for efficiently building multi-compartmental community-scale models (<https://journals.asm.org/doi/10.1128/msystems.01270-22>). Do you think you would gain additional insights if each taxon were compartmentalized?

We thank the reviewer for this insightful comment. Indeed, within the metamodel the microbial part is modeled by a bag-of-genes approach basically using a single compartment for all bacterial species combined. While this approach comes at the expense that this prevents the direct investigation of within-community interactions, it offers the benefit of a considerably improved computational tractability. Thus, including models of individual species' metabolic models in the metamodel rather than the combined microbiome metabolic model as is being done, for instance, in the human whole-body models ¹⁰ would have prevented the usage of the elementary flux mode sampling approach which took several weeks to complete even on a large compute cluster. In particular, since almost all bacterial species were detected in each individual mouse, this would have resulted in an expansion of the metamodels of each mouse by $181 \times 1500 = 271.500$ reactions which is about a magnitude larger than the typical metamodel of mice that we have reconstructed (around 15.000 reactions per model). Also the extent to which we could have conducted flux variability-based analyses would have severely been hampered. To overcome the limitation that our bag-of-gene approach prevents the study of aging-associated changes of within-community interactions, we have used community flux balance analysis directly on the microbiome metabolic reconstructions. We have added a corresponding discussion to our manuscript:

I. 1349 - 1353: “We decided to use a merged microbiome model instead of species-level metabolic reconstructions to maintain computational tractability of the metamodel for comprehensive downstream analysis (e.g. flux variability analysis and elementary flux mode sampling) which would have been impossible otherwise.”

I. 993 - 998: “Also, our community modeling approach assumes a community-level optimization of community growth rates which might not be realistic in all scenarios as it assumes a coordination of metabolic fluxes between species to maximize overall growth but comes at the advantage of a considerable speed up in computational time compared to approaches that consider only an individual-level optimization of growth rates such as, e.g. individual-based modeling approaches ¹¹¹.”

6) Figure 3: I was surprised to see that ATP is one of the metabolites produced by the microbiota that is taken up by the host. Why would microbes export so much ATP? I see there are some experimental studies of how microbially-derived ATP can impact immune function, but I wonder if this is largely due to cell lysis or due to active transport of APT out of

the cell? Do you think that this ATP transport may be over-estimated due to the 'bag of genes' approach used here? Should this export be confined to specific taxa? Would that impact the global result?

We thank the reviewer for this insightful comment. Indeed ATP is a rather expensive metabolite to export in metabolic terms. On the other hand, it is also a highly abundant metabolite due to its central role as energy currency metabolite. Beyond our discussion of previous work that conclusively demonstrate an uptake of bacterially produced ATP by the host (l. 628 - 632), export of ATP by microbial species has been reported in various contexts, also outside of host contexts (e.g. ¹¹) and ATP has even been described as an interkingdom signaling molecule (cf. ¹²). Additionally, previous studies have demonstrated that the export of ATP is directly dependent on bacterial metabolic activity, thereby ruling out cell lysis as the sole source of extracellular ATP¹³. We have added a corresponding statement to our discussion of host-microbial nucleotide co-metabolism (l. 625 - 628):

“The experimentally documented dependence of microbial nucleotide excretion on microbial cellular metabolism excludes bacterial cell lysis as the sole source of bacterially-produced ATP available to the host ⁸⁶.”

7) Also in Figure 3, you show that the brain interaction scores are no different from random, but then you go into analyzing these results anyway. Why?

While we find that brain interaction scores are no different from random interactions, this is probably not unexpected since, among the three organs we consider, the brain is physiologically speaking “farthest” from the microbiome and protected by the blood brain barrier. On the other hand, analyzing the response to microbial colonization, we find that induced genes have a considerably higher microbiome dependency in their associated reactions compared to unregulated genes. Finally, numerous prior works have documented interactions between microbiota and brain, further warranting the exploration of corresponding predictions by the metamodel.

8) Figure 4 and S4: Not sure how model growth rates were obtained based on the methods. Were these just fixed based on the relative abundances? I'm not totally sure how to interpret these trends if it's the median relative abundance that's decreasing with age. And there appears to be a contradiction between growth rates and the fraction of 'ruderal' species. Ruderals (fast growers) are increasing as a proportion of the community with age, but the median community growth rate is declining?

We thank the reviewer for this comment. Please see also our related response to your comment #15. Community growth rates were predicted as part of the optimization of community growth when using community FBA. We added a corresponding statement to the manuscript to clarify this point (l. 1004 - 1006):

“Additionally, this analysis yielded a predicted community growth rate which, when multiplied with the individual species' abundances, allows to derive the growth rate of each individual bacterial species (Fig. 4D).”

When determining community-level growth rates using CoPTR, we explicitly used median growth rates since weighting them by e.g. abundance would have introduced some artificial

bias that could have resulted in an unspecific correlation with community-FBA-derived community growth rates which also incorporate abundance information. We now also mention this in the methods (l. 938 - 941):

“We did not weight growth rate predictions by individual species’ abundances to obtain a community-level growth rate. Thereby we avoided spurious correlations with community growth rates predicted using community FBA since community FBA explicitly incorporates abundance information.”

We agree that the observation of an increased abundance of ruderal species which are typically characterized by faster growth somewhat contrasts with our inference of reduced microbial growth rates with age. However, besides fast growth, ruderals typically are first colonizers with low catabolic diversity that colonize niches after environmental perturbations. Thus, while they can potentially grow very fast, they also have a considerably reduced potential to interact with other species (as also indicated by our observation of a considerable increase in competitive interactions with age). Thus, since there is typically an intense interaction between bacterial species in the gut, an increased abundance of ruderals can result in a reduced overall community growth due to the less efficient utilization of dietary resources. We have added a corresponding statement to the main text (l. 355 - 357):

“... ruderals, which are first colonizers of niches and poor interaction partners due to reduced catabolic diversity⁴⁴ (Supplementary Fig. 4F).”

Moreover, we have adapted the legend of Supplementary Figure 4B accordingly to provide a more nuanced description of the ruderal strategy.

9) lines 252-255: It's interesting to note that in humans, the maintenance of Bacteroidota in old age is associated with poorer health, while healthier older individuals are relatively enriched in Bacillota (see your Wilmanski ref).

Thank you for this interesting point which we have added to our discussion (l. 569 - 574):

“At the species level, we observed an age-associated increase in species from the phylum Bacteroidota and a decline in those from the phylum Bacillota. These findings align with previous human studies, which have shown that the persistence of Bacteroidota in old age is linked to poorer health outcomes, whereas an enrichment of Bacillota is associated with better health in the elderly⁶⁴.”

10) line 269: In your community FBA model, it was unclear how you dealt with individual growth rates. Reference 43 appears to be the wrong citation for this method (<https://pubmed.ncbi.nlm.nih.gov/30273559/>). From the methods section, it looks you constrain the individual growth rates to be equivalent to the relative abundances, and then optimize for community growth? Or do you allow for growth rates to emerge from the objective function somehow? If you're constraining the rates using abundances, what does it mean to compare these 'rates' to the PTRs? I suppose the assumption is that abundance is equivalent to rate in an exponentially-growing system? Maybe add more discussion about these points in the text.

We thank the reviewer for pointing this out. Indeed the correct citation was Pryor et al., Cell 2019. We have corrected the reference. As the reviewer correctly points out, we construct a community-level biomass that drains the individual species' biomasses according to their abundance based on metagenomic reads. Thus, individual species' biomasses can only vary as part of the community-level biomass reaction flux but the proportion between the growth rates of species of a community remains the same. As we describe in the methods (l. 937-938), we determined median CoPTR of species for each community to calculate a metagenome-derived community growth rate from the CoPTR estimates. Thus, we did not compare species-level growth rates between FBA and CoPTR estimates but rather on the community level. Also, as the reviewer correctly points out, we assume that abundance is equivalent to rate due to the constant replenishment of bacterial biomass in the colon. Please also see our statement to the related comment #4 of reviewer 1. We added a corresponding statement in lines 990 - 993:

“The constant outflow of bacterial biomass via feces requires a constant growth of resident bacterial species to maintain population density and therefore supports optimization of growth as a reasonable assumption for modeling gut microbial communities.”

11) lines 447-450: It looks like you didn't implement any objective functions for the microbiota and the host in these meta-models, correct? These analyses are all based on flux variability analysis?

This assessment is correct. Within the metamodel we did not assume an objective function since especially for the host optimization of growth is unrealistic. Instead we characterized the flux space of the individual metamodels using flux variability analysis and elementary flux mode sampling as the reviewer correctly states. We added a corresponding statement to our manuscript (l. 1422 - 1424):

“Thus, we did not consider any optimization of growth of the metamodel since for most tissues in mammals there is only negligible cellular replication ¹³⁴.”

12) lines 454-457: Do you think it's circular to say that the host tended to rely upon central metabolic reactions? Aren't these metabolisms, by definition, best-captured by these models? Or do you think this result is surprising and not something you'd expect from randomly sampling the models?

Please see the related response to reviewer 1, comment #14. The reviewer is correct that there might be a bias due to the better coverage of central metabolic reactions by the reconstructed models. However, there are many other microbial pathways that in principle should have a similar high coverage such as amino acid or nucleotide metabolic pathways but we do not see a similar high frequency of host-microbiome-interactions involving these pathways. We have added a corresponding statement to our discussion (l. 558-561):

“However, this emphasis on interactions within central metabolic pathways may also be biased by the more extensive understanding of central metabolic reactions compared to other, less well-characterized areas of bacterial metabolism represented in the reconstructed models.”

13) lines 729-732: Were bile acids and mucins added to the diet before it was used to constrain the microbiota models?

We did not consider bile acids and mucins in the community modeling results.

14) line 809: Do you mean 'constraining' instead of 'draining'? Were the taxon biomasses fixed based on data?

We thank the reviewer for this comment. We corrected the wording and now write “A community-level biomass reaction was introduced that consumed the biomass of individual species according to their relative abundance as inferred from metagenomic data.” (l. 982 - 982). Indeed the ratios of growth rates of species are fixed to the community-level biomass reaction according to their abundance in the sample.

15) lines 834-843: Why model pair-wise? What if pairwise interactions are influenced by the community context? It should be possible to knock individual taxa out from a model, one at a time, and see how this impacts the growth of all the other taxa. This would provide community-specific interaction effects for each pair of taxa. A bit more complex, because the same pair may vary in their interaction across communities. But it may also be more realistic.

We thank the reviewer for this excellent suggestion. While it might be computationally infeasible to calculate community simulations for every possible removal, we still were able to follow the suggested approach for a selection of bacterial models.

To this end we choose the 20 MAGs with their abundance most significantly changing during aging (10 MAGs going up with age and 10 MAGs going down with age) for deletion from 50 microbial communities. We implemented this deletion approach by removing only one model (MAG) at a time from each individual mouse's microbiome community and simulated metabolite exchange as well as community growth via FBA (see our newly added methods section “Microbiome community modeling with deletion of single microbes”, l. 1018-1039). The change of growth rate was compared to the original growth of the complete community. Indeed the change of microbiome community growth rates showed a decrease when MAGs associated with young mice are removed, while the community growth increased when removing MAGs that were more abundant in old mice. Thus, MAGs going down with age tend to have a positive impact on community productivity and MAGs that go up with age tend to have a negative impact.

We have included two additional figure panels to represent this new analysis in supplementary figure 4D,E and added an according sentence to our results part (l. 344-348):

“Further, we evaluated the change of FBA-predicted community growth upon removal of single bacterial members, and observed that MAGs suppressed in aging had a beneficial effect on community productivity and growth, while MAGs which were enriched in old mice showed a negative impact (Supplementary Fig. 4D and E).”

16) line 881: Why have different FDR cutoffs? It makes it a bit harder to compare the number of hits across analyses, to see how strong the global signals are.

We filtered the GO terms with different p-value cut-offs since otherwise some tissues would show too many associations to plot them all. We have added a corresponding statement in the figure legend. As indicated, the unfiltered lists of associations between host and microbiota can be found in Supplementary Tables S2.1 - S2.4.

17) line 1159: So was FVA the only analysis run on the meta-models? Did you not try to implement objective functions?

Please see also our related response to your comment #11. We characterized the metamodel using flux variability analysis and elementary flux mode sampling.

Reviewer #3 (Remarks to the Author):

1) The manuscript by Best L, et al., presents an extensive analysis of mice microbiomes in relation to ageing, combining multi-omics data. The manuscript is well-organized and written, and the flow is easy to follow.

We thank the reviewer for this positive assessment of our work.

2) The main concern is about the use of MAGs (although derived from hybrid assembly) as a proxy for both taxonomic and functional profiling of the microbiome in mice, instead of directly using (and comparing with the profiles from MAGs) tools to directly taxonomic and functional profile metagenomes as whole. Indeed, MAGs are usually reconstructed from species for which enough sequenced data is available, so might miss low-abundant species that can still play key roles.

We thank the reviewer for the suggestion of a more direct functional and taxonomic profiling of our metagenomic sequencing data. To accommodate this suggestion, we applied the biobakery3 pipeline ¹⁴ using MetaPhlan3 and HUMAnN3 and included those results in the supplements.

We first excluded a similar analysis from our manuscript, because in our experience these profiling tools' databases are heavily dominated by human microbiota thus missing to some extent the unique features of murine gut microbiomes. One of the main reasons for our extensive characterisation of the murine gut microbiome was actually driven by the rather incomplete nature of some reference databases in regard to murine bacteria. Although this shortcoming is gradually being improved with more recent updates to those profiling databases. Additionally, an important aspect of our study is the reconstruction of metabolic models, which requires mostly complete draft genomes and thus implies the use of metagenomic assemblies.

We agree that indeed rare microbes might not be assembled to draft MAGs surpassing our filtering criteria. However, while there will be reads contained in the metagenomic sequencing data originating from those rare species, due to their low abundance, we would expect little contribution to the overall numbers of functionally and taxonomically predicted features.

A new methods section "Functional and taxonomic profiling of metagenomic reads" was added related to our biobakery3 analysis (l. 1193 - 1208). On the functional level we observed comparable yet much less significant associations, while the taxonomic results appear qualitatively and quantitatively comparable to our MAG-based analysis.

Taxonomically we found 69 species-level abundance changes with host age, while the general trend of reduced Bacillota spec. abundance and increased Bacteroidota spec. abundance that we saw in our MAG based analysis was conserved also in the MetaPhlan4 based analysis (Fig. 4A and Supplementary Fig. 4A). On the functional level (HUMAnN3), in agreement with our MAG-modeling based analysis, we observed a few host-microbiome-associations of the microbial lipid, co-factor and energy metabolism with host colon immunity and neuron maintenance respectively (Fig. 2B and Supplementary Fig. 2B).

We have included two additional figure panels to represent this new analysis in supplementary Figure 2B and 4A and added the following to the respective results sections (l. 165-167):

“By directly inferring functions from quality controlled metagenomic read data (HUMAnN3), we found fewer, yet comparable host-microbiome associations (Supplementary Fig. 2B and Supplementary Table S2.7).”

and (l. 326-327):

“We observed similar associations when directly inferring taxonomic changes from quality-controlled metagenomic reads (Supplementary Fig. 4A and Supplementary Table S4.15).”

We have added the corresponding figure as subpanel B to Supplementary Fig. 2:

Supplementary Fig. 2B: HUMAnN3 predicted microbial functions (MetaCyc) correlated to host colon transcription (GO biological processes) show comparable associations to MAG-based analysis in Fig. 2B. We employed the Biobakery3 pipeline to infer functional and taxonomic associations directly from quality controlled metagenomic reads and compared those results to our MAG based approach. We observed similar, yet fewer associations when correlating host to microbiome functions. Bacterial pyruvate fermentation and co-factor biosynthesis were negatively correlated with host side neuron survival. Microbial fatty acids were found positively associated with colon immune processes (MHC class I) and negatively with transcription regulation. These HUMAnN3 based analysis results were generally in concordance, although much less detailed, to our MAG and metabolic model derived host microbiome correlations (see Fig. 2B).

This figure was added as subpanel A to Supplementary Fig. 4:

Supplementary Fig. 4A: MetaPhlan4 derived species with significant changes in abundance with host age. The MetaPhlan-based approach identified 335 bacteria at the

*species level of which 69 were found differentially abundant with age. The species *Lactobacillus johnsonii* matched the MAG based analysis (Fig. 4A). However, due to different taxonomic databases and methods of annotation the resulting species names might vary. On phylum level the overall aging pattern of both MetaPhlAn and MAG-based taxonomic analysis yielded comparable results, which was a general decrease of *Bacillota* spec. and an increase of *Bacteroidota* spec. with host age.*

3) “reduction in metabolic activity within the aging microbiome [...] attributed to reduced beneficial interactions in the microbiome”  Could these be affected by the compositional change rather than the reduced interactions? Or are the reduced interactions a consequence of a different composition?

We thank the reviewer for pointing this out. Indeed we do not know whether the change in beneficial interactions drives the loss of metabolic activity or the other way around. Hence, we have toned down this statement in the abstract and now write “We observed a pronounced reduction in metabolic activity within the aging microbiome accompanied by reduced beneficial interactions between bacterial species.”

In principle, as we also state at the end of the response to comment #4 by reviewer 1 and also in the manuscript (l. 580 - 585), both changes (loss of interaction, change in composition, loss of metabolic activity) might also be driven by changes in intestinal transit time during aging.

4) “These pathways could serve as future targets for the development of microbiome-based therapies against aging”  “against” might be a strong concluding word, considering rephrasing.

We have adapted the phrase and now write “These pathways could serve as future targets for the development of microbiome-based anti-aging therapies.”

5) Why not use other tools, either k-mer or marker-based, for the taxonomic profiling? These could give a broader view of the microbiome than using only MAGs.

*We thank the reviewer for this comment, we have now added a marker based analysis with *biobkaery3*. Please see our relating response to your comment #2.*

6) Lines 104-105: “Notably, we used more stringent cutoffs ($\geq 80\%$ completeness and $\leq 10\%$ contamination)”  While this might be true for completeness, the usual threshold is at 50%, contamination is usually capped at 5%, so 10% might allow for more external DNA sequences that can hinder taxonomic assignment and phylogenetic reconstruction.

We agree that more loose contamination cut-offs might reduce the accuracy of taxonomic assignments. In our reporting of Medium and High Quality MAG drafts, we referred to the standards and metrics laid out by The Genome Standards Consortium ¹⁵. Accordingly, a MAG will be considered high quality with completion > 90%, contamination < 5 % and if genes for 23S, 16S, and 5S rRNA and at least 18 tRNAs are recovered. We want to emphasize that 133 of our 181 MAGs fulfill these very strict completion and contamination cutoffs but 25 of those 133 could be considered true High Quality MAGs only due to some

missing rRNA or tRNA genes. The lack of those genes however does not impact the quality of our MAG-derived metabolic models since they are derived from the genes contained in the MAGs. In order to increase the number of models and to obtain a more realistic representation of the underlying biology, we decided to also include less complete and slightly more contaminated MAGs rather than lose a lot of relevant sequencing data. The gap-filling step in the model reconstruction added required, but missing, genes in an otherwise complete pathway and thus can partially compensate for the lowered completeness threshold. In detail, only 18 of our MAGs showed a contamination score greater than 5 %, ranging from 5.1 to 9.8 % with a mean of 6.7 % and only two out of those 18 had a completeness score < 90 %. We believe that our relaxed completeness and contamination cutoffs are an agreeable tradeoff, especially when employed in metabolic model reconstruction which includes a gap-filling step and “orphan gene to pathway knowledge filter”. Indeed also other works involving metabolic model reconstruction from MAGs have used similar cutoffs ^{16,17}.

We have updated our methods section to include the following (l. 858 - 871):

“In our reporting of Medium and High Quality MAG drafts, we referred to the standards and metrics laid out by The Genome Standards Consortium ³⁰. Accordingly, a MAG will be considered high quality with completion > 90%, contamination < 5 % and if genes for 23S, 16S, and 5S rRNA and at least 18 tRNAs are recovered. While 133 of our 181 MAGs fulfill these very strict completion and contamination cutoffs, only 25 of those 133 could be considered true High Quality MAGs only due to some missing rRNA or tRNA genes. The lack of those genes however does not impact the quality of our MAG-derived metabolic models. Only 18 of our MAGs showed a contamination score greater than 5 %, ranging from 5.1 to 9.8 % with a mean of 6.7 % and only two out of those 18 had a completeness score < 90 % (see Supplementary Table S1.2). We employed slightly less strict cutoffs for contamination and completeness in order to include a larger variety of MAGs in our study. While the metabolic model construction from MAGs can partially compensate for lack of completeness via gap-filling and for contamination by pathway-completeness checks, our more loose contamination cut-offs might reduce the accuracy of taxonomic assignments.”

7) As per my comment above on the taxonomic assignment, functional profiling could have been done directly from the metagenomes and not using MAGs as a proxy. Why this was not done? The authors could also compare functional profiles using both methods to improve the reliability of the results.

Please see our related response to your comment #2. We now also included a direct functional profiling of the microbiome and found that our MAG-/model-based approach provides a much better signal in terms of aging-regulated microbial species and significant host-microbiome-correlations.

8) The whole “Metabolic metaorganism models reveal widespread metabolic interactions between host and microbiota” paragraph is well described, although only based on modeling and enrichment analysis. It would be very nice to have experimental validation to corroborate some results.

Concerning model validation, please see our response to comment #3 of reviewer 2.

References

1. Long, A. M., Hou, S., Ignacio-Espinoza, J. C. & Fuhrman, J. A. Benchmarking microbial growth rate predictions from metagenomes. *ISME J.* **15**, 183–195 (2021).
2. Minnebo, Y. *et al.* Gut microbiota response to in vitro transit time variation is mediated by microbial growth rates, nutrient use efficiency and adaptation to in vivo transit time. *Microbiome* **11**, 240 (2023).
3. Meijnikman, A. S., Nieuwdorp, M. & Schnabl, B. Endogenous ethanol production in health and disease. *Nat. Rev. Gastroenterol. Hepatol.* **21**, 556–571 (2024).
4. Simic, M., Ajdukovic, N., Veselinovic, I., Mitrovic, M. & Djurendic-Brenesel, M. Endogenous ethanol production in patients with diabetes mellitus as a medicolegal problem. *Forensic Sci. Int.* **216**, 97–100 (2012).
5. Zimmermann, J., Kaleta, C. & Waschina, S. gapseq: informed prediction of bacterial metabolic pathways and reconstruction of accurate metabolic models. *Genome Biol.* **22**, 81 (2021).
6. Macias-Ceja, D. C. *et al.* Succinate receptor mediates intestinal inflammation and fibrosis. *Mucosal Immunol.* **12**, 178–187 (2019).
7. Borkum, J. M. The Tricarboxylic Acid Cycle as a Central Regulator of the Rate of Aging: Implications for Metabolic Interventions. *Adv. Biol.* **7**, 2300095 (2023).
8. Bar, N. *et al.* A reference map of potential determinants for the human serum metabolome. *Nature* **588**, 135–140 (2020).
9. Olecka, M. *et al.* Nonlinear DNA methylation trajectories in aging male mice. *Nat. Commun.* **15**, 3074 (2024).
10. Thiele, I. *et al.* Personalized whole-body models integrate metabolism, physiology, and the gut microbiome. *Mol. Syst. Biol.* **16**, e8982 (2020).
11. Ivanova, E. P., Alexeeva, Y. V., Pham, D. K., Wright, J. P. & Nicolau, D. V. ATP level variations in heterotrophic bacteria during attachment on hydrophilic and hydrophobic surfaces. *Int. Microbiol. Off. J. Span. Soc. Microbiol.* **9**, 37–46 (2006).

12. Spari, D. & Beldi, G. Extracellular ATP as an Inter-Kingdom Signaling Molecule: Release Mechanisms by Bacteria and Its Implication on the Host. *Int. J. Mol. Sci.* **21**, 5590 (2020).
13. Mepin, R. *et al.* Release of extracellular ATP by bacteria during growth. *BMC Microbiol.* **13**, 301 (2013).
14. Beghini, F. *et al.* Integrating taxonomic, functional, and strain-level profiling of diverse microbial communities with bioBakery 3. *eLife* **10**, e65088 (2021).
15. Bowers, R. M. *et al.* Minimum information about a single amplified genome (MISAG) and a metagenome-assembled genome (MIMAG) of bacteria and archaea. *Nat. Biotechnol.* **35**, 725–731 (2017).
16. Stewart, R. D. *et al.* Assembly of 913 microbial genomes from metagenomic sequencing of the cow rumen. *Nat. Commun.* **9**, 870 (2018).
17. Giordano, N. *et al.* Genome-scale community modelling reveals conserved metabolic cross-feedings in epipelagic bacterioplankton communities. *Nat. Commun.* **15**, 2721 (2024).

Responses to Reviewer Comments (our own response in *blue in italics*):

Reviewer #1 (Remarks to the Author):

I do appreciate the author's effort to thoroughly reply to my previously raised concerns through clarifications in the text, additional analysis, and the inclusion of new data.

Although, an experimental validation of a specific example of the model predictions of metabolic interactions (e.g. measurement of metabolite levels) is still not provided, the authors provide additional analyses to consolidate the validity of their computational models. In order to help the reader to better put these additional analyses into their biological context, I suggest the following additions to the manuscript:

1) The authors mention that they found literature evidence for 42 of the 83 predicted interactions. To better evaluate experimental nature (and quality) of these interactions, I propose that the authors add a supplementary table, in which each of the 42 previously reported interactions is linked to the primary literature together with a short description of it underlying experimental evidence.

We thank the reviewer for this suggestion. However, we have already included such a table in the previous revision (Supplementary Table S3.9) and referenced it in the text (l. 229 in the previous revision). There might have been some issues with the Supplementary Table itself since the online submission system automatically converts Excel files to PDF. We now have uploaded it as Supplementary Data which prevents this conversion.

2) For the prediction of the microbiome dependence of specific serum metabolites, I suggest to list the metabolites together with their microbiome dependency and their measured serum variance in a supplementary table.

We thank the reviewer for this comment. This information was already provided in Supplementary Table S3.10 as we indicated in the legend of Supplementary Figure 3B (the figure referenced in the main text containing the data). However, we now also reference the supplementary table in the main text for clarity.

3) I suggest using 'metabolic features' instead of 'metabolic clusters' also in Fig. 4F (and not only in the text).

We have updated the figure accordingly.

Editorial comments:

In addition to adding the supplementary information, as asked by R1, we'd also like you to add to the discussion why only male mice were used in the study - something we found peculiar during our editorial assessment.

We thank the editor for this comment. Indeed we are fully aware of this limitation which we already acknowledged in the previous submission. The original reason for not including female mice in our study were twofold. On the one hand, we wanted to avoid potential

confounding factors associated with sex-specific variability. On the other hand, considering also female mice would have required setting up a separate aging cohort since sex-specific effect can only properly be assessed in a fully stratified analysis ^(1,2). This is particularly challenging for the old age group (30 months) since only 10 - 15% of the animals achieve that age ⁽³⁾ and we hence would have required to age at least 100 animals for that age group alone. However, due to the importance of considering sex-specific effects of aging, we are already in the process of sampling animals from a female aging cohort which we will analyse in a future work. We now provide a more detailed discussion of this issue in the main manuscript (l. 407 - 411):

“ A further limitation of our study was the exclusive use of male mice, as the logistical challenges of establishing a separate aging cohort for females precluded the inclusion of both sexes. Focusing initially on males ensured consistency by avoiding sex-specific variability. Consequently, sex-specific changes were not investigated in this study but will be considered in future research.

”

and the methods (l. 447 - 454):

“In our study, we focused exclusively on male mice for two primary reasons. First, we aimed to minimize potential confounding factors arising from fluctuations in sex hormones in female mice, which are known to significantly influence metabolic processes across tissues during aging ⁸⁰. Second, addressing sex differences in aging would have required a fully stratified experimental design ³ and, consequently, a separate cohort of female mice. Given that only 10–15% of animals typically reach the age of 30 months, achieving comparable sample sizes and statistical power for the oldest age group alone would have necessitated approximately 100 female mice.”

References:

1. Shapiro, J. R., Klein, S. L. & Morgan, R. Stop ‘controlling’ for sex and gender in global health research. *BMJ Glob. Health* **6**, (2021).
2. Galea, L. A., Choleris, E., Albert, A. Y., McCarthy, M. M. & Sohrabji, F. The Promises and Pitfalls of Sex Difference Research. *Front. Neuroendocrinol.* **56**, 100817 (2020).
3. Frahm, C. *et al.* Transcriptional profiling reveals protective mechanisms in brains of long-lived mice. *Neurobiol. Aging* **52**, 23–31 (2017).